# Beyond Accuracy: What Matters in Designing Well-Behaved Image Classification Models?

**Robin Hesse**[*][†]                                              *rhesse@mpi-inf.mpg.de*
*Max Planck Institute for Informatics, SIC*

**Doğukan Bağcı**[*]                                    *dogukan.bagci@stud.tu-darmstadt.de*
*Department of Computer Science*
*Technical University of Darmstadt*
*Zuse School ELIZA*

**Bernt Schiele**                                                  *schiele@mpi-inf.mpg.de*
*Max Planck Institute for Informatics, SIC*
*Zuse School ELIZA*

**Simone Schaub-Meyer**                          *simone.schaub@visinf.tu-darmstadt.de*
*Department of Computer Science*
*Technical University of Darmstadt*
*hessian.AI*

**Stefan Roth**                                      *stefan.roth@visinf.tu-darmstadt.de*
*Department of Computer Science*
*Technical University of Darmstadt*
*hessian.AI*
*Zuse School ELIZA*

**Reviewed on OpenReview:** *https://openreview.net/forum?id=E7HDtLCoT6*

## Abstract

Deep learning has become an essential part of computer vision, with deep neural networks (DNNs) excelling in predictive performance. However, they often fall short in other critical quality dimensions, such as robustness, calibration, or fairness. While existing studies have focused on a subset of these quality dimensions, none have explored a more general form of "well-behavedness" of DNNs. With this work, we address this gap by simultaneously studying nine different quality dimensions for image classification. Through a large-scale study, we provide a bird's-eye view by analyzing 326 backbone models and how different training paradigms and model architectures affect these quality dimensions. We reveal various new insights such that *(i)* vision-language models exhibit high class balance on ImageNet-1k classification and strong robustness against domain changes; *(ii)* training models initialized with weights obtained through self-supervised learning is an effective strategy to improve most considered quality dimensions; and *(iii)* the training dataset size is a major driver for most of the quality dimensions. We conclude our study by introducing the QUBA score (**Q**uality **U**nderstanding **B**eyond **A**ccuracy), a novel metric that ranks models across multiple dimensions of quality, enabling tailored recommendations based on specific user needs.[1]

---

[1]Project page: `https://visinf.github.io/beyond-accuracy`; [*]equal contribution; [†]work done while at Technical University of Darmstadt

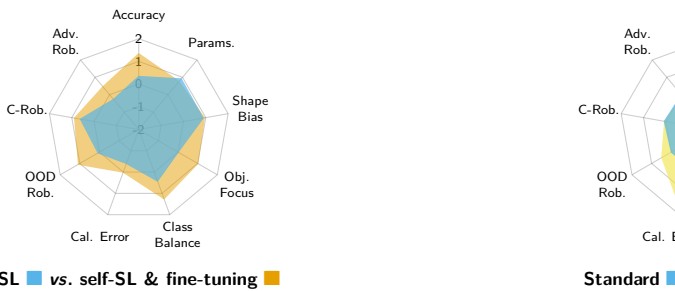

Figure 1: *Visualization of two of our main results.* We compare nine different quality dimensions for popular backbone models trained with standard supervised learning ■ (SL) against the corresponding backbones trained after initialization with weights obtained through self-supervised learning ■ *(left)* and when utilized in a vision-language (ViL) model ■ *(right)*. Axis units indicate the distance (in standard deviations) to the mean (0 line) of each quality dimension; see Eq. (1) and its explanation for details. Please refer to Tab. 2 (e) and (j) for raw values and Sec. 3 for an interpretation of the results.

# 1 Introduction

Today's computer vision research is heavily shaped by advances in the field of deep learning. While deep neural networks (DNNs) excel at predictive performance, often measured via the accuracy, it has been shown that they are flawed across various other quality dimensions, such as robustness (Goodfellow et al., 2015; Hendrycks & Dietterich, 2019), calibration (Guo et al., 2017), and fairness (Du et al., 2021). To address these challenges, the scientific community started various parallel streams of research focusing on *individual* aspects of DNN quality, developing mostly orthogonally. We argue that this orthogonal development is somewhat surprising, as the overarching goal for most applications should be the implementation of *more well-behaved* networks that excel in *many* quality dimensions (Liu et al., 2023). While some works study the relationship between a subset of quality dimensions, *e.g.*, how accuracy and calibration relate (Guo et al., 2017; Minderer et al., 2021), or how adversarial training (Goodfellow et al., 2015) improves calibration (Grabinski et al., 2022), we are not aware of any work that studies a broad range of quality dimensions *simultaneously*. Consequently, it is largely unknown how model improvements in one direction affect other quality dimensions. Here, we close this gap by studying how 326 backbone models perform along nine different quality dimensions for ImageNet-1k (Russakovsky et al., 2015) image classification. By doing so, we analyze how different training paradigms and model architectures can be used to improve these quality dimensions, uncover unknown connections between quality dimensions, and give recommendations on what models to use based on specific user needs. We expect our findings to be highly relevant for advancing the development of classification models that not only excel in accuracy but also across a wide range of DNN quality dimensions.

Our contributions can be summarized as: *(1)* We introduce a novel benchmark to measure a broad range of quality dimensions simultaneously, which is compatible with *any* DNN/backbone that performs ImageNet-1k (Russakovsky et al., 2015) classification. *(2)* In a large-scale study, we evaluate how 326 backbone models from prior work perform along nine considered quality dimensions. *(3)* We use this to analyze how different training paradigms and architectural changes can be utilized to improve the different quality dimensions. Among other things, we find that self-supervised pre-training followed by fine-tuning improves most quality dimensions and that vision-language models achieve high fairness (here measured as the class balance) on ImageNet-1k classification while being fairly robust against domain changes (see Fig. 1). *(4)* Building on trends in related work that examine relationships between individual quality dimensions (*e.g.*, Guo et al., 2017; Miller et al., 2021; Minderer et al., 2021), we analyze the relationships among *all* considered quality dimensions. *(5)* We conclude our study by introducing a novel *QUBA score* (**Q**uality **U**nderstanding **B**eyond **A**ccuracy) that ranks models across multiple dimensions of quality. We use this score to recommend top-performing models tailored to diverse user needs.

Table 1: *Overview of our considered DNN quality dimensions for image classification.* Arrows indicate if higher (↑) or lower (↓) is better. If a quality dimension is computed over multiple datasets or metrics, we use the geometric mean.

| Quality Dimension | Description (visual illustrations are provided in Appendix A) |
|---|---|
| Accuracy (↑) | Fraction of correctly classified (clean) images |
| Adversarial Robustness (↑) | Fraction of correctly classified images after an FGSM or PGD attack (normalized by clean accuracy) |
| C-Robustness (↑) | Fraction of correctly classified images after corrupting images (normalized by clean accuracy) |
| OOD Robustness (↑) | Fraction of correctly classified images from different domains (normalized by clean accuracy) |
| Calibration Error (↓) | Misalignment of the output confidence and the true probability of a correct classification |
| Class Balance (↑) | Standard deviation of the accuracies and average confidences across all individual classes |
| Object Focus (↑) | Fraction of decisions that are based on foreground and not on background |
| Shape Bias (↑) | Fraction of decisions that are based on shape and not on texture |
| Parameters (↓) | Number of parameters |

## 2 Evaluating quality beyond accuracy

We go beyond accuracy by exploring the general "well-behavedness" of DNNs. While this term is inherently ill-defined and task-dependent, we use it *informally* as the performance across the nine quality dimensions chosen in this study. Specifically, we consider *(1)* accuracy; three robustness metrics: *(2)* adversarial robustness, *(3)* corruption robustness, and *(4)* out-of-domain robustness; *(5)* calibration error; *(6)* fairness measured via class balance; two dimensions concerned with shortcut-learning: *(7)* object focus and *(8)* shape bias; and *(9)* computational cost measured via the number of parameters. These dimensions encompass a wide range of DNN qualities/properties and attracted considerable attention in related work, which is why we regard them as particularly important. We evaluate these quality dimensions simultaneously by merging corresponding evaluation protocols into a single, comprehensive benchmark. To enable a large-scale study with a feasible computational load, we only select protocols that require ImageNet-1k (Russakovsky et al., 2015) classification without model fine-tuning. While there are other important dimensions, such as explainability and out-of-distribution detection capabilities, they are challenging to evaluate as they are coupled to specific explanation/out-of-distribution detection methods that might vary across backbones (Hesse et al., 2023b) – thus, we excluded them. Also, we note that all evaluation protocols are merely proxies for the targeted dimension; they do not necessarily reflect true performance in that dimension, nor do they capture the full complexity of the underlying behavior. Consequently, different evaluation protocols may lead to different conclusions, which is worth keeping in mind when interpreting our findings.

We now outline related work, the considered quality dimensions, and how we measure them in this work; see Tab. 1 for a summary and Appendix A for more details.

**Accuracy.** The success of DNNs is largely driven by their superior accuracy, first showcased in 2012 by AlexNet (Krizhevsky et al., 2012) on ImageNet-1k (Russakovsky et al., 2015). This marked the beginning of the *"deep learning era,"* leading to increasingly powerful models (Dosovitskiy et al., 2021; He et al., 2016; Liu et al., 2022b; Simonyan & Zisserman, 2015). To measure a model's accuracy, we report the top-1 accuracy on the ImageNet-1k evaluation split.

**Adversarial robustness.** DNNs are vulnerable to adversarial attacks, *i.e.*, small perturbations in the input space (Szegedy et al., 2014). This vulnerability can be reduced by training with adversarial examples (Goodfellow et al., 2015; Madry et al., 2018) or by defensive distillation (Papernot et al., 2016). To assess adversarial robustness, we measure the geometric mean[2] of the accuracies after applying two popular attacks, FGSM (Goodfellow et al., 2015) and PGD (Madry et al., 2018). To reduce the dependence on the clean accuracy of the model, we report adversarial robustness *relative* to the clean ImageNet-1k accuracy. In Appendix C.4, we compare our adversarial robustness measure to *AutoAttack* (Croce et al., 2021), yielding very similar conclusions.

**Corruption robustness.** DNNs are susceptible to common image corruptions such as JPEG compression and contrast changes (Hendrycks & Dietterich, 2019), which can be reduced with special kinds of data augmentations (Hendrycks et al., 2020) or self-supervised learning (Hendrycks et al., 2019). To

---

[2]The geometric mean ensures that metrics of different scales contribute equally without one overshadowing the others.

assess a model's robustness to common corruptions (C-robustness), we measure the mean accuracy on ImageNet-C (Hendrycks & Dietterich, 2019), *i.e.*, the ImageNet evaluation split with different corruption types of increasing strength. To normalize C-robustness and to be consistent with our other robustness metrics, we again report the top-1 accuracy on the corrupted data *relative* to the clean ImageNet-1k accuracy.

**OOD robustness.** Out-of-domain (OOD) robustness is concerned with the generalizability of a model to OOD data (*e.g.*, Geirhos et al., 2019; Hendrycks et al., 2021; Wang et al., 2019). Contrary to adversarial and corruption robustness, datasets to assess the OOD robustness exhibit stronger visual domain shifts and contain new data samples. We assess OOD robustness by reporting the geometric mean of the relative accuracy on five established OOD datasets: ImageNet-R (Hendrycks et al., 2021), ImageNet-Sketch (Wang et al., 2019), as well as Stylized-ImageNet, Edge, and Silhouette from Geirhos et al. (2019).

**Calibration error.** Calibration measures how well a model's output confidence reflects the probability of a correct prediction. Guo et al. (2017) found DNNs to be poorly calibrated, spurring the developments of approaches like deep ensembles (Lakshminarayanan et al., 2017) and label smoothing (Müller et al., 2019). We report the geometric mean of the expected calibration error (ECE) (Guo et al., 2017; Nixon et al., 2019) and the adaptive calibration error (ACE) (Nixon et al., 2019).

**Class balance.** A well-behaved model should behave fairly. Fairness can, *e.g.*, be improved by the class-wise weighting of the cross-entropy loss (Benz et al., 2020). As fairness has multiple facets (Verma & Rubin, 2018) and there is no standardized fairness metric, we rely on a simplified notion of "class balance": no class should be particularly favored or disadvantaged (Benz et al., 2020; Kuzucu et al., 2024). More specifically, we evaluate the class balance of a model in two ways: *(1)* the standard deviation of ImageNet-1k class accuracies, similar to Croce et al. (2021), and *(2)* the standard deviation of average class confidences, similar to Kuzucu et al. (2024). To align with other metrics, we subtract these values from 1, so higher scores indicate greater class balance, and we aggregate both measures using the geometric mean. While aiming to approximate fairness, we note that ImageNet-1k itself exhibits dataset biases, and thus our conclusions about fairness may not fully reflect real-world concerns.

**Object focus.** ImageNet-trained DNNs rely on background features to the extent that they can be fooled by changing the background (Xiao et al., 2021; Zhu et al., 2017). This can be avoided by training on images where the background signals are decorrelated from the class labels (Xiao et al., 2021). Similar to Xiao et al. (2021), we measure object focus by assessing accuracy drops when replacing backgrounds with those from other classes.

**Shape bias.** ImageNet-trained CNNs exhibit a texture bias rather than a shape bias (Geirhos et al., 2019). Since this can hurt generalizability and robustness, models with increased shape bias are preferred and have been developed (Nam et al., 2021; Shi et al., 2020). The shape bias is measured using images with a conflict between the shape and texture cues – *e.g.*, an image of a cat (shape) with the skin of an elephant (texture). These images are fed into the model to determine whether it predicts based on texture (texture bias) or shape (shape bias) (Geirhos et al., 2019).

Unlike the other quality dimensions, a higher shape bias is not inherently better, as some applications may benefit from a stronger focus on texture. Nonetheless, we include shape bias as a dimension and, for consistency with related work and since it is more in line with human vision (Geirhos et al., 2019; Wang et al., 2020), assume that higher values are preferable. This assumption does not affect the core of our analysis. Further, in Sec. 5, we introduce the QUBA score, where the shape bias weight can be adjusted or even inverted to reflect specific preferences.

**Parameters.** Deep neural networks (DNNs) should be memory-efficient and fast to reduce resource consumption and operate sustainably. Since actual memory and runtime performance depend heavily on specific implementations and hardware, we follow established practice and use the number of model parameters as a hardware- and implementation-independent

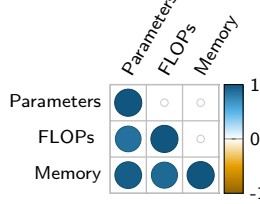

Figure 2: *Rank correlation matrix for the considered metrics on computational cost for our full model zoo of 326 models. All entries have a p-value below 0.05, indicating statistical significance.*

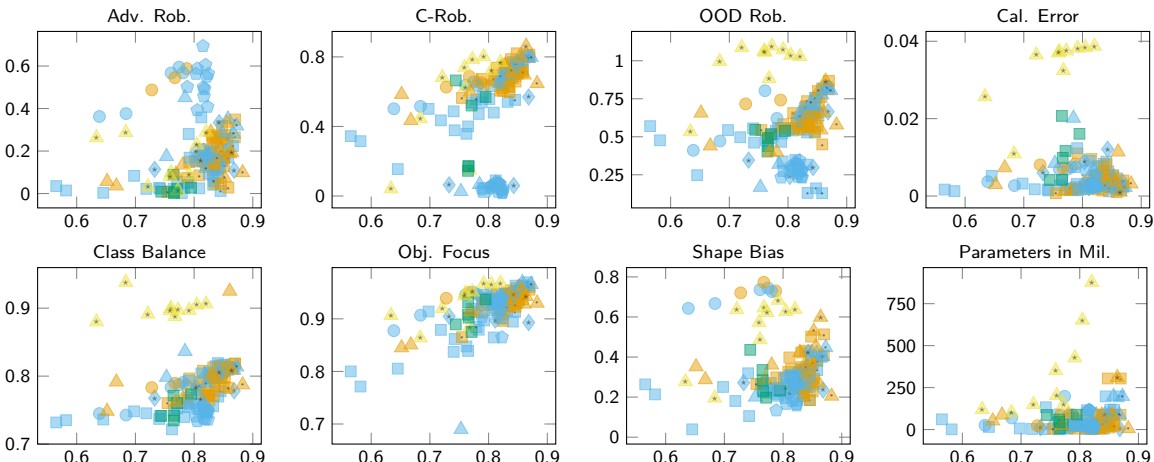

Figure 3: *Different quality dimensions (y axis) vs. accuracy (x axis).* To reduce clutter in the plots, we only plot representative models instead of our full model zoo; please refer to the project page for interactive plots with all models. To emphasize the effect of different training strategies and model architectures, we group models visually: the training dataset size is marked by symbols within each marker (no symbol for ImageNet-1k, dot (·) for ImageNet-21k, star (⋆) for large-scale datasets); different training strategies by shapes (standard supervised training as squares ■, adversarial training as circles ●, self-supervised (pre-)training as triangles ▲, semi-supervised training as diamonds ◆, A[1,2,3] training as pentagons ⬠); and different architectures by color (blue ■ for CNNs, orange ■ for Transformers, green ■ for B-cos models, and yellow ■ for vision-language (ViL) models).

proxy for computational cost (Kaplan et al., 2020; Tay et al., 2022). To

validate its suitability as a proxy, we visualize the Spearman's rank correlation for 326 models (see Appendix E.3) between the number of parameters, required memory, and theoretical FLOPs in Fig. 2. All three measures are strongly correlated (correlation coefficients > 0.85), confirming that the number of parameters is a reliable proxy. Thus, for practical purposes and ease of comparison in future studies (memory consumption is implementation dependent, and the computation for the theoretical number of FLOPs needs to be adjusted for novel model components), we adopt the number of parameters as our primary metric. However, we note that parameter count alone cannot fully capture the efficiency of a model and should be interpreted with caution (Dehghani et al., 2022). Various studies are concerned with reducing the number of parameters or the computational cost of DNNs while keeping the accuracy as high as possible (Hesse et al., 2023a; Li et al., 2022b; Tan & Le, 2019; 2021).

## 3 What makes a model more well-behaved?

Equipped with the above quality metrics, we now study how different design choices affect DNN quality. This not only sheds light on *quality beyond accuracy* of the current state of the art for image classification, but also facilitates the development of more well-behaved DNNs in the future. In Tab. 2, we compare the average of each quality dimension for different setups. Since not all backbones can be considered in each configuration (*e.g.*, not all have adversarially trained variants), we ensure fair comparisons by selecting the subset of models that is consistently available for all configurations of the respective setup (with no or minor modifications; see Appendix D.1). In Fig. 3, we plot the different quality dimensions for selected models against accuracy, distinguishing between model groups.

**Experimental setup.** We use 326 publicly available models in our large-scale study. See Appendix E.3 for an overview, including numerical results and implementation details for each quality dimension.

### 3.1 Different training strategies

**Training dataset size.** We compare models trained on ImageNet-1k (Russakovsky et al., 2015) to those trained on the larger ImageNet-21k (Deng et al., 2009) dataset in Fig. 3 (no symbol *vs.* dot (·)) and Tab. 2 (a) and (b). Training on a larger dataset improves nearly all quality dimensions for CNNs and Transformers, particularly accuracy, C-robustness, and calibration – likely by promoting more general and less overfitted features. Interestingly, with a larger training dataset, the adversarial robustness of Transformers decreases.

> **Conclusion:** Training on a larger dataset improves most of the considered quality dimensions.

**Adversarial training.** We compare adversarially trained models against the corresponding standard supervised models in Fig. 3 (circles ● *vs.* squares ▪) and Tab. 2 (c). Adversarial training (AT) improves shape bias (Engstrom et al., 2019; Geirhos et al., 2021; Wang et al., 2020), OOD robustness (Engstrom et al., 2019), and, expectedly, adversarial robustness. Accuracy and class balance significantly worsen with AT. The latter is in line with results of Benz et al. (2020); Xu et al. (2021a). Interestingly, we observe a trend of increasing calibration error with AT, which extends findings in (Grabinski et al., 2022) that found evidence that adversarially trained ResNets (He et al., 2016) exhibit improved calibration errors.

> **Conclusion:** AT improves adversarial/OOD robustness and shape bias. It impairs accuracy and class balance.

**Self-supervised training.** Self-supervised learning eliminates the need for dataset annotations and thus allows for training on significantly larger datasets. We compare models initialized with weights obtained through self-supervised learning to standard supervised models in Fig. 3 (triangles ▲ *vs.* squares ▪) and Tab. 2 (d) and (e). We consider self-supervised models in two standard transfer settings: *(i)* models where only the final classification layer is trained (linear probing, LP) and *(ii)* models that are fully fine-tuned on ImageNet-1k (E2E). We analyze how each approach affects the different quality dimensions. LP models (Tab. 2 (d)) generally underperform compared to supervised models, except in OOD robustness, calibration, and shape bias – likely due to the larger gap between training and testing distributions. The reduced object focus is notable, as Caron et al. (2021) found that self-supervised Transformers produce attention maps that closely align with objects. This suggests that attention maps may not serve as reliable explanations, a finding consistent with (Hesse et al., 2023b; 2024). The slight parameter difference stems from DINOv2 (Oquab et al., 2024) using a $14 \times 14$ patch size instead of the original $16 \times 16$.

On the other hand, fine-tuning (E2E) self-supervised models (Tab. 2 (e)) improves most quality dimensions (except calibration) – probably due to larger pre-training datasets typically used for self-supervised learning and a smaller domain gap than LP. The improvement in class balance is particularly surprising, as we expected the larger training datasets used for self-supervised training to have much stronger class imbalances than the evenly distributed ImageNet dataset. We hypothesize that the class balance is nonetheless improved because the self-supervised models are pre-trained without any class information, resulting in features less tailored to specific classes. As a result, these features yield a more balanced distribution of class accuracies and class confidences.

> **Conclusion:** Self-supervised models with linear probing perform worse than supervised models in most quality dimensions. Fine-tuning self-supervised models in an end-to-end fashion improves most quality dimensions.

**Semi-supervised training.** We also measure how semi-supervised training (Xie et al., 2020; Yalniz et al., 2019), *i.e.*, training on a combination of labeled and unlabeled data, compares to supervised training in Fig. 3 (diamonds ♦ *vs.* squares ▪) and Tab. 2 (f). Among the dimensions with statistically significant changes, semi-supervised training has similar effects as self-supervised training with E2E fine-tuning, probably also due

Table 2: *Average quality dimensions for models with different configurations.* We evaluate various setups, focusing on different training strategies (a–g) and different architectural choices (h–j). In each configuration, we report the average score for each quality dimension across the models associated with that configuration. As different models are available for different setups, each setup considers a distinct selection of models being compared. As a result, both the models and their total number (indicated by the number beside each setup) vary across setups to maintain a fair basis for comparison. The number of asterisks represents the statistical significance of differences in the average scores of a quality dimension across configurations within each setup: *** for $p < 0.05$, ** for $p < 0.1$, and * for $p < 0.2$, based on $t$-test results.

| Setup | Configuration | Acc.↑ | Adv. Rob.↑ | C-Rob.↑ | OOD Rob.↑ | Cal. Error↓ | Class Balance↑ | Obj. Focus↑ | Shape Bias↑ | Params. in Mil.↓ |
|---|---|---|---|---|---|---|---|---|---|---|
| (a) 14 | CNNs (IN-1k) | 0.82 | 0.14 | 0.67 | 0.60 | 0.0048 | 0.80 | 0.93 | 0.29 | 69 |
| | CNNs (IN-21k) | **0.84**\* | **0.15** | **0.71** | 0.60 | **0.0026**\*\*\* | 0.80 | **0.95** | **0.33**\* | 69 |
| (b) 16 | Transformers (IN-1k) | 0.81 | **0.21** | 0.69 | 0.61 | 0.0046 | 0.79 | 0.94 | 0.39 | 85 |
| | Transformers (IN-21k) | **0.84**\*\*\* | 0.17\* | **0.75**\*\* | **0.69**\*\* | **0.0027**\*\*\* | **0.80**\*\* | **0.95** | 0.42 | 85 |
| (c) 11 | Supervised models | **0.82** | 0.12 | **0.66** | 0.60 | 0.0056 | 0.80 | **0.94** | 0.29 | 86 |
| | Adversarially trained models | 0.74\*\*\* | **0.52**\*\*\* | 0.62 | **0.63** | 0.0068\* | 0.78\*\* | 0.93 | **0.72**\*\*\* | 86 |
| (d) 13 | Supervised models | **0.81** | 0.20 | **0.66** | 0.58 | 0.0047 | **0.81** | **0.93** | 0.34 | **89** |
| | Self-supervised models (LP) | 0.75\* | 0.10\*\*\* | 0.61 | **0.59** | **0.0029**\*\* | 0.78\*\* | 0.88\*\* | **0.40**\* | 91 |
| (e) 25 | Supervised models | 0.81 | 0.16 | 0.67 | 0.58 | **0.0034** | 0.79 | 0.93 | 0.38 | **94** |
| | Self-supervised models (E2E) | **0.84**\*\*\* | **0.24**\*\*\* | 0.73 | **0.73**\*\*\* | 0.0045 | **0.81**\*\*\* | **0.95**\*\*\* | **0.39**\* | 95 |
| (f) 13 | Supervised models | 0.80 | 0.13 | **0.58** | 0.59 | 0.0048 | 0.78 | 0.92 | 0.24 | 28 |
| | Semi-supervised models | **0.82**\* | **0.20**\*\*\* | 0.45\* | 0.59 | 0.0059 | **0.80**\* | 0.93 | **0.29**\*\*\* | 28 |
| (g) 19 | Supervised models | 0.79 | 0.12 | **0.50** | **0.52** | 0.0044 | **0.78** | 0.92 | 0.25 | 39 |
| | A1 supervised models (600 epochs) | 0.80 | **0.47**\*\*\* | 0.06\*\*\* | 0.31\*\*\* | **0.0030**\* | 0.75\*\*\* | 0.92 | **0.27**\*\* | 39 |
| | A2 supervised models (300 epochs) | 0.80 | 0.41\*\*\* | 0.06\*\*\* | 0.30\*\*\* | **0.0026**\*\*\* | 0.75\*\*\* | **0.93**\*\*\* | 0.25 | 39 |
| | A3 supervised models (100 epochs) | 0.78 | 0.32\*\*\* | 0.04\*\*\* | 0.27\*\*\* | 0.0048 | 0.76\*\*\* | 0.91 | 0.17\*\*\* | 39 |
| (h) 46 | CNNs | 0.81 | 0.11 | 0.62 | 0.54 | 0.0048 | 0.79 | 0.92 | 0.29 | 40 |
| | Transformers | 0.81 | **0.20** | **0.69**\*\*\* | **0.62**\*\*\* | **0.0046** | **0.80**\* | **0.93**\*\*\* | 0.32 | 40 |
| (i) 12 | Standard models | **0.77** | **0.07** | **0.58** | **0.56** | 0.0033 | **0.77** | 0.93 | 0.27 | 37 |
| | B-cos models | 0.75\* | 0.02\*\*\* | 0.26\*\*\* | 0.46\*\*\* | 0.0115\*\*\* | 0.75\* | 0.90\*\*\* | 0.27 | **36** |
| (j) 24 | Standard models | **0.81** | **0.18** | **0.62** | 0.56 | **0.0044** | 0.79 | 0.93 | 0.35 | **152** |
| | Vision-language models | 0.74\*\*\* | 0.10\* | 0.60 | **1.00**\*\*\* | 0.0337\*\*\* | **0.90**\*\*\* | 0.93 | **0.56**\*\*\* | 275\*\*\* |

to the combination of a large-scale training dataset and relatively close training and testing domains. Only C-robustness is negatively affected statistically significantly.

> **Conclusion:** Semi-supervised training improves accuracy, adversarial robustness, class balance, and shape bias. Only C-robustness is clearly impaired.

**A[1,2,3] training.** Wightman et al. (2021) introduce several training strategies, termed A1, A2, and A3, which incorporate best practices for training DNNs – *e.g.*, multi-label classification objectives, data augmentation techniques, and the use of advanced optimizers. Most importantly, the three strategies vary in their training duration: A1 is trained for 600, A2 for 300, and A3 for 100 epochs. We compare the training strategies to standard supervised models in Fig. 3 (pentagons ⬠ *vs.* squares ■) and Tab. 2 (g). While some training strategies of the standard supervised models might overlap with the A[1,2,3] training, the long training of A1 is not utilized in any of the standard models. The accuracy is slightly increasing for the setups with increased training times (A[1,2]; statistically insignificant). Interestingly, adversarial robustness significantly improves with the A[1,2,3] training, while C-robustness and OOD robustness decrease. We believe the improved training enhances adversarial robustness by expanding the distance between decision boundaries and data points, but reduces generalizability by encouraging "overfitting" to the training distribution. Calibration error decreases for the setups with increased training times (A[1,2]), which extends findings of Minderer et al. (2021) that showed that calibration error increases with longer training when measured only on BiT models (Kolesnikov et al., 2020). Class balance is reduced for all the setups; object focus remains fairly stable, and the shape bias increases with longer training, confirming results of Hermann et al. (2020).

> **Conclusion:** Adversarial robustness, calibration, and shape bias improve with longer training times. C/OOD-robustness and class balance are impeded.

## 3.2 Different model designs

Now that we have covered various training strategies and their effect on different quality dimensions, we analyze the effect of specific architectural choices.

**Is the time of CNNs over?** We compare models based on convolutions (CNNs) and attention (Transformers) in Fig. 3 (blue ■ *vs.* orange ■) and Tab. 2 (h). Since Vision Transformers (Dosovitskiy et al., 2021) were introduced only in 2020, we exclude CNNs proposed before 2020 and compare only newer CNN architectures with Transformers for a fairer evaluation. To further improve the fairness of our comparison, we make sure that we have an equal number of CNNs and Transformers from different setups (*e.g.*, adversarial training) and only compare them when they have a similar number of parameters (within a 1-million difference). Despite our efforts to ensure a balanced comparison, this setup gives us less control over certain variables than our other experiments. Therefore, these results should be interpreted with caution. Remarkably, CNNs and Transformers perform equally in accuracy. However, Transformers outperform CNNs in all the other quality dimensions. Our results on robustness nicely complement those of Bai et al. (2021), who compared the robustness of CNNs and Transformers but considered only ResNet50 (He et al., 2016) and DeiT-S/16 (Touvron et al., 2021a).

> **Conclusion:** Transformers consistently outperform CNNs across almost all quality dimensions.

**B-cos transform.** Initially introduced to improve interpretability, the B-cos transform (Böhle et al., 2022) can substitute the linear transformations in a DNN. It encourages the weights to align with the input and potentially affects the model beyond the improved interpretability. We thus analyze B-cos models in Fig. 3 (green ■) and compare them to the corresponding standard models in Tab. 2 (i). Besides shape bias and the number of parameters, all considered quality dimensions drop significantly when using the B-cos transform. A potential reason for this is the inductive bias of weight-input alignment, limiting the model's expressiveness.

> **Conclusion:** The B-cos transform negatively affects most of the considered quality dimensions.

**Vision-language (ViL) models.** With ViL models becoming increasingly relevant, we study their performance across the considered quality dimensions in Fig. 3 (yellow ■) and compare them to their corresponding backbones trained in a supervised fashion in Tab. 2 (j). Please note that Tu et al. (2023) conducted a similar study, however, focusing exclusively on CLIP models (Radford et al., 2021) and covering a slightly different set of quality dimensions. Since ViL models perform zero-shot classification by mapping the 1000 ImageNet-1k class labels into their feature space and then predicting the class label closest to the feature embedding of the given image, their accuracy is notably lower than that of the supervised models. Also, they contain significantly more parameters due to the additional language encoder. They exhibit decreased adversarial robustness and C-robustness while strongly improving OOD robustness (Radford et al., 2021). At first glance, one might attribute the improved OOD robustness to the models being trained on significantly larger datasets that include domains similar to those in the OOD datasets (Liu et al., 2023). While this is certainly a factor (Mayilvahanan et al., 2025), a closer look reveals that ViL models still outperform other models trained on similarly large datasets (see Appendix C.2), suggesting that they offer advantages beyond just dataset size. While Minderer et al. (2021) found that CLIP is fairly well calibrated when trained on WebImageText (WIT) (Radford et al., 2021), Tu et al. (2023) found that CLIP calibration can decrease when trained on other datasets. We extend their finding by observing that other ViL models also exhibit significantly worse calibration than standard models. Class balance and shape bias improve by a large margin – the former probably for similar reasons as for self-supervised models.

> **Conclusion:** ViL models excel in OOD robustness, class balance, and shape bias. However, they fall behind in accuracy (zero-shot), calibration, and parameters.

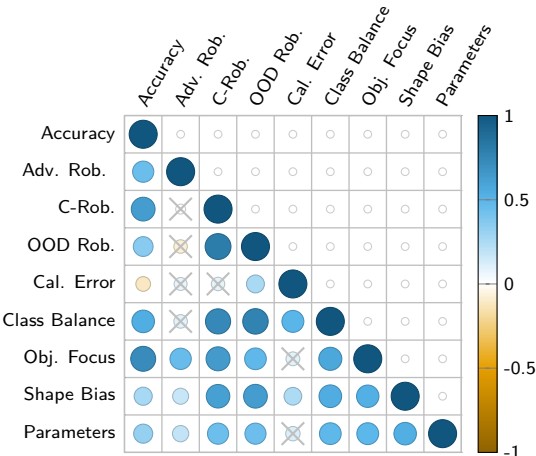

Figure 4: *Rank correlation matrix for the considered quality dimensions among our full model zoo.* All non-crossed-out entries have a *p*-value below 0.05, indicating statistical significance. Crossed-out entries correspond to *p*-values above 0.05 and are therefore not statistically significant.

## 4 Relationships between quality dimensions

**Comparison to related work.** While most previous work is concerned with *improving* quality dimensions, there are also studies examining their *relationships*. However, not all relationships have been explored, and prior studies used fewer and older models, which can lead to contradictory findings (see Appendix C.3). To address this gap, we investigate the relationship between *numerous* quality dimensions for our *extensive* model zoo, plotting the Spearman's rank correlation matrix for all nine considered quality dimensions across 326 models in Fig. 4 – please refer to Appendix C.1 for correlation matrices of specific model subgroups. Our analysis confirms that accuracy is positively correlated with OOD robustness (Miller et al., 2021), object focus (Xiao et al., 2021), shape bias (Hermann et al., 2020), and with the number of parameters (Liu et al., 2025). The number of parameters positively correlates with OOD robustness (Liu et al., 2025) and adversarial robustness (Madry et al., 2018; Nakkiran, 2019). Increasing the shape bias improves adversarial robustness (Geirhos et al., 2021; Jo & Bengio, 2017), accuracy (Geirhos et al., 2019), and OOD robustness (Geirhos et al., 2019). Accuracy and calibration error exhibit a negative correlation, aligning with Minderer et al. (2021), who found a negative correlation in more recent Transformer models, and contradicting Guo et al. (2017), who observed a positive correlation between these metrics in older backbone models. Unlike hypothesized by Tsipras et al. (2019) and confirming Yang et al. (2020), accuracy and adversarial robustness are positively correlated. Contrary to Liu et al. (2025) and Jo & Bengio (2017), adversarial robustness is not statistically significantly correlated with C-robustness and OOD robustness. While Grabinski et al. (2022) found that adversarial training improves calibration, we find no statistically significant correlation between adversarial robustness and calibration.

> **Conclusion:** We provide a bigger picture of related work using our extensive model zoo to validate known quality relationships, resolve conflicting findings, and extend recent findings regarding a link between adversarial and OOD robustness / C-robustness / calibration.

**Discovering new relationships.** While we cannot discuss all findings in Fig. 4, some insights – to our knowledge – have not been reported for backbone models in image classification. For example, accuracy and class balance are strongly correlated, meaning higher-accuracy models have less discrepancy between the best and worst-performing classes. Further, object focus is strongly correlated with all quality dimensions but the calibration error, rendering models with improved object focus an interesting research direction. This may be because models with greater object focus capture fewer surface-level statistical regularities and instead develop higher-level conceptual understanding (Jo & Bengio, 2017). Interestingly, calibration error is only

Table 3: *QUBA score and quality dimensions for the five top-performing models.* The configuration lists the architecture, training dataset, and training paradigm. [†] indicates models trained with knowledge distillation.

| Model | Configuration | QUBA Score↑ | Acc.↑ | Adv. Rob.↑ | C-Rob.↑ | OOD Rob.↑ | Cal. Error↓ | Class Balance↑ | Obj. Focus↑ | Shape Bias↑ | Params. in Mil.↓ |
|---|---|---|---|---|---|---|---|---|---|---|---|
| EfficientNet-B6 (Xie et al., 2020) | CNN, JFT-300M (Hinton et al., 2015); (Sun et al., 2017) + IN1k, semi-SL[†] | 0.94 | 0.86 | 0.25 | 0.77 | 0.83 | 0.0048 | 0.82 | 0.95 | 0.35 | **43** |
| Hiera-B (Ryali et al., 2023) | Transformer, IN1k, self-SL (E2E) | 0.95 | 0.85 | 0.23 | 0.76 | 0.76 | 0.0130 | 0.93 | 0.94 | 0.34 | 51 |
| ConvNeXtV2-B (Woo et al., 2023) | CNN, IN21k, self-SL (E2E) | 0.96 | 0.87 | **0.28** | 0.79 | 0.82 | **0.0023** | 0.81 | 0.96 | 0.40 | 88 |
| Hiera-B-Plus (Ryali et al., 2023) | Transformer, IN1k, self-SL (E2E) | 1.03 | 0.85 | 0.24 | 0.78 | 0.74 | 0.0130 | **0.93** | 0.95 | **0.43** | 69 |
| EVA02-B/14 (Fang et al., 2024b) | Transformer, IN21k, self-SL (E2E) | **1.08** | **0.88** | 0.21 | **0.81** | **0.86** | 0.0039 | 0.83 | **0.97** | 0.34 | 87 |

statistically significantly correlated with OOD robustness, class balance, and shape bias, highlighting the need for dedicated calibration research. Lastly, most considered quality dimensions (excluding the number of parameters and calibration) improve together, indicating that developing models that excel in a wide range of quality dimensions is feasible. This observation aligns with Liu et al. (2023), who argue that many desirable properties of trustworthy machine learning are underpinned by shared foundations, suggesting that improvements in one aspect may benefit others.

> **Conclusion:** Accuracy and class balance are strongly correlated, object focus is strongly correlated with most quality dimensions, calibration error is not correlated with most quality dimensions, and there are only a few trade-offs between the considered dimensions.

## 5 Which backbone to use?

We conclude our analysis by ranking models to provide recommendations on the best model choices. Ranking models across multiple quality dimensions is a non-trivial task with no one-size-fits-all solution, as user priorities vary depending on specific needs. Nonetheless, we aim to identify models that perform well across a wide range of dimensions and, thus, require an effective way to summarize the different quality scores with flexible weightings to reflect different user needs.

Probably the most straightforward way to summarize our results would be to take the average of all quality dimensions. However, given that these dimensions have vastly different ranges and scales, this approach would not treat all dimensions fairly. Another alternative would be to compute the mean rank: for each quality dimension, the models are ranked, and then the geometric mean of these individual ranks is calculated. However, using ranks has two key limitations. First, ranks are uniformly distributed, whereas the raw scores are not, meaning that the difference in mean rank between two models would not accurately capture the actual difference in their model quality. Second, if future studies introduce new models, the set of models will change, altering most of the rankings. As a result, mean ranks would no longer be consistent across different papers.

**QUBA score.** To address these issues, we leverage an intriguing property of our large model zoo: its size makes it representative of a broad range of models, enabling us to estimate a meaningful mean $\mu_i$ and standard deviation $\sigma_i$ for each quality dimension $i$ (we exclude the bottom and top 10% models to reduce outlier sensitivity). We then express each model's quality scores $s_i^{\text{model}}$ in terms of how many standard deviations they deviate from the mean. The final *QUBA score* (**Q**uality **U**nderstanding **B**eyond **A**ccuracy) for a model is the weighted arithmetic mean of these scores:

$$\text{QUBA}_{\text{model}} = \left(\frac{1}{\sum_{i=1}^{9} w_i}\right) \sum_{i=1}^{9} w_i \frac{s_i^{\text{model}} - \mu_i}{\sigma_i}. \tag{1}$$

By default, we use a balanced weighting where the three robustness dimensions are weighted at $w_i = 1/3$ to prevent them from overshadowing the results and, similarly, assign $w_i = 1/2$ to object focus and shape bias,

as both are related to shortcut learning. All other weights are set to 1 (different weightings are analyzed below). Since calibration errors and the number of parameters should be as small as possible, they are multiplied by $-1$ before computing the mean, so that higher values indicate better performance. Intuitively, the QUBA score reflects how many standard deviations a model deviates from the "average model" across the considered dimensions.

Our approach solves both limitations of the mean rank: the distances now have a consistent and meaningful interpretation, and the score can be calculated independently of the considered model set (since the mean and standard deviation for each quality dimension are assumed to be fixed).

To validate that our model zoo is large enough to produce reliable estimates of the mean and standard deviation for each quality dimension – and to assess the robustness of the QUBA score – we randomly sample 100 models from our full model zoo and compute the QUBA mean and standard deviation. This process is repeated five times. For each of the five resulting QUBA variants, we rank all 326 models and compute the rank correlation between the resulting rankings. The average rank correlation is very high (0.97), indicating that the QUBA rankings are stable and not overly dependent on the specific subset of models used.

**The best models.** We report results for the top five QUBA score models in Tab. 3. Of our 326 models, EVA02-B/14 (IN21k) (Fang et al., 2023) achieves the best QUBA score. Compared to the other top-performing models, it achieves the highest accuracy, C-robustness, OOD robustness, and object focus. It lags behind in adversarial robustness, calibration, class balance, shape bias, and the number of parameters. The second-best model, Hiera-B-Plus (Ryali et al., 2023), ranks lower in accuracy and calibration but performs well in the other dimensions, excelling in class balance and shape bias. In third place, the convolutional model ConvNeXtV2-B (IN21k) (Woo et al., 2023) leads in adversarial robustness and calibration while achieving good results in all other dimensions but the parameter count. The last two models, Hiera-B (Ryali et al., 2023) and EfficientNet-B6 (Xie et al., 2020) have a particularly low parameter count. Remarkably, all five models have been trained semi- or self-supervised, making these promising training paradigms for developing well-behaved models. Our analysis highlights that even the five top-performing models vary strongly among the quality dimensions, highlighting the need to consider a wide range of quality dimensions simultaneously in the design process of new models.

> **Conclusion:** The models with the highest QUBA scores excel across various quality dimensions, with each model showcasing distinct strengths.

**A closer look at popular models.** There are many popular models that did not make it into the top five above. We here go over some of the most popular models and briefly discuss their performance according to the considered quality dimensions (see Tab. 4). SwinV2-b/12to16 (Liu et al., 2022a) is the best *supervised* model, particularly excelling in accuracy and object focus, while having quite a large number of parameters. DINOv2-B-reg (LP) (Darcet et al., 2024) exhibits a very good calibration and achieves good results in most other metrics. ViT-b/16-MAE (E2E) (He et al., 2022) is in no dimension particularly good or bad. Although ViT-b/16 (Dosovitskiy et al., 2021) and ResNet50 (He et al., 2016) are still two of the most popular backbones, they perform quite poorly, with QUBA ranks of 124 and 214, respectively. The ResNet50 has a comparably low number of parameters. CLIP-L/14 (Radford et al., 2021) suffers particularly in the calibration error and the number of parameters. On the other hand, it exhibits a high shape bias. Based on these findings, we suggest that the vision community should reconsider its selection of canonical backbone models.

> **Conclusion:** Widely used models such as ResNet50 and ViT underperform in several of the evaluated quality dimensions. This suggests that the vision community should critically reconsider its choice of canonical backbone models.

**Different weightings.** As outlined above, different practitioners could have different requirements on their models, depending on the task at hand. To reflect this in our analysis, in Fig. 5, we plot the five top-performing models when weighting one (group) of the considered quality dimensions twice as much as

Table 4: *QUBA score and quality dimensions for particularly popular models that did not make it in the top five.* The configuration lists the architecture, the training dataset, and the training paradigm.

| Model | Configuration | QUBA Score↑ /Rank↓ | Acc.↑ | Adv. Rob.↑ | C-Rob.↑ | OOD Rob.↑ | Cal. Error↓ | Class Balance↑ | Obj. Focus↑ | Shape Bias↑ | Params. in Mil.↓ |
|---|---|---|---|---|---|---|---|---|---|---|---|
| CLIP-L/14 (Radford et al., 2021) | ViL, WIT400m (Radford et al., 2021), self-SL | -0.65/243 | 0.76 | **0.32** | 0.76 | **1.04** | 0.0110 | **0.89** | 0.94 | **0.60** | 427 |
| ResNet50 (He et al., 2016) | CNN, IN1k, SL | -0.31/214 | 0.76 | 0.03 | 0.51 | 0.50 | 0.0021 | 0.75 | 0.93 | 0.22 | **25** |
| ViT-b/16 (Dosovitskiy et al., 2021) | Transformer, IN1k, SL | 0.20/124 | 0.81 | 0.18 | 0.66 | 0.56 | 0.0034 | 0.79 | 0.93 | 0.40 | 86 |
| ViT-b/16-MAE (He et al., 2022) | Transformer, IN1k, self-SL (E2E) | 0.36/84 | 0.84 | 0.25 | 0.71 | 0.58 | 0.0049 | 0.80 | 0.95 | 0.36 | 86 |
| DINOv2-B-reg (Darcet et al., 2024) | Transformer, LVD142m (Oquab et al., 2024), self-SL (LP) | 0.74/25 | 0.85 | 0.12 | 0.79 | 0.79 | **0.0011** | 0.80 | 0.94 | 0.49 | 90 |
| SwinV2-b/12to16 (Liu et al., 2022a) | Transformer, IN21k, SL | **0.90/8** | **0.86** | 0.26 | **0.81** | 0.81 | 0.0040 | 0.82 | **0.96** | 0.41 | 87 |

the other dimensions when computing the weighted mean for the QUBA score. EVA02-B/14 (IN21k) (Fang et al., 2023) leads in five setups, highlighting its versatility and quality beyond accuracy. The top five models remain fairly consistent when emphasizing accuracy, robustness, calibration, and shortcut learning, though their ranking within the top five varies slightly. Interestingly, the Hiera (Ryali et al., 2023) model family, self-supervised models only trained on ImageNet-1k, dominates strongly when focusing on class balance. Besides for class balance, the training dataset and architecture are quite heterogeneous for most of the setups. For the training paradigm, semi- and self-supervised learning dominate.

> **Conclusion:** When focusing on specific quality dimensions, the top five models remain fairly stable. For class balance, the Hiera model family is dominating.

**Limitations.** Naturally, our work comes with limitations. First, while we focus on image classification, which certainly is a relevant field, some downstream tasks rely on the evaluated backbone models for other purposes, and there is no guarantee that our findings will directly translate. Second, similar to the previous point, our analysis is limited to models trained on ImageNet-1k, and we cannot guarantee that the results generalize to other datasets. However, extending our analysis to another dataset is challenging: assuming an average training time of only 10 hours per model, retraining all 326 models on another dataset would require 3260 hours ($\sim$ 136 days) of compute, which is infeasible with our compute resources. Further, ImageNet-1k remains highly relevant, with numerous impactful papers focusing primarily on it and its variations. Third, we acknowledge that there are numerous different protocols to assess different dimensions of DNN quality. While we cannot include all protocols, we aimed for a *(i)* representative selection of *(ii)* established and *(iii)* easy-to-use (requiring no fine-tuning) protocols. However, our benchmark can easily be adapted or even extended with other protocols and quality dimensions. Fourth, to provide a comprehensive bird's-eye view, we prioritize breadth over depth here. While we briefly discuss various findings and why they might occur, this means we cannot provide detailed analyses for specific observations – indeed, studying theoretical connections for just two quality dimensions is often the scope of a full paper (*e.g.*, Minderer et al., 2021; Tsipras et al., 2019; Xiao et al., 2021). We rather consider this paper as groundwork, paving the way for future research to conduct more fine-grained, in-depth investigations. Fifth, our chosen evaluation protocols capture only certain aspects of the evaluated dimensions, and different protocols might yield different conclusions. That said, our benchmark design and its simple applicability allow us to conduct one of the largest studies to date. We thus argue that our design choices are justified and that our analysis makes numerous valuable contributions.

# 6 Conclusion

In this work, we provide a bird's-eye view of nine quality dimensions for ImageNet-1k image classification across 326 vision backbones by conducting one of the largest studies to date. This broad perspective allows us to examine how various training strategies and model architectures impact these dimensions, finding that larger training datasets and self-supervised pre-training followed by end-to-end fine-tuning enhance almost all

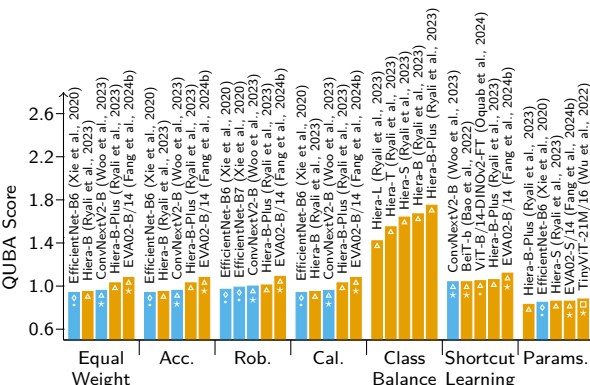

Figure 5: *Top five QUBA score models under different weightings.* We report the top five models when weighing specific (groups of) quality dimensions twice as strongly. See Fig. 3 for color and marker details.

measured quality dimensions. Additionally, we explore the relationships between these quality dimensions, providing novel insights such that object focus is strongly correlated with most of the considered dimensions. Our analysis is rounded off by ranking models based on our proposed QUBA score, which is possible due to our large model zoo. We highlight that no single model is universally superior and instead provide recommendations on which models excel for specific requirements. To conclude, we encourage researchers to consider a broad range of quality dimensions together, rather than focusing on individual ones, to foster the development of more well-behaved image classification models. Our work facilitates this by offering an easy-to-use benchmark, along with a comprehensive analysis of how design choices influence these quality dimensions, their interrelationships, and how models can be ranked across multiple quality dimensions.

## Acknowledgments

RH and SR have received funding from the European Research Council (ERC) under the European Union's Horizon 2020 research and innovation programme (grant agreement No. 866008). SSM has been funded by the Deutsche Forschungsgemeinschaft (DFG, German Research Foundation) – project No. 529680848. Further, SR was supported by the DFG under Germany's Excellence Strategy (EXC 3066/1 "The Adaptive Mind," project No. 533717223). Additionally, SR and SSM have received funding from the DFG under Germany's Excellence Strategy (EXC-3057/1 "Reasonable Artificial Intelligence", Project No. 533677015). Moreover, DB is supported by the Konrad Zuse School of Excellence in Learning and Intelligent Systems (ELIZA) through the DAAD programme Konrad Zuse Schools of Excellence in Artificial Intelligence, sponsored by the Federal Ministry of Education and Research.

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

"Toy Poodle"     Adv. attack     "Bucket"

Figure 6: *Illustration of an adversarial attack.*

## A   Details on the considered quality dimensions

In the following, we provide more details on the considered quality dimensions and the corresponding evaluation protocols. We let $f$ be the model of interest and use the ImageNet-1k evaluation split with $N$ images $\{x_n \mid n \in 1 \ldots N\}$ belonging to one of $C$ classes $\{c_n \in 1 \ldots C \mid n \in 1 \ldots N\}$ for most protocols.

**Accuracy.**   To measure the predictive performance of the considered models, we let $[\cdot]$ denote the Iverson bracket (Knuth, 1992) and report the ImageNet-1k top-1 accuracy

$$A = \frac{1}{N} \sum_{n=1}^{N} \big[ f(x_n) = c_n \big]. \tag{2}$$

**Adversarial robustness.**   To assess the adversarial robustness of a model, we use the two popular attacks, FGSM (Goodfellow et al., 2015) and PGD (Madry et al., 2018). In the Fast Gradient Sign Method (FGSM), adversarial examples are generated by computing the sign of the gradient of the cross-entropy loss $\mathcal{L}$ with respect to the original input, scaling it with a small factor $\epsilon$, and adding the result to the original image (see Fig. 6). Formally, we obtain the FGSM accuracy (FGSM-A) via

$$\text{FGSM-A} = \frac{1}{N} \sum_{n=1}^{N} \big[ f(\hat{x}_n) = c_n \big], \tag{3}$$

with $\hat{x}_n = x + \epsilon \cdot \text{sign}(\nabla_x \mathcal{L})$. Projected Gradient Descent (PGD) extends FGSM by applying it repeatedly, yielding

$$\text{PGD-A} = \frac{1}{N} \sum_{n=1}^{N} \Big[ f(\hat{x}_n^{(I)}) = c_n \Big], \tag{4}$$

with $\hat{x}_n^{(i+1)} = \hat{x}_n^{(i)} + \epsilon \cdot \text{sign}\big(\nabla_{\hat{x}_n^{(i)}} \mathcal{L}\big)$ and $\hat{x}^{(0)} = x$. We use $\epsilon = {}^8\!/_{255}$ and $I = 10$ (Kim, 2020). To reduce the dependence on the clean accuracy of the model, we report adversarial robustness relative to the accuracy A from Eq. (2). We combine the results from the two attacks using their geometric mean (GM), resulting in the final adversarial robustness

$$\text{AR} = \text{GM} \left( \frac{\text{FGSM-A}}{\text{A}}, \frac{\text{PGD-A}}{\text{A}} \right). \tag{5}$$

**Corruption robustness.**   To assess a model's robustness to common corruptions (CR) like JPEG compression or contrast changes (see Fig. 7), we measure the accuracy on ImageNet-C (Hendrycks & Dietterich, 2019), *i.e.*, the ImageNet evaluation split with different corruption types of increasing strength. We here follow Hendrycks & Dietterich (2019) and use the standard mean instead of the geometric mean to summarize the results for different corruption types and strengths. To normalize the C-robustness and to be consistent with our other robustness metrics, we deviate from Hendrycks & Dietterich (2019) and again report the top-1 accuracy on the corrupted data ($A_{\text{Corr}}$) relative to the clean ImageNet-1k accuracy (A), yielding

$$\text{CR} = \frac{A_{\text{Corr}}}{\text{A}}. \tag{6}$$

**OOD robustness.**   To measure the out-of-domain robustness of a model, we report the geometric mean of the relative accuracy (normalized by A) on five out-of-domain datasets. Specifically, we use ImageNet-R (Hendrycks et al., 2021), ImageNet-Sketch (Wang et al., 2019), as well as Stylized-ImageNet, Edge, and

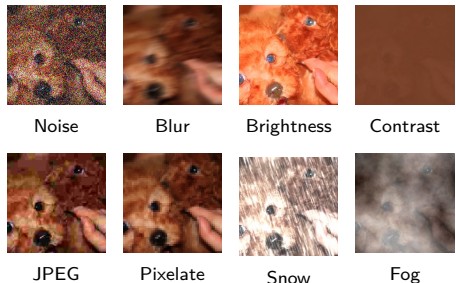

Figure 7: *Example corruptions from ImageNet-C (Hendrycks & Dieterrich, 2019).*

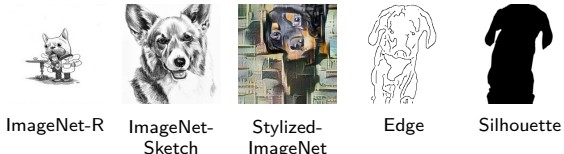

Figure 8: *Example images from the considered OOD datasets (Geirhos et al., 2019; Hendrycks et al., 2021; Wang et al., 2019).*

Silhouette from Geirhos et al. (2019). For ImageNet-Sketch and Stylized-ImageNet, we use the versions also used in Geirhos et al. (2021). Please refer to Fig. 8 for example images of the different datasets.

**Calibration error.** Calibration means that the output confidence of a model faithfully reflects the probability of the prediction being correct. We use two established metrics for measuring the calibration error (CE). The expected calibration error (ECE) (Guo et al., 2017; Nixon et al., 2019) divides the predictions into $B$ bins $b$ based on the output confidence of the model and compares how well the confidences $\text{conf}(b)$ of the predictions in that bin are aligned with the accuracy $\text{A}_b$ of the predictions in that bin:

$$\text{ECE} = \sum_{b=1}^{B} \frac{n_b}{N} \left| \text{A}_b - \text{conf}(b) \right| , \tag{7}$$

with $n_b$ denoting the number of predictions in bin $b$. Since a common criticism of the ECE is the use of a fixed bin range, we additionally report the adaptive calibration error (ACE) (Nixon et al., 2019) that measures the discrepancy between $\text{A}_{r,c}$ and $\text{conf}(r,c)$, *i.e.*, the accuracy and confidence of images in the adaptive calibration range $r$ for class label $c$:

$$\text{ACE} = \frac{1}{CR} \sum_{c=1}^{C} \sum_{r=1}^{R} \left| \text{A}_{r,c} - \text{conf}(r,c) \right| . \tag{8}$$

As in Guo et al. (2017); Nixon et al. (2019), we use 15 bins for both protocols and again report the geometric mean (GM) of both errors, *i.e.*,

$$\text{CE} = \text{GM}(\text{ECE}, \text{ACE}). \tag{9}$$

**Class balance.** We consider a model fair if none of the classes is classified less well than the others (Benz et al., 2020). We evaluate the class balance of accuracies ($\text{F}_{\text{Acc}}$) of a model similar to Croce et al. (2021) and subtract the standard deviation of ImageNet-1k class accuracies from 1 (this ensures that higher scores indicate a higher class balance; the standard deviation cannot exceed 1):

$$\text{F}_{\text{Acc}} = 1 - \sqrt{\frac{1}{C} \sum_{c=1}^{C} (A_c - A)^2} , \tag{10}$$

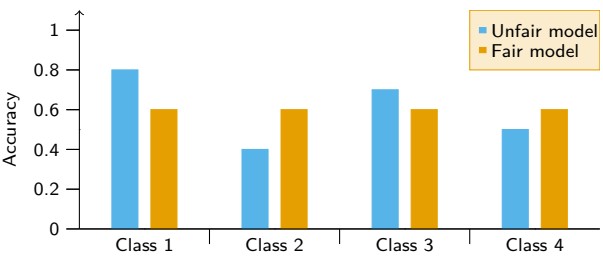

Figure 9: *Visual illustration of the class accuracies of a fair model* vs. *an unfair one, both with equal average accuracy.*

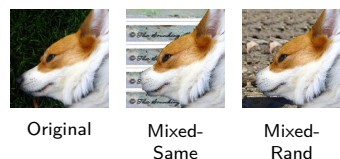

Figure 10: *Example images from Xiao et al. (2021) to estimate the object focus.*

with $A_c$ denoting the accuracy for images of class $c$. Intuitively, a high value indicates that the accuracies of each class are similar, and thus, the model behaves fairly, as illustrated in Fig. 9. Similar to Kuzucu et al. (2024), we also consider a model fair if the average confidence (target softmax outputs) for each class is balanced. We compute the class balance of confidences ($F_{Conf}$) of a model by subtracting the standard deviation of ImageNet-1k average class confidences from 1:

$$F_{Conf} = 1 - \sqrt{\frac{1}{C}\sum_{c=1}^{C}(Conf_c - Conf)^2}\,, \tag{11}$$

with $Conf_c$ denoting the average confidence for images of class $c$ and $Conf$ denoting the average confidence for all images. The final class balance score (F) is the geometric mean (GM) of $F_{Acc}$ and $F_{Conf}$, *i.e.*,

$$F = GM(F_{Acc}, F_{Conf}). \tag{12}$$

**Object focus.**     To compute the object focus (OF), we first compute the background focus BF $=$ $A_{\text{Mixed-Same}} - A_{\text{Mixed-Rand}}$ (Xiao et al., 2021). Mixed-Rand is a dataset where image backgrounds are substituted with backgrounds from random classes and, therefore, contain no class information (see Fig. 10). Mixed-Same is a dataset where image backgrounds are substituted with backgrounds from the same class to account for editing artifacts in Mixed-Same. Intuitively, we measure the drop in accuracy when changing the image background with the background from another class to assess if the model focuses on background signals. Next, we compute the inverse of the background focus to obtain the object focus OF $= 1 - $ BF.

**Shape bias.**     Geirhos et al. (2019) showed that ImageNet-trained CNNs exhibit a strong texture bias, meaning that decisions are formed on the basis of texture information rather than shape information. As a stronger shape bias is said to be advantageous for robustness and more in line with how humans form decisions, we follow Geirhos et al. (2019) and report the shape bias (SB) as follows:

$$SB = \frac{\sum_{n=1}^{N}\left[f(\tilde{x}_n) = c_n^{(shape)}\right]}{\sum_{m=1}^{N}\left(\left[f(\tilde{x}_m) = c_m^{(shape)}\right] + \left[f(\tilde{x}_m) = c_m^{(texture)}\right]\right)}\,, \tag{13}$$

with $\tilde{x}_n$ being synthetically generated images with a texture-shape cue conflict, *i.e.*, where the shape is from one class $c_n^{(shape)}$ and the texture is from another class $c_n^{(texture)}$, *e.g.*, as in Fig. 11.

**Parameters.**     As memory efficiency and inference time depend highly on the implementation and hardware used, impeding future comparisons, we report the number of parameters as a proxy.

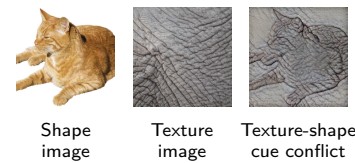

Figure 11: *Example images from Geirhos et al. (2019) to estimate the shape bias.*

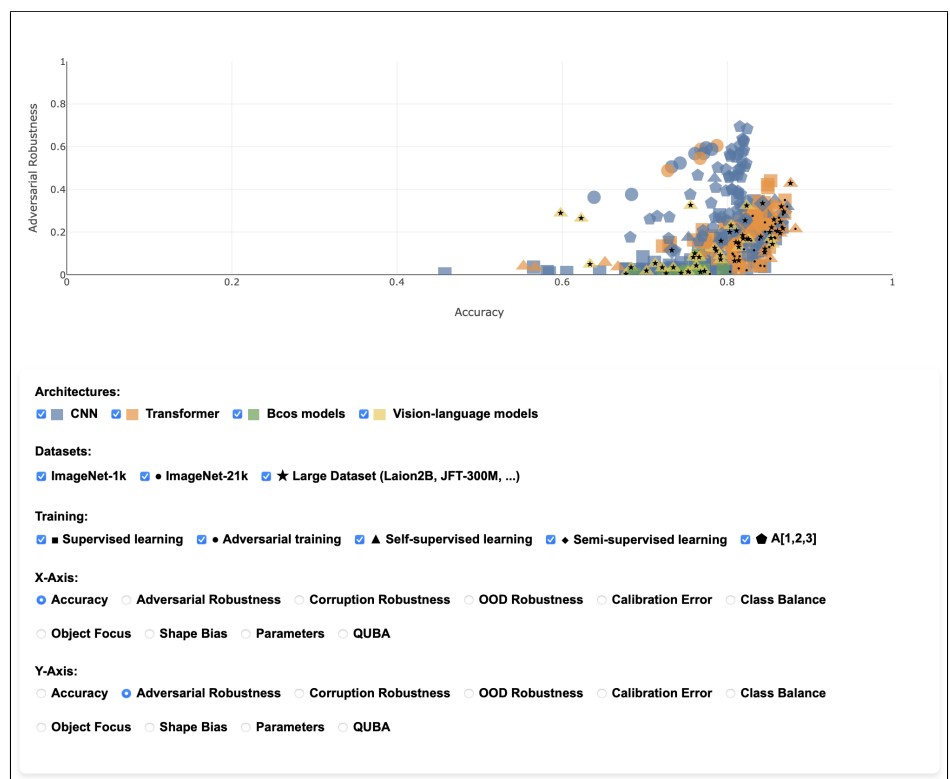

Figure 12: *Preview of our interactive plot.* It can be found in the supplement under `interactive_plot.html` or on our project page.

## B    Interactive scatter plot

In Fig. 3 of the main paper, we only plot a representative subset of models to reduce clutter. In the supplement, we additionally include an interactive plot, called `interactive_plot.html`, that can be opened with standard browsers. It includes all 326 models and allows the filtering of the models based on the training dataset, training paradigm, and architecture. Also, different quality dimensions can be chosen for the $x$ and $y$-axis to visualize different relationships. Hovering the cursor over a marker reveals a tooltip displaying the model's name and scores for the considered quality dimensions, offering detailed performance insights at a glance. A preview of the interactive plot is included in Fig. 12.

## C    Additional experiments

### C.1    Relationships in model subgroups

During our analysis, we noticed that the rank correlation matrices for certain subgroups of the models can change compared to the correlation matrix of all models shown in Fig. 4 of the main paper. This has two important implications: First, it is crucial to look at as many models as possible to make general statements,

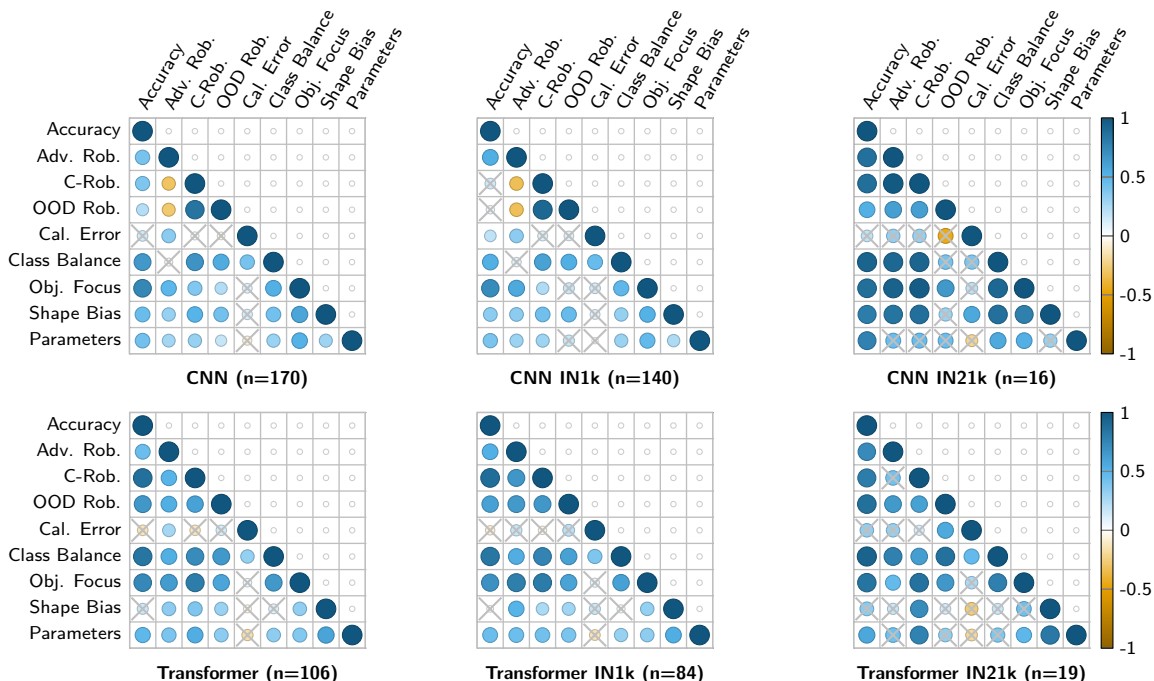

Figure 13: *Rank correlation matrices for model subgroups.* We investigate the rank correlations of different quality dimensions for all CNNs *(top left)*, all CNNs trained on ImageNet-1k *(top middle)*, all CNNs trained on ImageNet-21k *(top right)*, all Transformers *(bottom left)*, all Transformers trained on ImageNet-1k *(bottom middle)*, and all Transformers trained on ImageNet-21k *(bottom right)*. Crossed-out entries indicate a *p*-value above 0.05, and thus, are not statistically significant.

which shows that our confirmation of (or contradiction to) existing results with a large model zoo is a valuable contribution. Second, general statements should be taken cautiously, as they do not necessarily apply to all model groups. We thus continue our analysis by studying the correlation matrices for different model subgroups. In Fig. 13, we plot the correlation matrices for all CNNs and Transformers, respectively. Overall, both matrices are similar to the matrix for all models (Fig. 4 of the main paper). However, for CNNs, C-robustness and OOD robustness are negatively correlated with adversarial robustness, while the opposite holds for Transformers. We observe that the calibration error is positively correlated with accuracy for CNNs and negatively for Transformers (however, both correlations are not statistically significant). This also aligns with another large-scale study that found that the calibration error is decreasing with more accuracy for recent state-of-the-art Transformers (Minderer et al., 2021). Notably, accuracy and shape bias are strongly correlated for CNN-based models, while they exhibit only a weak correlation for Transformer models. Moreover, for Transformers, an increased number of parameters is quite strongly correlated with desirable properties of all other quality dimensions, which is less pronounced for CNNs.

Given that a larger fraction of the newer Transformer models were trained on ImageNet-21k, whereas most CNNs were trained on ImageNet-1k, some of the correlations could also be due to the training dataset size. To account for this, in Fig. 13, we further plot the correlation matrices for CNNs and Transformers trained exclusively on ImageNet-1k and ImageNet-21k, respectively. Interestingly, the negative correlation between adversarial robustness and C-robustness/OOD robustness for CNNs is only apparent when trained on ImageNet-1k but not when trained on ImageNet-21k. Calibration error and OOD robustness have a strong negative correlation for ImageNet-21k CNNs while having almost no correlation for ImageNet-1k CNNs (however, not statistically significant). Generally, the correlations for CNNs are much more pronounced for the models trained on the larger dataset. For Transformers, the statistically significant correlations are quite similar for models trained on ImageNet-1k and ImageNet-21k.

Table 5: *Top 15 models with the highest OOD accuracy.* Vision-language (ViL) models are clearly dominating the list.

| Model | Configuration | OOD Acc. ↑ |
|---|---|---|
| SigLIP2-l/16  (Tschannen et al., 2025) | ViL, WebLI (Chen et al., 2023), self-SL (E2E) | 0.87 |
| MetaCLIP-L/14  (Xu et al., 2024) | ViL, MetaCLIP-400M (Xu et al., 2024), self-SL | 0.85 |
| MobileCLIP-B (LT)  (Vasu et al., 2024) | ViL, DataCompDR-1B (Vasu et al., 2024), self-SL | 0.85 |
| CLIP-L/14-CommPool XL-DFN2B  (Fang et al., 2024a) | ViL, DFN2B (Fang et al., 2024a), self-SL (E2E) | 0.84 |
| SigLIP-l/16  (Zhai et al., 2023) | ViL, WebLI (Chen et al., 2023), self-SL | 0.83 |
| CLIP-L/14-DataCompXL  (Gadre et al., 2023) | ViL, DataCompXL (Gadre et al., 2023), self-SL (E2E) | 0.83 |
| MobileCLIP-B  (Vasu et al., 2024) | ViL, DataCompDR-1B (Vasu et al., 2024), self-SL | 0.83 |
| SigLIP2-b/16  (Tschannen et al., 2025) | ViL, WebLI (Chen et al., 2023), self-SL (E2E) | 0.82 |
| CLIP-ConvNeXt-L  (Schuhmann et al., 2022; Wortsman et al., 2022) | ViL, Laion2B (Schuhmann et al., 2022), self-SL | 0.81 |
| CLIP-L/14-Laion2B  (Schuhmann et al., 2022; Wortsman et al., 2022) | ViL, Laion2B (Schuhmann et al., 2022), self-SL (E2E) | 0.80 |
| SigLIP-b/16  (Zhai et al., 2023) | ViL, WebLI (Chen et al., 2023), self-SL | 0.80 |
| CLIP-ConvNeXt-L-320px  (Schuhmann et al., 2022; Wortsman et al., 2022) | ViL, Laion2B (Schuhmann et al., 2022), self-SL | 0.80 |
| CLIP-B/16-DataCompXL  (Gadre et al., 2023) | ViL, DataCompXL (Gadre et al., 2023), self-SL (E2E) | 0.79 |
| ViT-l/14-DINOv2-reg-LP  (Darcet et al., 2024) | Transformer, LVD142m (Oquab et al., 2024), self-SL (E2E) | 0.79 |
| MetaCLIP-B/16  (Xu et al., 2024) | ViL, MetaCLIP-400M (Xu et al., 2024), self-SL | 0.78 |

## C.2   OOD robustness for models trained on large-scale datasets

In Sec. 3 of the main paper, in the paragraph about vision-language (ViL) models, we state that vision-language models outperform other models trained on similarly large datasets when it comes to out-of-domain robustness. In Tab. 5, we support this statement with numerical results. To this end, we report the "raw" OOD accuracy, *i.e.*, the OOD accuracy without normalizing by the clean accuracy, for the 15 models with the highest OOD accuracy. The best ViL model achieves an OOD accuracy of 0.87 while the best self-supervised model (pre-)trained on a large-scale dataset only achieves an OOD accuracy of 0.79. Further, vision-language models clearly dominate in the list (14 out of 15). These results suggest that the increased robustness of ViL models is not only due to the increased dataset size but also due to other factors that are likely linked to the language part of the models.

## C.3   Reproducing conflicting results

We report several results that extend findings from related work using smaller model pools. To verify these results, we reproduce their experiments with similar model pools in Tab. 6. When using comparable models, we successfully replicate most of their findings, indicating that the discrepancies arise from the limited model pools in related work. This underscores the importance of our large model zoo.

## C.4   Comparison between our adversarial robustness protocol and AutoAttack

In the main paper, we employ a combination of FGSM and PGD adversarial attacks to assess adversarial robustness. However, within the community, alternative evaluation standards have emerged, such as *AutoAttack* – a widely adopted adversarial benchmark introduced in *RobustBench* (Croce et al., 2021). It consists of an ensemble of four parameter-free attacks: two PGD variants using cross-entropy and difference-of-logits ratio losses, a targeted FAB attack (Croce & Hein, 2020), and a black-box Square attack (Andriushchenko et al., 2020).

We opted for a more straightforward setup, as we found the success rate of *AutoAttack* to be excessively high for our purposes – nearly all non-adversarially robust models are reduced below 0.1 accuracy, effectively collapsing. Moreover, prior work has criticized *AutoAttack* for producing perturbations that significantly alter the input images, making adversarial examples easily detectable, and for exhibiting sensitivity to image resolution (Lorenz et al., 2021).

Nevertheless, given that *AutoAttack* remains an important benchmark in adversarial robustness research, we include a comparison between the results obtained using *AutoAttack* and those produced by our proposed evaluation protocol in this section. First, we evaluate how different training paradigms and architectures (*cf*. Sec. 3) affect adversarial robustness under both our protocol and *AutoAttack*, as shown in Tab. 7. For nearly all configurations, the overall conclusions remain consistent. Only in setups (e) and (f) do we observe improvements in adversarial robustness when using our protocol, while robustness under *AutoAttack* remains unchanged (though these differences are not statistically significant). Next, we compare the correlation

Table 6: *Comparison of our results that contradict/extend findings from related work.* In the main paper, we highlight a few findings that diverge from results in related work. To verify these discrepancies, we reproduce their findings using a similar model pool. In doing so, we find that most of their conclusions hold when considering similar models.

| Finding from related work on their respective models | Used models in related work | Reproduction with similar models | Our finding | Used models |
|---|---|---|---|---|
| Adversarial Training improves calibration error (Grabinski et al., 2022) | ResNet18, ResNet50 (He et al., 2016), WRN-50-2 (Zagoruyko & Komodakis, 2016) *vs.* ResNet18, ResNet50, WRN-50-2 from Salman et al. (2020) NOTE: Grabinski et al. (2022) conducted additional experiments with non-ImageNet models | **Average ECE** Standard Models: 0.2722 Robust Models: 0.0876 **Conclusion**: We can reproduce their finding when using a similar model pool | Adversarial training impairs calibration when considering a broad and diverse set of models | See Tab. 9 (c) |
| Longer training worsens ECE (Minderer et al., 2021) | Five self-trained versions of BiT-L-R50x1 and BiT-L-R101x3 (Minderer et al., 2021) NOTE: Since the used or similar checkpoints are not publicly available, we use the A[1,2] (Wightman et al., 2021) versions of ResNet18, ResNet34, ResNet50, ResNet101 and ResNet152, which are at least somewhat similar to BiT (Kolesnikov et al., 2020), for reproduction | **Average ECE** Standard models: 0.0396 A[1] models: 0.0845 A[2] models: 0.1025 **Conclusion**: We can reproduce their finding when using a similar model pool | Longer training improves calibration error when considering a broad and diverse set of models | See Tab. 9 (g) |
| Accuracy is positively correlated with calibration error (Guo et al., 2017) | DenseNet161 (Huang et al., 2017), ResNet152 (He et al., 2016) NOTE: Guo et al. (2017) conducted additional experiments with non-ImageNet models | **Accuracy & ECE** ResNet152 (He et al., 2016): 0.7832 & 0.05 DenseNet161 (Huang et al., 2017): 0.7711 & 0.06 **Conclusion**: We can *not* reproduce their finding when using a similar model pool | We found that accuracy is negatively correlated with calibration error ($p < 0.05$) when considering a broad and diverse set of models | Entire model zoo |
| Adversarial robustness is positively correlated with C-robustness and OOD robustness for a given architecture (Liu et al., 2025) | VGG13 (Simonyan & Zisserman, 2015), VGG16 (Simonyan & Zisserman, 2015), VGG19 (Simonyan & Zisserman, 2015), XciT-S (Ali et al., 2021), XciT-M (Ali et al., 2021), XciT-L (Ali et al., 2021), ResNet50 (He et al., 2016), ResNet101 (He et al., 2016), ResNet152 (He et al., 2016), Wide-ResNet50 (Salman et al., 2020), DenseNet121 (Huang et al., 2017), DenseNet161 (Huang et al., 2017), DenseNet201 (Huang et al., 2017), ConvNeXT-S (Liu et al., 2022b), ConvNeXT-S (21k) (Liu et al., 2022b), ConvNeXT-B (Liu et al., 2022b), ConvNeXT-B (21k) (Liu et al., 2022b), ConvNeXT-L (Liu et al., 2022b), ConvNeXT-L (21k) (Liu et al., 2022b), ViT-s/16 (Dosovitskiy et al., 2021), ViT-s/16 (21k) (Dosovitskiy et al., 2021), ViT-b/16 (Dosovitskiy et al., 2021), ViT-b/16 (21k) (Dosovitskiy et al., 2021), ViT-b/16 (MAE) (He et al., 2022), ViT-l/16 (Dosovitskiy et al., 2021), ViT-l/16 (21k) (Dosovitskiy et al., 2021), ViT-l (MAE) (He et al., 2022), Swin-S (Liu et al., 2021), Swin-b (Liu et al., 2021), Swin-b (21k) (Liu et al., 2021), ResNet50 (Mocov3) (Chen et al., 2021b), T2T-14 (Yuan et al., 2021), T2T-19 (Yuan et al., 2021), T2T-24 (Yuan et al., 2021), Swin-L (Liu et al., 2021) | Correlation between adversarial robustness and c-robustness: 0.6547 (p = 0.0). Correlation between adversarial robustness and OOD robustness: 0.6942 (p=0.0) **Conclusion**: We can reproduce their finding when using a similar model pool | We found no significant correlations between adversarial robustness and c-robustness (p=0.84) or OOD robustness (p=0.15) when considering a broad and diverse set of models | Entire model zoo |

matrices among our full model zoo (*cf*. Sec. 4) with the two different adversarial robustness protocols in Fig. 14. Again, all the statistically significant conclusions remain the same. Lastly, we compare how model rankings change (*cf*. Sec. 5) under the two different evaluation protocols for adversarial robustness. While the rank correlation between the two adversarial robustness metrics remains fairly high (0.64), the overall ordering of models changes substantially. This shift occurs because, under *AutoAttack*, nearly the entire model zoo collapses – reducing the mean/standard deviation of adversarial robustness from 0.2/0.11 to 0.04/0.03. Consequently, the adversarially trained models that still achieve relatively high accuracies of around 0.6–0.7 deviate by many more standard deviations from the mean. As a result, the adversarial robustness score dominates the overall ranking, with only adversarially robust models appearing among the top five (see Table 8). Although this effect could easily be mitigated by reweighting our proposed QUBA score, as discussed in Sec. 5, we consider this finding an indication that *AutoAttack* is not the most suitable evaluation protocol for adversarial robustness within our benchmark. Nevertheless, aside from the unweighted rankings – which must be weighted according to user requirements in any case – almost none of the conclusions in our work would differ under *AutoAttack*.

# D  Experimental details

The code provided in the supplemental material gives detailed instructions on how to use our proposed benchmark and how to reproduce our main results. To include new models, one only needs to add the respective model file and weights.

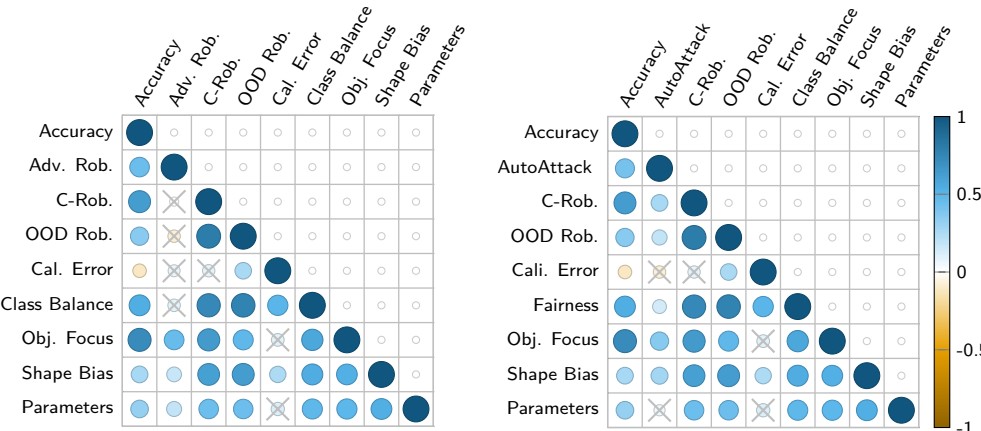

Figure 14: *Rank correlation matrix for the considered quality dimensions among our full model zoo, comparing our adversarial robustness protocol to* AutoAttack. We investigate differences of the rank correlations among all our models with our adversarial robustness metric *(left)* and AutoAttack *(right)*. Crossed-out entries indicate a *p*-value above 0.05, and thus, are not statistically significant. All the statistically significant conclusions remain unchanged.

Table 7: *Average adversarial robustness and AutoAttack for models with different configurations.* We evaluate various setups, focusing on different training strategies (a–g) and different architectural choices (h–j). In each configuration, we report the average score for both metrics across the models associated with that configuration. As different models are available for different setups, each setup considers a distinct selection of models being compared. As a result, both the models and their total number (indicated by the number beside each setup) vary across setups to maintain a fair basis for comparison. The number of asterisks represents the statistical significance of differences in the average scores of a quality dimension across configurations within each setup: *** for $p < 0.05$, ** for $p < 0.1$, and * for $p < 0.2$, based on $t$-test results.

| Setup | | Configuration | Adv. Rob. ↑ | Auto- Attack ↑ |
|---|---|---|---|---|
| (a) | 14 | CNNs (IN-1k) | 0.14 | 0.03 |
| | | CNNs (IN-21k) | **0.15** | **0.04** |
| (b) | 16 | Transformers (IN-1k) | **0.21** | **0.09** |
| | | Transformers (IN-21k) | 0.17* | 0.05** |
| (c) | 11 | Supervised models | 0.12 | 0.04 |
| | | Adversarially trained models | **0.52**\*** | **0.68**\*** |
| (d) | 13 | Supervised models | **0.20** | **0.08** |
| | | Self-supervised models (LP) | 0.10*** | 0.06 |
| (e) | 25 | Supervised models | 0.16 | 0.07 |
| | | Self-supervised models (E2E) | **0.24**\*** | 0.07 |
| (f) | 13 | Supervised models | 0.13 | 0.04 |
| | | Semi-supervised models | **0.20**\*** | 0.04 |
| (g) | 19 | Supervised models | 0.12 | 0.03 |
| | | A1 supervised models (600 epochs) | **0.47**\*** | **0.05**\*** |
| | | A2 supervised models (300 epochs) | 0.41*** | **0.05**\*** |
| | | A3 supervised models (100 epochs) | 0.32*** | **0.05**\*** |
| (h) | 46 | CNNs | 0.11 | 0.11 |
| | | Transformers | **0.20** | **0.20** |
| (i) | 12 | Standard models | **0.07** | **0.03** |
| | | B-cos models | 0.02*** | 0.00*** |
| (j) | 24 | Standard models | **0.18** | **0.09** |
| | | Vision-language models | 0.10* | 0.01*** |

To ensure comprehensive comparisons in future work, we publish the code and results of our model zoo (`https://visinf.github.io/beyond-accuracy`). This will allow practitioners to evaluate and compare the considered quality dimensions easily for their model. In the next subsections, we will describe additional experimental details.

Table 8: *QUBA score and quality dimensions for the top five performing models using AutoAttack as metric for adversarial robustness.* The configuration lists the architecture, training dataset, and training paradigm. † indicates models trained with knowledge distillation.

| Model | Configuration | QUBA Score↑ | Acc.↑ | Adv. Rob.↑ | C-Rob.↑ | OOD Rob.↑ | Cal. Error↓ | Class Balance↑ | Obj. Focus↑ | Shape Bias↑ | Params. in Mil.↓ |
|---|---|---|---|---|---|---|---|---|---|---|---|
| ViT-S (Singh et al., 2023) | Transformer, IN1k, AT | 1.27 | 0.73 | 0.66 | 0.63 | 0.72 | 0.0081 | 0.78 | 0.94 | 0.72 | **22** |
| Swin-B (Liu et al., 2025) | Transformer, IN1k, AT | 1.39 | 0.77 | **0.74** | 0.65 | 0.74 | 0.0083 | 0.79 | 0.94 | 0.73 | 87 |
| ViT-B Singh et al. (2023) | Transformer, IN1k, AT | 1.44 | 0.77 | 0.71 | **0.69** | 0.50 | 0.0072 | 0.79 | **0.95** | **0.77** | 87 |
| ConvNeXt-B Liu et al. (2025) | CNN, IN1k, AT | 1.45 | **0.77** | 0.73 | 0.64 | 0.62 | 0.0075 | **0.79** | 0.95 | 0.73 | 88 |
| ConNeXt-B Singh et al. (2023) | CNN, IN1k, AT | **1.45** | 0.76 | 0.72 | 0.64 | **0.80** | **0.0069** | 0.78 | 0.95 | 0.74 | 88 |

## D.1 Comparisons

In Tab. 2 of the main paper, we compare the average of each quality dimension for different setups. We report the models that have been used for each comparison in Tab. 9.

Table 9: *Models used for each comparison in Tab. 2 of the main paper.* In some setups, the same model appears multiple times within a configuration because the corresponding configuration contains multiple models that need to be compared to that single model (*e.g.*, there might be two different ViL models using the same backbone). This duplication ensures that the final average accounts for these cases correctly.

| Setup | # of Models | Configuration | Models |
|---|---|---|---|
| (a) | 14 | CNNs (IN-1k) | MobileNetV3-l (Howard et al., 2019), ResNet50 (He et al., 2016), ResNet101 (He et al., 2016), EfficientNet-v2-S (Tan & Le, 2021), EfficientNet-v2-M (Tan & Le, 2021), EfficientNet-v2-L (Tan & Le, 2021), ConvNeXt-T (Liu et al., 2022b), ConvNeXt-S (Liu et al., 2022b), ConvNeXt-B (Liu et al., 2022b), ConvNeXt-L (Liu et al., 2022b), ConvNeXtV2-N (Woo et al., 2023), ConvNeXtV2-T (Woo et al., 2023), ConvNeXtV2-B (Woo et al., 2023), ConvNeXtV2-L (Woo et al., 2023) |
| | | CNNs (IN-21k) | MobileNetV3-l (Howard et al., 2019), BiTM-ResNet50x1 (Kolesnikov et al., 2020), BiTM-ResNet101x1 (Kolesnikov et al., 2020), EfficientNet-v2-S (Tan & Le, 2021), EfficientNet-v2-M (Tan & Le, 2021), EfficientNet-v2-L (Tan & Le, 2021), ConvNeXt-T (Liu et al., 2022b), ConvNeXt-S (Liu et al., 2022b), ConvNeXt-L (Liu et al., 2022b), ConvNeXtV2-N (Woo et al., 2023), ConvNeXtV2-T (Woo et al., 2023), ConvNeXtV2-B (Woo et al., 2023), ConvNeXtV2-L (Woo et al., 2023) |
| (b) | 16 | Transformers (IN-1k) | DeiT3-s (Touvron et al., 2022), DeiT3-m (Touvron et al., 2022), DeiT3-b (Touvron et al., 2022), DeiT3-l (Touvron et al., 2022), SwinV2-b/16 (Liu et al., 2022a), TinyViT-5M/16 (Wu et al., 2022), TinyViT-11M/16 (Wu et al., 2022), TinyViT-21M/16 (Wu et al., 2022), DeiT-s (Touvron et al., 2021a), ViT-b/16 (Dosovitskiy et al., 2021), ViT-b/32 (Dosovitskiy et al., 2021), ViT-l/16 (Dosovitskiy et al., 2021), ViT-b/32 (Dosovitskiy et al., 2021), ViT-b/16 (Dosovitskiy et al., 2021), ViT-b/16 (Dosovitskiy et al., 2021), DeiT-s (Touvron et al., 2021a) |
| | | Transformers (IN-21k) | DeiT3-s (Touvron et al., 2022), DeiT3-m (Touvron et al., 2022), DeiT3-b (Touvron et al., 2022), DeiT3-l (Touvron et al., 2022), SwinV2-b/12to16 (Liu et al., 2022a), TinyViT-5M/16 (Wu et al., 2022), TinyViT-11M/16 (Wu et al., 2022), TinyViT-21M/16 (Wu et al., 2022), ViT-s/16 (Steiner et al., 2022), ViT-l/16 (Steiner et al., 2022), ViT-b/32 (Steiner et al., 2022), ViT-l/16 (Steiner et al., 2022), BeiT-b (Bao et al., 2022), EVA02-B/14 (Fang et al., 2024b), EVA02-S/14 (Fang et al., 2024b) |
| (c) | 11 | Supervised models | WRN-50-2 (Zagoruyko & Komodakis, 2016), ResNet50 (He et al., 2016), Swin-b (Liu et al., 2021), ConvNeXt-B (Liu et al., 2022b), ConvNeXt-L (Liu et al., 2022b), ConvNeXt-T (Liu et al., 2022b), ConvNeXt-S (Liu et al., 2022b), ConvNeXt-B (Liu et al., 2022b), ConvNeXt-L (Liu et al., 2022b), ViT-b/16 (Dosovitskiy et al., 2021), DeiT-s (Touvron et al., 2021a) |
| | | Adversarially trained models | WRN-50-2 (Salman et al., 2020), ResNet50 (Salman et al., 2020), Swin-b (Liu et al., 2025), ConvNeXt-B (Liu et al., 2025), ConvNeXt-L (Liu et al., 2025), ConvNeXt-T (Singh et al., 2023), ConvNeXt-S (Singh et al., 2023), ConNeXt-B (Singh et al., 2023), ConvNeXt-L (Singh et al., 2023), ViT-b/16 (Singh et al., 2023), ViT-s/16 (Singh et al., 2023) |
| (d) | 13 | Supervised models | MViTv2-b (Li et al., 2022a), ViT-b/16 (Dosovitskiy et al., 2021), ResNet50 (He et al., 2016), ViT-b/16 (Dosovitskiy et al., 2021), ViT-l/16 (Dosovitskiy et al., 2021), ViT-b/16 (Dosovitskiy et al., 2021), DeiT-s (Touvron et al., 2021a), DeiT-s (Touvron et al., 2021a), Hiera-T (Ryali et al., 2023), Hiera-S (Ryali et al., 2023), ViT-l/16 (Dosovitskiy et al., 2021), ViT-b/16 (Dosovitskiy et al., 2021), DeiT-s (Touvron et al., 2021a) |
| | | Self-supervised models (LP) | Hiera-B (Ryali et al., 2023), ViT-b/16-MAE (He et al., 2022), ResNet50-DINO (Caron et al., 2021), ViT-b/16-DINO (Caron et al., 2021), ViT-l/14-DINOv2-LP (Oquab et al., 2024), ViT-b/14-DINOv2-LP (Oquab et al., 2024), ViT-s/14-DINOv2-LP (Oquab et al., 2024), ViT-s/16-DINO (Caron et al., 2021), Hiera-T (Ryali et al., 2023), ViT-l/14-DINOv2-reg-LP (Darcet et al., 2024), ViT-b/14-DINOv2-reg-LP (Darcet et al., 2024), ViT-s/14-DINOv2-reg-LP (Darcet et al., 2024) |
| (e) | 25 | Supervised models | MViTv2-b (Li et al., 2022a), ViT-b/16 (Dosovitskiy et al., 2021), ResNet50 (He et al., 2016), ViT-b/16 (Dosovitskiy et al., 2021), ConvNeXt-T (Liu et al., 2022b), ConvNeXt-B (Liu et al., 2022b), ConvNeXt-L (Liu et al., 2022b), ConvNeXt-T (Liu et al., 2022b), ConvNeXt-B (Liu et al., 2022b), ConvNeXt-L (Liu et al., 2022b), ViT-b/16 (Steiner et al., 2022), ViT-b/16 (Steiner et al., 2022), ViT-s/16 (Steiner et al., 2022), ViT-t/16 (Steiner et al., 2022), ViT-l/16 (Dosovitskiy et al., 2021), ViT-b/16 (Dosovitskiy et al., 2021), DeiT-s (Touvron et al., 2021a), ViT-l/16 (Dosovitskiy et al., 2021), ViT-b/16 (Dosovitskiy et al., 2021), DeiT-s (Touvron et al., 2021a), ViT-b/16 (Dosovitskiy et al., 2021), ViT-b/16 (Dosovitskiy et al., 2021), ViT-b/32 (Dosovitskiy et al., 2021), ViT-b/32 (Dosovitskiy et al., 2021) |
| | | Self-supervised models (E2E) | Hiera-B (Ryali et al., 2023), ViT-b/16-MAE (He et al., 2022), ResNet50-DINO (Caron et al., 2021), ViT-b/16-DINO (Caron et al., 2021), ConvNeXtV2-T (Woo et al., 2023), ConvNeXtV2-B (Woo et al., 2023), ConvNeXtV2-T (Woo et al., 2023), ConvNeXtV2-B (Woo et al., 2023), ConvNeXtV2-L (Woo et al., 2023), BeiT-b (Bao et al., 2022), BeiTV2-b (Peng et al., 2022), EVA02-S/14 (Fang et al., 2024b), EVA02-B/14 (Fang et al., 2024b), EVA02-T/14 (Fang et al., 2024b), ViT-l/14-DINOv2-FT (Oquab et al., 2024), ViT-b/14-DINOv2-FT (Oquab et al., 2024), ViT-s/14-DINOv2-FT (Oquab et al., 2024), ViT-l/14-DINOv2-reg-LP (Darcet et al., 2024), ViT-b/14-DINOv2-reg-LP (Darcet et al., 2024), ViT-s/14-DINOv2-reg-LP (Darcet et al., 2024), CLIP-B/16-OpenAI-FT-Vision-Encoder (Cherti et al., 2023), CLIP-B/16-Laion2B-FT-Vision-Encoder (Cherti et al., 2023), CLIP-B/32-OpenAI-FT-Vision-Encoder (Cherti et al., 2023), CLIP-B/32-Laion2B-FT-Vision-Encoder (Cherti et al., 2023) |
| (f) | 13 | Supervised models | EfficientNet-B0 (Tan & Le, 2019), EfficientNet-B1 (Tan & Le, 2019), EfficientNet-B2 (Tan & Le, 2019), EfficientNet-B3 (Tan & Le, 2019), EfficientNet-B4 (Tan & Le, 2019), EfficientNet-B5 (Tan & Le, 2019), EfficientNet-B6 (Tan & Le, 2019), EfficientNet-B7 (Tan & Le, 2019), ResNet50 (He et al., 2016), ResNeXt50-32x4d (Xie et al., 2017), ResNet18 (He et al., 2016), ResNeXt101-32x8d (Xie et al., 2017), ResNeXt50-32x4d (Xie et al., 2017) |
| | | Semi-supervised models | EfficientNet-B0 (Xie et al., 2020), EfficientNet-B1 (Xie et al., 2020), EfficientNet-B2 (Xie et al., 2020), EfficientNet-B3 (Xie et al., 2020), EfficientNet-B4 (Xie et al., 2020), EfficientNet-B5 (Xie et al., 2020), EfficientNet-B6 (Xie et al., 2020), EfficientNet-B7 (Xie et al., 2020), ResNet50 (Yalniz et al., 2019), ResNeXt50-32x4d (Yalniz et al., 2019), ResNet18 (Yalniz et al., 2019), ResNeXt101-32x8d (Yalniz et al., 2019), ResNeXt50-32x4d (Yalniz et al., 2019) |

| | | | |
|---|---|---|---|
| (g) | 19 | Supervised models | EfficientNet-v2-M (Tan & Le, 2021), EfficientNet-v2-S (Tan & Le, 2021), SeNet154 (Hu et al., 2018), RegNet-y-8gf (Radosavovic et al., 2020), RegNet-y-4gf (Radosavovic et al., 2020), RegNet-y-16gf (Radosavovic et al., 2020), RegNet-y-32gf (Radosavovic et al., 2020), ResNet101 (He et al., 2016), ResNet18 (He et al., 2016), ResNet152 (He et al., 2016), ResNet34 (He et al., 2016), ResNet50d (He et al., 2019), ResNet50 (He et al., 2016), ResNeXt50-32x4d (Xie et al., 2017), EfficientNet-B0 (Tan & Le, 2019), EfficientNet-B1 (Tan & Le, 2019), EfficientNet-B2 (Tan & Le, 2019), EfficientNet-B3 (Tan & Le, 2019), EfficientNet-B4 (Tan & Le, 2019) |
| | | A1 supervised models | EfficientNet-v2-M (Wightman et al., 2021), EfficientNet-v2-S (Wightman et al., 2021), SeNet154 (Wightman et al., 2021), RegNet-y-4gf (Wightman et al., 2021), RegNet-y-8gf (Wightman et al., 2021), RegNet-y-16gf (Wightman et al., 2021), RegNet-y-32gf (Wightman et al., 2021), ResNet101 (Wightman et al., 2021), ResNet18 (Wightman et al., 2021), ResNet152 (Wightman et al., 2021), ResNet34 (Wightman et al., 2021), ResNet50d (Wightman et al., 2021), ResNet50 (Wightman et al., 2021), ResNeXt50-32x4d (Wightman et al., 2021), EfficientNet-B0 (Wightman et al., 2021), EfficientNet-B1 (Wightman et al., 2021), EfficientNet-B2 (Wightman et al., 2021), EfficientNet-B3 (Wightman et al., 2021), EfficientNet-B4 (Wightman et al., 2021) |
| | | A2 supervised models | EfficientNet-v2-M (Wightman et al., 2021), EfficientNet-v2-S (Wightman et al., 2021), SeNet154 (Wightman et al., 2021), RegNet-y-4gf (Wightman et al., 2021), RegNet-y-8gf (Wightman et al., 2021), RegNet-y-16gf (Wightman et al., 2021), RegNet-y-32gf (Wightman et al., 2021), ResNet101 (Wightman et al., 2021), ResNet18 (Wightman et al., 2021), ResNet152 (Wightman et al., 2021), ResNet34 (Wightman et al., 2021), ResNet50d (Wightman et al., 2021), ResNet50 (Wightman et al., 2021), ResNeXt50-32x4d (Wightman et al., 2021), EfficientNet-B0 (Wightman et al., 2021), EfficientNet-B1 (Wightman et al., 2021), EfficientNet-B2 (Wightman et al., 2021), EfficientNet-B3 (Wightman et al., 2021), EfficientNet-B4 (Wightman et al., 2021) |
| | | A3 supervised models | EfficientNet-v2-M (Wightman et al., 2021), EfficientNet-v2-S (Wightman et al., 2021), SeNet154 (Wightman et al., 2021), RegNet-y-4gf (Wightman et al., 2021), RegNet-y-8gf (Wightman et al., 2021), RegNet-y-16gf (Wightman et al., 2021), RegNet-y-32gf (Wightman et al., 2021), ResNet101 (Wightman et al., 2021), ResNet18 (Wightman et al., 2021), ResNet152 (Wightman et al., 2021), ResNet34 (Wightman et al., 2021), ResNet50d (Wightman et al., 2021), ResNet50 (Wightman et al., 2021), ResNeXt50-32x4d (Wightman et al., 2021), EfficientNet-B0 (Wightman et al., 2021), EfficientNet-B1 (Wightman et al., 2021), EfficientNet-B2 (Wightman et al., 2021), EfficientNet-B3 (Wightman et al., 2021), EfficientNet-B4 (Wightman et al., 2021) |
| (h) | 46 | CNNs | RegNet-y-400mf (Radosavovic et al., 2020), RegNet-y-800mf (Radosavovic et al., 2020), RegNet-y-1-6gf (Radosavovic et al., 2020), RegNet-y-3-2gf (Radosavovic et al., 2020), RegNet-y-8gf (Radosavovic et al., 2020), RegNet-y-16gf (Radosavovic et al., 2020), EfficientNet-v2-S (Tan & Le, 2021), EfficientNet-v2-L (Tan & Le, 2021), ConvNeXt-T (Liu et al., 2022b), ConvNeXt-S (Liu et al., 2022b), ConvNeXt-B (Liu et al., 2022b), RegNet-y-4gf (Radosavovic et al., 2020), ConvNeXt-B (Liu et al., 2022b), RegNet-y-800mf (Radosavovic et al., 2020), ConvNeXt-T (Liu et al., 2022b), ConvNeXt-T (Liu et al., 2022b), RegNet-y-1-6gf (Radosavovic et al., 2020), RegNet-y-800mf (Radosavovic et al., 2020), RegNet-y-1-6gf (Radosavovic et al., 2020), EfficientNet-v2-S (Tan & Le, 2021), RegNet-y-8gf (Radosavovic et al., 2020), ConvNeXt-T (Liu et al., 2022b), ConvNeXt-S (Liu et al., 2022b), ConvNeXt-B (Liu et al., 2022b), ConvNeXt-T (Liu et al., 2022b), ConvNeXt-S (Liu et al., 2022b), RegNet-y-800mf (Radosavovic et al., 2020), RegNet-y-1-6gf (Radosavovic et al., 2020), EfficientNet-v2-S (Tan & Le, 2021), EfficientNet-v2-S (Tan & Le, 2021), ConvNeXt-T (Liu et al., 2022b), ConvNeXt-S (Liu et al., 2022b), ConvNeXt-B (Liu et al., 2022b), ConvNeXt-T (Liu et al., 2022b), EfficientNet-v2-S (Tan & Le, 2021), RegNet-y-1-6gf (Radosavovic et al., 2020), EfficientNet-v2-S (Tan & Le, 2021), EfficientNet-v2-S (Tan & Le, 2021), ConvNeXt-B (Liu et al., 2022b), EfficientNet-v2-S (Tan & Le, 2021), EfficientNet-v2-S (Tan & Le, 2021), ConvNeXt-B (Liu et al., 2022b), ConvNeXt-B (Liu et al., 2025), ConNeXt-B (Singh et al., 2023), ConvNeXtV2-T (Woo et al., 2023) |
| | | Transformers | PiT-t (Heo et al., 2021), DeiT-t (Touvron et al., 2021a), CaiT-xxs24 (Touvron et al., 2021b), LeViT-256 (Graham et al., 2021), LeViT-384 (Graham et al., 2021), XCiT-m24-16 (Ali et al., 2021), DeiT-s (Touvron et al., 2021a), MaxViT-b (Tu et al., 2022), Swin-t (Liu et al., 2021), Swin-s (Liu et al., 2021), ViT-b/32 (Dosovitskiy et al., 2021), CoaT-s-lite (Xu et al., 2021b), Swin-b (Liu et al., 2021), ConViT-t (d'Ascoli et al., 2021), ConViT-s (d'Ascoli et al., 2021), CrossViT-15† (Chen et al., 2021a), PiT-xs (Heo et al., 2021), CoaT-t-lite (Xu et al., 2021b), CoaT-mi-lite (Xu et al., 2021b), DeiT3-s (Touvron et al., 2022), DeiT3-m (Touvron et al., 2022), SwinV2-t/8 (Liu et al., 2022a), SwinV2-b/8 (Liu et al., 2022a), SwinV2-t/16 (Liu et al., 2022a), SwinV2-s/16 (Liu et al., 2022a), SwinV2-b/16 (Liu et al., 2022a), TinyViT-5M/16 (Wu et al., 2022), TinyViT-11M/16 (Wu et al., 2022), TinyViT-21M/16 (Wu et al., 2022), ViT-s/16 (Steiner et al., 2022), DaViT-t (Ding et al., 2022), DaViT-s (Ding et al., 2022), DaViT-b (Ding et al., 2022), InceptionNext-t (Yu et al., 2024), InceptionNext-s (Yu et al., 2024), FastViT-sa12 (Vasu et al., 2023), FastViT-sa24 (Vasu et al., 2023), DeiT3-s (Touvron et al., 2022), SwinV2-b/12to16 (Liu et al., 2022a), TinyViT-21M/16 (Wu et al., 2022), ViT-s/16 (Steiner et al., 2022), ViT-l/16 (Steiner et al., 2022), Swin-b (Liu et al., 2025), Swin-b (Liu et al., 2025), Hiera-T (Ryali et al., 2023) |
| (i) | 12 | Standard models | ResNet18 (He et al., 2016), ResNet34 (He et al., 2016), ResNet50 (He et al., 2016), ResNet152 (He et al., 2016), ResNet101 (He et al., 2016), DenseNet121 (Huang et al., 2017), DenseNet161 (Huang et al., 2017), DenseNet169 (Huang et al., 2017), DenseNet201 (Huang et al., 2017), ViT-b/16 (Dosovitskiy et al., 2021), ConvNeXt-T (Liu et al., 2022b), ConvNeXt-B (Liu et al., 2022b) |
| | | B-cos models | ResNet18 (Böhle et al., 2022), ResNet34 (Böhle et al., 2022), ResNet50 (Böhle et al., 2022), ResNet101 (Böhle et al., 2022), ResNet152 (Böhle et al., 2022), DenseNet121 (Böhle et al., 2022), DenseNet161 (Böhle et al., 2022), DenseNet169 (Böhle et al., 2022), DenseNet201 (Böhle et al., 2022), ViT-b/16 (Böhle et al., 2022), ConNeXt-B (Böhle et al., 2022), ConvNeXt-T (Böhle et al., 2022) |
| (j) | 24 | Standard models | ResNet50 (He et al., 2016), ResNet101 (He et al., 2016), ViT-b/16 (Dosovitskiy et al., 2021), ViT-b/32 (Dosovitskiy et al., 2021), FastViT-sa12 (Vasu et al., 2023), FastViT-sa24 (Vasu et al., 2023), FastViT-sa36 (Vasu et al., 2023), ViT-b/16 (Dosovitskiy et al., 2021), ViT-b/16 (Dosovitskiy et al., 2021), ConvNeXt-L (Liu et al., 2022b), ViT-l/16 (Dosovitskiy et al., 2021), ViT-l/16 (Dosovitskiy et al., 2021), ConvNeXt-B (Liu et al., 2022b), ConvNeXt-L (Liu et al., 2022b), ConvNeXt-L (Liu et al., 2022b), ViT-b/16 (Dosovitskiy et al., 2021), ViT-b/16 (Dosovitskiy et al., 2021), ViT-b/16 (Dosovitskiy et al., 2021), ViT-l/16 (Dosovitskiy et al., 2021), ViT-l/16 (Dosovitskiy et al., 2021), ViT-l/16 (Dosovitskiy et al., 2021), ViT-l/16 (Dosovitskiy et al., 2021), ViT-b/16 (Dosovitskiy et al., 2021), ViT-l/16 (Dosovitskiy et al., 2021) |
| | | Vision-language models | CLIP-ResNet50 (Radford et al., 2021), CLIP-ResNet101 (Radford et al., 2021), CLIP-B/16 (Radford et al., 2021), CLIP-B/32 (Radford et al., 2021), MobileCLIP-S0 (Vasu et al., 2024), MobileCLIP-S1 (Vasu et al., 2024), MobileCLIP-S2 (Vasu et al., 2024), MobileCLIP-B (Vasu et al., 2024), MobileCLIP-B (LT) (Vasu et al., 2024), SigLIP-l/16 (Zhai et al., 2023), MetaCLIP-B/16 (Xu et al., 2024), MetaCLIP-L/14 (Xu et al., 2024), CLIP-ConvNeXt-B-320px (Schuhmann et al., 2022; Wortsman et al., 2022), CLIP-ConvNeXt-L-320px (Schuhmann et al., 2022; Wortsman et al., 2022), CLIP-ConvNeXt-L (Schuhmann et al., 2022; Wortsman et al., 2022), CLIP-B/16-DataCompXL (Gadre et al., 2023), CLIP-B/16-Laion2B (Schuhmann et al., 2022; Wortsman et al., 2022), CLIP-B/16-CommonPool-XL-DFN2B (Fang et al., 2024a), CLIP-L/14-OpenAI (Radford et al., 2021), CLIP-L/14-DataCompXL (Gadre et al., 2023), CLIP-L/14-Laion2B (Schuhmann et al., 2022; Wortsman et al., 2022), CLIP-L/14-CommPool XL-DFN2B (Fang et al., 2024a), SigLIP2-b/16 (Tschannen et al., 2025), SigLIP2-l/16 (Tschannen et al., 2025) |

Table 10: *Correlation matrix with corresponding p-values (in parentheses) for all models (numerical results for Fig. 4 in the main paper).*

| | Acc. | Adv. Rob. | C-Rob. | OOD Rob | Cal. Error | Class Balance | Obj. Focus | Shape Bias | Parameters |
|---|---|---|---|---|---|---|---|---|---|
| Acc | 1.00 (.00) | | | | | | | | |
| Adv. Rob. | .44 (.00) | 1.00 (.00) | | | | | | | |
| C-Rob. | .62 (.00) | .01 (.84) | 1.00 (.00) | | | | | | |
| OOD Rob. | .35 (.00) | −0.08 (.15) | .80 (.00) | 1.00 (.00) | | | | | |
| Cal. Err. | −0.12 (.03) | .07 (.23) | .08 (.16) | .25 (.00) | 1.00 (.00) | | | | |
| Class Balance | .53 (.00) | .08 (.14) | .73 (.00) | .76 (.00) | .49 (.00) | 1.00 (.00) | | | |
| Obj. Foc. | .72 (.00) | .45 (.00) | .64 (.00) | .47 (.00) | .09 (.11) | .57 (.00) | 1.00 (.00) | | |
| Shape Bias | .26 (.00) | .17 (.00) | .61 (.00) | .62 (.00) | .24 (.00) | .54 (.00) | .52 (.00) | 1.00 (.00) | |
| Parameters | .31 (.00) | .18 (.00) | .44 (.00) | .43 (.00) | .11 (.05) | .47 (.00) | .48 (.00) | .53 (.00) | 1.00 (.00) |

## D.2 Training self-supervised models

To increase the number of models we can use to compare self-supervised and supervised models, we trained additional models for the end-to-end fine-tuning setup (E2E) and the linear probing setup (LP). For the E2E setup, we fine-tune the official pre-trained DINO (Caron et al., 2021) checkpoints for ResNet50 (He et al., 2016) and ViT-b/16 (Dosovitskiy et al., 2021). For both models, we follow the training procedure of He et al. (2022), as also done in Goldblum et al. (2023). Specifically, we train the ResNet50 for 100 epochs with an AdamW (Loshchilov & Hutter, 2017) optimizer using a batch size of 128, a weight decay of 0.05, and a learning rate of 0.001 with a cosine scheduler. We use 5 warm-up epochs with a learning rate of 0.0001. For the ViT-b/16, we follow the same procedure but with a learning rate of 0.008. Furthermore, we also fine-tuned the official pre-trained DINOv2 (Oquab et al., 2024) checkpoints for ViT-s/14, ViT-b/14, and ViT-l/14 as well as their respective version with register tokens (Darcet et al., 2024). For training all DINOv2 models, we follow Touvron et al. (2022) as done in Oquab et al. (2024). To be more precise, we trained each model for 50 epochs with a batch size of 128 for the small variant and a batch size of 64 for the base and large variants. We used an AdamW (Loshchilov & Hutter, 2017) optimizer, a weight decay of 0.02, and a learning rate of 0.0003. Additionally, we use 5 warm-up epochs with a learning rate of $10^{-6}$. We also use the ThreeAugment augmentation method that was introduced in (Touvron et al., 2022) together with color jitter and CutMix (Yun et al., 2019).

For the LP setup, we train four additional models, ViT-b/16 (with Masked Autoencoder (He et al., 2022) pre-training) and Hiera-tiny/small/base (Ryali et al., 2023). We only train the classification head of each model and follow the training of He et al. (2022), consistent with Ryali et al. (2023). We train a linear layer for 90 epochs with a LARS (You et al., 2017) optimizer using a learning rate of 0.2 and a batch size of 512.

## E Numerical results

### E.1 Correlation matrix in the main paper

As our visualization of the correlation matrix in Fig. 4 of the main paper summarizes much information and partly relies on color vision, we additionally report the numerical results in Tab. 10.

### E.2 Mean and standard deviation of each quality dimension

For our proposed QUBA score described in Sec. 5 of the main paper, we compute a representative mean and standard deviation for each quality dimension. In Tab. 11, we report these values.

Table 11: *Mean and standard deviation for each quality dimension as described in Sec. 5 of the main paper.*

| Quality Dimension | Mean | Standard Deviation |
|---|---|---|
| Accuracy | 0.80 | 0.03 |
| Adversarial Robustness | 0.19 | 0.11 |
| C-Robustness | 0.53 | 0.23 |
| OOD Robustness | 0.57 | 0.15 |
| Calibration Error | 0.0045 | 0.0027 |
| Class Balance | 0.78 | 0.02 |
| Object Focus | 0.93 | 0.02 |
| Shape Bias | 0.31 | 0.08 |
| Parameters in Mil. | 55 | 43 |

### E.3 Model zoo

We use 326 models in our large-scale study. For a comprehensive overview of all models, their configuration (architecture, training dataset, and training paradigm), their QUBA score, and their scores for each quality dimension, please refer to Tab. 12, where models are listed in the order of increasing QUBA score. Each model is implemented in PyTorch (Paszke et al., 2019); we use most models as they are, and do not change them in any way (except for training some of the self-supervised models to perform ImageNet-1k classification, *cf.* Appendix D.2). The selected models were chosen based on several criteria. We only considered models that are publicly accessible and free to use for academic research. We aimed to include the most popular models and models achieving high ImageNet-1k accuracies. We additionally included some particularly interesting models with designs that differ substantially from the more widely established models.

Table 12: *Overview of our model zoo and the corresponding numerical results.* The configuration of each model lists the architecture, the training dataset, and the training paradigm. "SL" refers to supervised learning and "AT" to adversarial training. "LP" and "E2E" indicate if a self-supervised model has been adjusted for ImageNet-1k classification by only training a linear layer (linear probing, LP) or by fine-tuning the full network on the ImageNet-1k dataset (E2E). Models are sorted according to their QUBA score in increasing order.

| Model | Configuration | QUBA Score↑ | Acc.↑ | Adv. Rob.↑ | C-Rob.↑ | OOD Rob.↑ | Cal. Error↓ | Class Balance↑ | Obj. Focus↑ | Shape Bias↑ | Params. in Mil.↓ |
|---|---|---|---|---|---|---|---|---|---|---|---|
| SigLIP2-l/16 (Tschannen et al., 2025) | ViL, WebLI(Chen et al., 2023), self-SL (E2E) | -3.45 | 0.82 | 0.32 | 0.80 | 1.06 | 0.0389 | 0.91 | 0.98 | 0.69 | 882 |
| BagNet9 (Brendel & Bethge, 2019) | CNN, IN1k(Russakovsky et al., 2015) , SL | -3.01 | 0.46 | 0.00 | 0.15 | 0.25 | 0.0052 | 0.74 | 0.81 | 0.04 | 16 |
| SigLIP-l/16 (Zhai et al., 2023) | ViL, WebLI(Chen et al., 2023), self-SL | -2.82 | 0.80 | 0.23 | 0.72 | 1.04 | 0.0384 | 0.90 | 0.97 | 0.63 | 652 |
| BagNet17 (Brendel & Bethge, 2019) | CNN, IN1k(Russakovsky et al., 2015) , SL | -2.51 | 0.58 | 0.01 | 0.22 | 0.28 | 0.0042 | 0.72 | 0.79 | 0.03 | 16 |
| CLIP-L14-Laion2B (Schuhmann et al., 2022; Wortsman et al., 2022) | ViL, Laion2B(Schuhmann et al., 2022), self-SL (E2E) | -2.43 | 0.75 | 0.01 | 0.71 | 1.07 | 0.0370 | 0.89 | 0.95 | 0.54 | 428 |
| CLIP-L14-DataCompXL (Gadre et al., 2023) | ViL, DataCompXL(Gadre et al., 2023), self-SL (E2E) | -2.10 | 0.79 | 0.07 | 0.78 | 1.05 | 0.0380 | 0.90 | 0.96 | 0.61 | 428 |
| AlexNet (Krizhevsky et al., 2012) | CNN, IN1k(Russakovsky et al., 2015) , SL | -2.09 | 0.57 | 0.04 | 0.34 | 0.57 | 0.0017 | 0.73 | 0.80 | 0.26 | 61 |
| CLIP-b/32 (Radford et al., 2021) | ViL, WIT400M(Radford et al., 2021), self-SL | -2.06 | 0.63 | 0.05 | 0.64 | 1.12 | 0.0318 | 0.88 | 0.86 | 0.58 | 151 |
| CLIP-ConvNeXt-L (Schuhmann et al., 2022; Wortsman et al., 2022) | ViL, Laion2B(Schuhmann et al., 2022), self-SL | -2.06 | 0.76 | 0.08 | 0.74 | 1.06 | 0.0370 | 0.90 | 0.94 | 0.57 | 352 |
| CLIP-ConvNeXt-L-320px (Schuhmann et al., 2022; Wortsman et al., 2022) | ViL, Laion2B(Schuhmann et al., 2022), self-SL | -2.05 | 0.77 | 0.08 | 0.74 | 1.04 | 0.0371 | 0.89 | 0.95 | 0.55 | 352 |
| MetaCLIP-L/14 (Xu et al., 2024) | ViL, MetaCLIP-400M(Xu et al., 2024), self-SL | -2.03 | 0.79 | 0.09 | 0.80 | 1.07 | 0.0382 | 0.90 | 0.97 | 0.68 | 428 |
| CLIP-L14-CommPool XL-DFN2B (Fang et al., 2024a) | ViL, DFN2B(Fang et al., 2024a), self-SL (E2E) | -2.00 | 0.81 | 0.13 | 0.77 | 1.04 | 0.0386 | 0.91 | 0.96 | 0.59 | 428 |
| SigLIP2-b/16 (Tschannen et al., 2025) | ViL, WebLI(Chen et al., 2023), self-SL (E2E) | -1.99 | 0.78 | 0.12 | 0.67 | 1.04 | 0.0380 | 0.90 | 0.96 | 0.55 | 375 |
| SqueezeNet (Iandola et al., 2016) | CNN, IN1k(Russakovsky et al., 2015) , SL | -1.97 | 0.58 | 0.01 | 0.32 | 0.48 | 0.0013 | 0.74 | 0.77 | 0.21 | 1 |
| CLIP-ResNet101 (Radford et al., 2021) | ViL, WIT400M(Radford et al., 2021), self-SL | -1.97 | 0.62 | 0.26 | 0.04 | 0.53 | 0.0257 | 0.88 | 0.91 | 0.28 | 120 |
| BagNet33 (Brendel & Bethge, 2019) | CNN, IN1k(Russakovsky et al., 2015) , SL | -1.87 | 0.65 | 0.01 | 0.28 | 0.31 | 0.0041 | 0.72 | 0.85 | 0.05 | 18 |
| Hiera-S (Ryali et al., 2023) | Transformer, IN1k(Russakovsky et al., 2015) , self-SL (LP) | -1.87 | 0.55 | 0.04 | 0.53 | 0.55 | 0.0053 | 0.76 | 0.79 | 0.47 | 35 |
| Hiera-T (Ryali et al., 2023) | Transformer, IN1k(Russakovsky et al., 2015) , self-SL (LP) | -1.85 | 0.57 | 0.03 | 0.47 | 0.56 | 0.0044 | 0.76 | 0.78 | 0.39 | 28 |
| ResNet18 (Böhle et al., 2022) | Bcos, IN1k(Russakovsky et al., 2015) , SL | -1.81 | 0.69 | 0.00 | 0.11 | 0.42 | 0.0190 | 0.79 | 0.88 | 0.22 | 12 |
| CLIP-ConvNeXt-B-320px (Schuhmann et al., 2022; Wortsman et al., 2022) | ViL, Laion2B(Schuhmann et al., 2022), self-SL | -1.78 | 0.71 | 0.05 | 0.64 | 1.05 | 0.0356 | 0.89 | 0.92 | 0.49 | 179 |
| CLIP-B16-Laion2B (Schuhmann et al., 2022; Wortsman et al., 2022) | ViL, Laion2B(Schuhmann et al., 2022), self-SL (E2E) | -1.76 | 0.70 | 0.02 | 0.62 | 1.08 | 0.0357 | 0.89 | 0.92 | 0.51 | 150 |

| Model | Configuration | QUBA Score ↑ | Acc. ↑ | Adv. Rob. ↑ | C-Rob. ↑ | OOD Rob. ↑ | Cal. Error ↓ | Class Balance ↑ | Obj. Focus ↑ | Shape Bias ↑ | Params. in Mil. ↓ |
|---|---|---|---|---|---|---|---|---|---|---|---|
| ResNet50-DINO (Caron et al., 2021) | CNN, IN1k(Russakovsky et al., 2015) , self-SL (LP) | -1.76 | 0.75 | 0.17 | 0.03 | 0.17 | 0.0084 | 0.80 | 0.69 | 0.20 | 26 |
| ShuffleNet-v2-05 (Ma et al., 2018) | CNN, IN1k(Russakovsky et al., 2015) , SL | -1.62 | 0.61 | 0.01 | 0.36 | 0.48 | 0.0028 | 0.72 | 0.85 | 0.27 | 1 |
| VGG11 (Simonyan & Zisserman, 2015) | CNN, IN1k(Russakovsky et al., 2015) , SL | -1.60 | 0.69 | 0.02 | 0.38 | 0.46 | 0.0014 | 0.74 | 0.86 | 0.11 | 133 |
| SigLIP-b/16 (Zhai et al., 2023) | ViL, WebLI(Chen et al., 2023), self-SL | -1.56 | 0.76 | 0.10 | 0.63 | 1.06 | 0.0373 | 0.90 | 0.95 | 0.49 | 203 |
| ResNet18 (Wightman et al., 2021) | CNN, IN1k(Russakovsky et al., 2015) , A3 | -1.55 | 0.68 | 0.17 | 0.04 | 0.31 | 0.0057 | 0.75 | 0.83 | 0.14 | 12 |
| ResNet152 (Böhle et al., 2022) | Bcos, IN1k(Russakovsky et al., 2015) , SL | -1.55 | 0.76 | 0.09 | 0.14 | 0.40 | 0.0207 | 0.77 | 0.91 | 0.34 | 60 |
| MetaCLIP-B/16 (Xu et al., 2024) | ViL, MetaCLIP-400M(Xu et al., 2024), self-SL | -1.54 | 0.72 | 0.03 | 0.68 | 1.09 | 0.0365 | 0.89 | 0.92 | 0.64 | 150 |
| VGG13 (Simonyan & Zisserman, 2015) | CNN, IN1k(Russakovsky et al., 2015) , SL | -1.54 | 0.70 | 0.02 | 0.37 | 0.44 | 0.0015 | 0.74 | 0.86 | 0.11 | 133 |
| MobileCLIP-S0 (Vasu et al., 2024) | ViL, DataCompDR-1B(Vasu et al., 2024), self-SL | -1.53 | 0.68 | 0.00 | 0.15 | 0.88 | 0.0325 | 0.89 | 0.88 | 0.65 | 54 |
| ResNet50 (Böhle et al., 2022) | Bcos, IN1k(Russakovsky et al., 2015) , SL | -1.51 | 0.76 | 0.01 | 0.14 | 0.41 | 0.0208 | 0.77 | 0.92 | 0.25 | 26 |
| CLIP-B16-DataCompXL (Gadre et al., 2023) | ViL, DataCompXL(Gadre et al., 2023), self-SL (E2E) | -1.49 | 0.73 | 0.03 | 0.66 | 1.07 | 0.0367 | 0.90 | 0.93 | 0.55 | 150 |
| VGG16 (Simonyan & Zisserman, 2015) | CNN, IN1k(Russakovsky et al., 2015) , SL | -1.49 | 0.72 | 0.02 | 0.40 | 0.43 | 0.0017 | 0.75 | 0.86 | 0.11 | 138 |
| VGG11-bn (Simonyan & Zisserman, 2015) | CNN, IN1k(Russakovsky et al., 2015) , SL | -1.48 | 0.70 | 0.02 | 0.42 | 0.50 | 0.0017 | 0.74 | 0.87 | 0.11 | 133 |
| CLIP-b/16 (Radford et al., 2021) | ViL, WIT400M(Radford et al., 2021), self-SL | -1.48 | 0.68 | 0.03 | 0.64 | 1.11 | 0.0288 | 0.88 | 0.92 | 0.48 | 150 |
| ResNet34 (Böhle et al., 2022) | Bcos, IN1k(Russakovsky et al., 2015) , SL | -1.46 | 0.72 | 0.01 | 0.15 | 0.46 | 0.0121 | 0.74 | 0.88 | 0.30 | 22 |
| MobileCLIP-S1 (Vasu et al., 2024) | ViL, DataCompDR-1B(Vasu et al., 2024), self-SL | -1.41 | 0.73 | 0.00 | 0.19 | 0.68 | 0.0344 | 0.89 | 0.92 | 0.65 | 85 |
| VGG19 (Simonyan & Zisserman, 2015) | CNN, IN1k(Russakovsky et al., 2015) , SL | -1.41 | 0.72 | 0.03 | 0.41 | 0.44 | 0.0017 | 0.75 | 0.87 | 0.12 | 144 |
| ViT-b/16-MAE (He et al., 2022) | Transformer, IN1k(Russakovsky et al., 2015) , self-SL (LP) | -1.41 | 0.67 | 0.04 | 0.43 | 0.44 | 0.0073 | 0.79 | 0.85 | 0.29 | 87 |
| MnasNet-075 (Tan et al., 2019) | CNN, IN1k(Russakovsky et al., 2015) , SL | -1.37 | 0.71 | 0.06 | 0.47 | 0.53 | 0.0185 | 0.81 | 0.88 | 0.21 | 3 |
| VGG13-bn (Simonyan & Zisserman, 2015) | CNN, IN1k(Russakovsky et al., 2015) , SL | -1.36 | 0.72 | 0.02 | 0.40 | 0.48 | 0.0017 | 0.75 | 0.88 | 0.12 | 133 |
| MobileCLIP-S2 (Vasu et al., 2024) | ViL, DataCompDR-1B(Vasu et al., 2024), self-SL | -1.36 | 0.74 | 0.01 | 0.18 | 0.74 | 0.0351 | 0.89 | 0.92 | 0.69 | 99 |
| CLIP-B16-CommonPool-XL-DFN2B (Fang et al., 2024a) | ViL, DFN2B(Fang et al., 2024a), self-SL (E2E) | -1.36 | 0.76 | 0.04 | 0.66 | 1.03 | 0.0374 | 0.90 | 0.94 | 0.54 | 150 |
| MobileCLIP-B (Vasu et al., 2024) | ViL, DataCompDR-1B(Vasu et al., 2024), self-SL | -1.25 | 0.77 | 0.01 | 0.66 | 1.08 | 0.0372 | 0.90 | 0.94 | 0.64 | 150 |
| MnasNet-05 (Tan et al., 2019) | CNN, IN1k(Russakovsky et al., 2015) , SL | -1.23 | 0.68 | 0.02 | 0.40 | 0.48 | 0.0041 | 0.75 | 0.84 | 0.23 | 2 |
| VGG16-bn (Simonyan & Zisserman, 2015) | CNN, IN1k(Russakovsky et al., 2015) , SL | -1.23 | 0.73 | 0.02 | 0.45 | 0.48 | 0.0020 | 0.75 | 0.89 | 0.11 | 138 |
| Hiera-B (Ryali et al., 2023) | Transformer, IN1k(Russakovsky et al., 2015) , self-SL (LP) | -1.21 | 0.65 | 0.06 | 0.58 | 0.66 | 0.0030 | 0.75 | 0.85 | 0.35 | 52 |

| Model | Configuration | QUBA Score↑ | Acc.↑ | Adv. Rob.↑ | C-Rob.↑ | OOD Rob.↑ | Cal. Error↓ | Class Balance↑ | Obj. Focus↑ | Shape Bias↑ | Params. in Mil.↓ |
|---|---|---|---|---|---|---|---|---|---|---|---|
| DenseNet121 (Böhle et al., 2022) | Bcos, IN1k(Russakovsky et al., 2015) , SL | -1.17 | 0.74 | 0.01 | 0.17 | 0.44 | 0.0099 | 0.74 | 0.90 | 0.23 | 8 |
| MobileCLIP-B (LT) (Vasu et al., 2024) | ViL, DataCompDR-1B(Vasu et al., 2024), self-SL | -1.16 | 0.77 | 0.01 | 0.79 | 1.10 | 0.0375 | 0.90 | 0.95 | 0.62 | 150 |
| EfficientNet-B0 (Wightman et al., 2021) | CNN, IN1k(Russakovsky et al., 2015) , A3 | -1.14 | 0.73 | 0.17 | 0.02 | 0.24 | 0.0054 | 0.75 | 0.86 | 0.16 | 5 |
| VGG19-bn (Simonyan & Zisserman, 2015) | CNN, IN1k(Russakovsky et al., 2015) , SL | -1.13 | 0.74 | 0.02 | 0.48 | 0.50 | 0.0021 | 0.75 | 0.89 | 0.15 | 144 |
| ShuffleNet-v2-15 (Ma et al., 2018) | CNN, IN1k(Russakovsky et al., 2015) , SL | -1.10 | 0.73 | 0.04 | 0.51 | 0.57 | 0.0143 | 0.78 | 0.89 | 0.21 | 4 |
| ConNeXt-B (Böhle et al., 2022) | Bcos, IN1k(Russakovsky et al., 2015) , SL | -1.10 | 0.77 | 0.02 | 0.52 | 0.52 | 0.0117 | 0.76 | 0.88 | 0.20 | 28 |
| ResNet34 (Wightman et al., 2021) | CNN, IN1k(Russakovsky et al., 2015) , A3 | -1.08 | 0.73 | 0.27 | 0.04 | 0.29 | 0.0051 | 0.75 | 0.86 | 0.18 | 22 |
| EfficientNet-B1 (Wightman et al., 2021) | CNN, IN1k(Russakovsky et al., 2015) , A3 | -1.07 | 0.74 | 0.11 | 0.02 | 0.22 | 0.0048 | 0.75 | 0.88 | 0.14 | 8 |
| DenseNet169 (Böhle et al., 2022) | Bcos, IN1k(Russakovsky et al., 2015) , SL | -1.06 | 0.75 | 0.02 | 0.17 | 0.43 | 0.0088 | 0.74 | 0.91 | 0.25 | 14 |
| ConvNeXt-T (Böhle et al., 2022) | Bcos, IN1k(Russakovsky et al., 2015) , SL | -1.06 | 0.79 | 0.03 | 0.57 | 0.54 | 0.0161 | 0.77 | 0.94 | 0.23 | 88 |
| ResNet18 (Wightman et al., 2021) | CNN, IN1k(Russakovsky et al., 2015) , A1 | -1.05 | 0.71 | 0.27 | 0.05 | 0.32 | 0.0052 | 0.74 | 0.87 | 0.24 | 12 |
| MobileNetV3-s (Howard et al., 2019) | CNN, IN1k(Russakovsky et al., 2015) , SL | -1.04 | 0.68 | 0.03 | 0.49 | 0.56 | 0.0017 | 0.74 | 0.84 | 0.32 | 2 |
| ShuffleNet-v2-1 (Ma et al., 2018) | CNN, IN1k(Russakovsky et al., 2015) , SL | -1.02 | 0.69 | 0.01 | 0.41 | 0.49 | 0.0031 | 0.73 | 0.88 | 0.23 | 2 |
| EfficientNet-B3 (Wightman et al., 2021) | CNN, IN1k(Russakovsky et al., 2015) , A3 | -0.99 | 0.78 | 0.22 | 0.02 | 0.21 | 0.0137 | 0.79 | 0.89 | 0.15 | 12 |
| ResNet18 (Wightman et al., 2021) | CNN, IN1k(Russakovsky et al., 2015) , A2 | -0.98 | 0.71 | 0.26 | 0.05 | 0.33 | 0.0022 | 0.73 | 0.89 | 0.20 | 12 |
| VIT-l/16 (Dosovitskiy et al., 2021) | Transformer, IN1k(Russakovsky et al., 2015) , SL | -0.93 | 0.80 | 0.24 | 0.54 | 0.50 | 0.0032 | 0.78 | 0.92 | 0.35 | 304 |
| MnasNet-13 (Tan et al., 2019) | CNN, IN1k(Russakovsky et al., 2015) , SL | -0.93 | 0.77 | 0.13 | 0.53 | 0.51 | 0.0187 | 0.81 | 0.91 | 0.19 | 6 |
| DenseNet161 (Böhle et al., 2022) | Bcos, IN1k(Russakovsky et al., 2015) , SL | -0.88 | 0.77 | 0.02 | 0.18 | 0.43 | 0.0069 | 0.73 | 0.92 | 0.29 | 29 |
| ResNet18 (He et al., 2016) | CNN, IN1k(Russakovsky et al., 2015) , SL | -0.85 | 0.70 | 0.03 | 0.47 | 0.56 | 0.0018 | 0.74 | 0.88 | 0.24 | 12 |
| VIT-l/32 (Dosovitskiy et al., 2021) | Transformer, IN1k(Russakovsky et al., 2015) , SL | -0.82 | 0.77 | 0.21 | 0.58 | 0.65 | 0.0030 | 0.77 | 0.92 | 0.57 | 306 |
| ViT-b/16 (Böhle et al., 2022) | Bcos, IN1k(Russakovsky et al., 2015) , SL | -0.78 | 0.74 | 0.01 | 0.66 | 0.55 | 0.0042 | 0.74 | 0.89 | 0.44 | 87 |
| MobileNetV2 (Sandler et al., 2018) | CNN, IN1k(Russakovsky et al., 2015) , SL | -0.77 | 0.72 | 0.02 | 0.44 | 0.50 | 0.0018 | 0.75 | 0.88 | 0.18 | 4 |
| ResNet50 (Salman et al., 2020) | CNN, IN1k(Russakovsky et al., 2015) , AT | -0.75 | 0.64 | 0.36 | 0.50 | 0.41 | 0.0037 | 0.74 | 0.88 | 0.64 | 26 |
| EfficientNet-B2 (Wightman et al., 2021) | CNN, IN1k(Russakovsky et al., 2015) , A3 | -0.75 | 0.77 | 0.16 | 0.02 | 0.20 | 0.0040 | 0.75 | 0.91 | 0.15 | 9 |
| ResNet50-DINO (Caron et al., 2021) | CNN, IN1k(Russakovsky et al., 2015) , self-SL (E2E) | -0.74 | 0.78 | 0.45 | 0.07 | 0.32 | 0.0201 | 0.84 | 0.92 | 0.23 | 26 |
| ResNet18 (Yalniz et al., 2019) | CNN, IG1B(Yalniz et al., 2019) + IN1k(Russakovsky et al., 2015) , semi-SL | -0.73 | 0.73 | 0.11 | 0.06 | 0.35 | 0.0060 | 0.77 | 0.90 | 0.27 | 12 |

| Model | Configuration | QUBA Score↑ | Acc.↑ | Adv. Rob.↑ | C-Rob.↑ | OOD Rob.↑ | Cal. Error↓ | Class Balance↑ | Obj. Focus↑ | Shape Bias↑ | Params. in Mil.↓ |
|---|---|---|---|---|---|---|---|---|---|---|---|
| ShuffleNet-v2-2 (Ma et al., 2018) | CNN, IN1k(Russakovsky et al., 2015) , SL | -0.72 | 0.76 | 0.05 | 0.55 | 0.57 | 0.0133 | 0.78 | 0.92 | 0.24 | 7 |
| ResNet50 (Wightman et al., 2021) | CNN, IN1k(Russakovsky et al., 2015) , A3 | -0.71 | 0.78 | 0.34 | 0.05 | 0.31 | 0.0063 | 0.75 | 0.91 | 0.14 | 26 |
| ResNeXt50-32x4d (Wightman et al., 2021) | CNN, IN1k(Russakovsky et al., 2015) , A3 | -0.70 | 0.79 | 0.50 | 0.04 | 0.23 | 0.0075 | 0.75 | 0.90 | 0.13 | 25 |
| CLIP-ResNet50 (Radford et al., 2021) | ViL, WIT400M(Radford et al., 2021), self-SL | -0.70 | 0.60 | 0.29 | 0.45 | 1.00 | 0.0109 | 0.94 | 0.86 | 0.20 | 102 |
| DenseNet201 (Böhle et al., 2022) | Bcos, IN1k(Russakovsky et al., 2015) , SL | -0.69 | 0.75 | 0.00 | 0.18 | 0.49 | 0.0039 | 0.74 | 0.91 | 0.27 | 20 |
| ViT-l/32 (Steiner et al., 2022) | Transformer, IN21k(Deng et al., 2009), SL | -0.67 | 0.82 | 0.09 | 0.75 | 0.63 | 0.0022 | 0.77 | 0.92 | 0.47 | 307 |
| EfficientNet-B0 (Wightman et al., 2021) | CNN, IN1k(Russakovsky et al., 2015) , A2 | -0.66 | 0.77 | 0.18 | 0.02 | 0.23 | 0.0026 | 0.74 | 0.90 | 0.18 | 5 |
| MnasNet-1 (Tan et al., 2019) | CNN, IN1k(Russakovsky et al., 2015) , SL | -0.66 | 0.73 | 0.02 | 0.45 | 0.51 | 0.0022 | 0.75 | 0.88 | 0.22 | 4 |
| ResNet101 (Böhle et al., 2022) | Bcos, IN1k(Russakovsky et al., 2015) , SL | -0.65 | 0.77 | 0.00 | 0.17 | 0.49 | 0.0042 | 0.73 | 0.93 | 0.28 | 44 |
| GoogLeNet (Szegedy et al., 2015) | CNN, IN1k(Russakovsky et al., 2015) , SL | -0.65 | 0.70 | 0.08 | 0.50 | 0.54 | 0.0032 | 0.75 | 0.91 | 0.25 | 7 |
| CLIP-L14-OpenAI (Radford et al., 2021) | ViL, WIT400M(Radford et al., 2021), self-SL (E2E) | -0.65 | 0.76 | 0.32 | 0.76 | 1.04 | 0.0110 | 0.89 | 0.94 | 0.60 | 428 |
| MobileNetV3-l (Howard et al., 2019) | CNN, IN1k(Russakovsky et al., 2015) , SL | -0.65 | 0.74 | 0.03 | 0.52 | 0.58 | 0.0026 | 0.75 | 0.87 | 0.26 | 6 |
| ResNet152 (Wightman et al., 2021) | CNN, IN1k(Russakovsky et al., 2015) , A3 | -0.65 | 0.81 | 0.40 | 0.06 | 0.30 | 0.0078 | 0.76 | 0.92 | 0.19 | 60 |
| RegNet-y-400mf (Radosavovic et al., 2020) | CNN, IN1k(Russakovsky et al., 2015) , SL | -0.63 | 0.74 | 0.02 | 0.48 | 0.54 | 0.0018 | 0.75 | 0.86 | 0.22 | 4 |
| ResNet34 (Wightman et al., 2021) | CNN, IN1k(Russakovsky et al., 2015) , A1 | -0.61 | 0.76 | 0.47 | 0.06 | 0.36 | 0.0051 | 0.74 | 0.90 | 0.29 | 22 |
| ResNet101 (Wightman et al., 2021) | CNN, IN1k(Russakovsky et al., 2015) , A3 | -0.59 | 0.80 | 0.39 | 0.06 | 0.29 | 0.0067 | 0.75 | 0.93 | 0.17 | 44 |
| ResNet34 (Wightman et al., 2021) | CNN, IN1k(Russakovsky et al., 2015) , A2 | -0.58 | 0.76 | 0.38 | 0.05 | 0.37 | 0.0026 | 0.73 | 0.91 | 0.23 | 22 |
| ResNet34 (He et al., 2016) | CNN, IN1k(Russakovsky et al., 2015) , SL | -0.55 | 0.73 | 0.03 | 0.52 | 0.55 | 0.0020 | 0.75 | 0.90 | 0.26 | 22 |
| EfficientNet-B0 (Wightman et al., 2021) | CNN, IN1k(Russakovsky et al., 2015) , A1 | -0.53 | 0.77 | 0.21 | 0.02 | 0.24 | 0.0027 | 0.74 | 0.91 | 0.26 | 5 |
| DeiT-t (Touvron et al., 2021a) | Transformer, IN1k(Russakovsky et al., 2015) , SL | -0.52 | 0.72 | 0.13 | 0.61 | 0.55 | 0.0053 | 0.77 | 0.90 | 0.24 | 6 |
| ResNet50d (He et al., 2019) | CNN, IN1k(Russakovsky et al., 2015) , SL | -0.52 | 0.79 | 0.28 | 0.05 | 0.34 | 0.0124 | 0.79 | 0.95 | 0.21 | 26 |
| ResNet50d (Wightman et al., 2021) | CNN, IN1k(Russakovsky et al., 2015) , A3 | -0.49 | 0.79 | 0.27 | 0.04 | 0.24 | 0.0030 | 0.75 | 0.93 | 0.14 | 26 |
| WRN-50-2 (Salman et al., 2020) | CNN, IN1k(Russakovsky et al., 2015) , AT | -0.46 | 0.68 | 0.38 | 0.52 | 0.47 | 0.0027 | 0.74 | 0.91 | 0.67 | 69 |
| RegNet-y-32gf (Radosavovic et al., 2020) | CNN, IN1k(Russakovsky et al., 2015) , SL | -0.44 | 0.81 | 0.05 | 0.60 | 0.54 | 0.0026 | 0.77 | 0.92 | 0.27 | 145 |
| ResNet50 (Wightman et al., 2021) | CNN, IN1k(Russakovsky et al., 2015) , A1 | -0.43 | 0.80 | 0.55 | 0.05 | 0.32 | 0.0034 | 0.73 | 0.89 | 0.21 | 26 |
| PiT-t (Heo et al., 2021) | Transformer, IN1k(Russakovsky et al., 2015) , SL | -0.42 | 0.73 | 0.13 | 0.60 | 0.59 | 0.0051 | 0.77 | 0.90 | 0.30 | 5 |
| SeNet154 (Hu et al., 2018) | CNN, IN1k(Russakovsky et al., 2015) , SL | -0.42 | 0.81 | 0.36 | 0.09 | 0.35 | 0.0091 | 0.80 | 0.93 | 0.31 | 115 |

| Model | Configuration | QUBA Score↑ | Acc.↑ | Adv. Rob.↑ | C-Rob.↑ | OOD Rob.↑ | Cal. Error↓ | Class Balance↑ | Obj. Focus↑ | Shape Bias↑ | Params. in Mil.↓ |
|---|---|---|---|---|---|---|---|---|---|---|---|
| ResNeXt50-32x4d (Yalniz et al., 2019) | CNN, YFCC100M(Thomee et al., 2016) + IN1k(Russakovsky et al., 2015) , semi-SL | -0.41 | 0.80 | 0.20 | 0.05 | 0.24 | 0.0088 | 0.78 | 0.93 | 0.24 | 25 |
| WRN-101-2 (Zagoruyko & Komodakis, 2016) | CNN, IN1k(Russakovsky et al., 2015) , SL | -0.40 | 0.79 | 0.05 | 0.57 | 0.54 | 0.0023 | 0.76 | 0.94 | 0.28 | 127 |
| RegNet-y-4gf (Wightman et al., 2021) | CNN, IN1k(Russakovsky et al., 2015) , A3 | -0.39 | 0.79 | 0.41 | 0.03 | 0.27 | 0.0050 | 0.75 | 0.94 | 0.16 | 21 |
| SeNet154 (Wightman et al., 2021) | CNN, IN1k(Russakovsky et al., 2015) , A3 | -0.38 | 0.82 | 0.50 | 0.06 | 0.26 | 0.0025 | 0.76 | 0.94 | 0.18 | 115 |
| RegNet-y-32gf (Wightman et al., 2021) | CNN, IN1k(Russakovsky et al., 2015) , A3 | -0.38 | 0.82 | 0.52 | 0.07 | 0.30 | 0.0022 | 0.76 | 0.94 | 0.18 | 145 |
| ViT-t/16 (Steiner et al., 2022) | Transformer, IN21k(Deng et al., 2009), SL | -0.37 | 0.75 | 0.01 | 0.56 | 0.54 | 0.0007 | 0.76 | 0.87 | 0.27 | 6 |
| ConViT-t (d'Ascoli et al., 2021) | Transformer, IN1k(Russakovsky et al., 2015) , SL | -0.37 | 0.73 | 0.15 | 0.62 | 0.58 | 0.0051 | 0.77 | 0.91 | 0.27 | 6 |
| EfficientNet-B1 (Wightman et al., 2021) | CNN, IN1k(Russakovsky et al., 2015) , A2 | -0.36 | 0.79 | 0.19 | 0.03 | 0.26 | 0.0021 | 0.74 | 0.92 | 0.23 | 8 |
| RegNet-y-800mf (Radosavovic et al., 2020) | CNN, IN1k(Russakovsky et al., 2015) , SL | -0.35 | 0.76 | 0.03 | 0.52 | 0.52 | 0.0017 | 0.76 | 0.89 | 0.23 | 6 |
| ResNet50 (Yalniz et al., 2019) | CNN, YFCC100M(Thomee et al., 2016) + IN1k(Russakovsky et al., 2015) , semi-SL | -0.34 | 0.79 | 0.16 | 0.07 | 0.33 | 0.0022 | 0.77 | 0.90 | 0.22 | 26 |
| RegNet-y-4gf (Radosavovic et al., 2020) | CNN, IN1k(Russakovsky et al., 2015) , SL | -0.33 | 0.79 | 0.17 | 0.02 | 0.28 | 0.0015 | 0.76 | 0.90 | 0.25 | 21 |
| DenseNet121 (Huang et al., 2017) | CNN, IN1k(Russakovsky et al., 2015) , SL | -0.33 | 0.74 | 0.06 | 0.54 | 0.51 | 0.0017 | 0.75 | 0.92 | 0.22 | 8 |
| ResNet50 (He et al., 2016) | CNN, IN1k(Russakovsky et al., 2015) , SL | -0.31 | 0.76 | 0.03 | 0.51 | 0.50 | 0.0021 | 0.75 | 0.93 | 0.22 | 26 |
| EfficientNet-v2-S (Wightman et al., 2021) | CNN, IN1k(Russakovsky et al., 2015) , A3 | -0.30 | 0.81 | 0.20 | 0.05 | 0.31 | 0.0022 | 0.76 | 0.91 | 0.17 | 24 |
| RegNet-y-16gf (Wightman et al., 2021) | CNN, IN1k(Russakovsky et al., 2015) , A3 | -0.29 | 0.81 | 0.47 | 0.04 | 0.27 | 0.0024 | 0.76 | 0.94 | 0.17 | 84 |
| EfficientNet-B1 (Wightman et al., 2021) | CNN, IN1k(Russakovsky et al., 2015) , A1 | -0.29 | 0.79 | 0.27 | 0.03 | 0.30 | 0.0022 | 0.74 | 0.92 | 0.27 | 8 |
| EfficientNet-B2 (Wightman et al., 2021) | CNN, IN1k(Russakovsky et al., 2015) , A2 | -0.29 | 0.80 | 0.25 | 0.03 | 0.25 | 0.0033 | 0.74 | 0.93 | 0.24 | 9 |
| ResNeXt101-64x4d (Xie et al., 2017) | CNN, IN1k(Russakovsky et al., 2015) , SL | -0.29 | 0.83 | 0.35 | 0.69 | 0.61 | 0.0196 | 0.85 | 0.95 | 0.24 | 84 |
| WRN-50-2 (Zagoruyko & Komodakis, 2016) | CNN, IN1k(Russakovsky et al., 2015) , SL | -0.29 | 0.78 | 0.05 | 0.54 | 0.47 | 0.0023 | 0.76 | 0.94 | 0.23 | 69 |
| ResNet50 (Wightman et al., 2021) | CNN, IN1k(Russakovsky et al., 2015) , A2 | -0.29 | 0.80 | 0.49 | 0.06 | 0.31 | 0.0027 | 0.74 | 0.92 | 0.19 | 26 |
| SeNet154 (Wightman et al., 2021) | CNN, IN1k(Russakovsky et al., 2015) , A2 | -0.28 | 0.82 | 0.46 | 0.07 | 0.29 | 0.0032 | 0.76 | 0.94 | 0.30 | 115 |
| BiTM-ResNet50x3 (Kolesnikov et al., 2020) | CNN, IN21k(Deng et al., 2009), SL | -0.28 | 0.84 | 0.04 | 0.63 | 0.69 | 0.0016 | 0.79 | 0.93 | 0.21 | 217 |
| RegNet-y-32gf (Wightman et al., 2021) | CNN, IN1k(Russakovsky et al., 2015) , A2 | -0.27 | 0.82 | 0.51 | 0.09 | 0.31 | 0.0029 | 0.76 | 0.94 | 0.31 | 145 |
| MobileNetV3-l (Howard et al., 2019) | CNN, IN21k(Deng et al., 2009), SL | -0.25 | 0.78 | 0.00 | 0.56 | 0.54 | 0.0020 | 0.76 | 0.88 | 0.27 | 6 |
| ResNet101 (He et al., 2016) | CNN, IN1k(Russakovsky et al., 2015) , SL | -0.24 | 0.77 | 0.04 | 0.57 | 0.52 | 0.0023 | 0.76 | 0.92 | 0.29 | 44 |

| Model | Configuration | QUBA Score↑ | Acc.↑ | Adv. Rob.↑ | C-Rob.↑ | OOD Rob.↑ | Cal. Error↓ | Class Balance↑ | Obj. Focus↑ | Shape Bias↑ | Params. in Mil.↓ |
|---|---|---|---|---|---|---|---|---|---|---|---|
| ResNet50d (Wightman et al., 2021) | CNN, IN1k(Russakovsky et al., 2015) , A1 | -0.24 | 0.81 | 0.40 | 0.05 | 0.27 | 0.0028 | 0.74 | 0.93 | 0.21 | 26 |
| SeNet154 (Wightman et al., 2021) | CNN, IN1k(Russakovsky et al., 2015) , A1 | -0.24 | 0.82 | 0.69 | 0.08 | 0.32 | 0.0035 | 0.75 | 0.93 | 0.30 | 115 |
| DenseNet169 (Huang et al., 2017) | CNN, IN1k(Russakovsky et al., 2015) , SL | -0.23 | 0.76 | 0.07 | 0.57 | 0.56 | 0.0025 | 0.75 | 0.93 | 0.26 | 14 |
| TinyViT-5M/16 (Wu et al., 2022) | Transformer, IN1k(Russakovsky et al., 2015) , SL | -0.21 | 0.79 | 0.15 | 0.64 | 0.58 | 0.0110 | 0.80 | 0.93 | 0.24 | 5 |
| RegNet-y-16gf (Radosavovic et al., 2020) | CNN, IN1k(Russakovsky et al., 2015) , SL | -0.21 | 0.80 | 0.03 | 0.59 | 0.57 | 0.0026 | 0.77 | 0.92 | 0.27 | 84 |
| ResNeXt50-32x4d (Yalniz et al., 2019) | CNN, IG1B(Yalniz et al., 2019) + IN1k(Russakovsky et al., 2015) , semi-SL | -0.21 | 0.82 | 0.25 | 0.06 | 0.27 | 0.0094 | 0.79 | 0.93 | 0.28 | 25 |
| ViT-b/16-DINO (Caron et al., 2021) | Transformer, IN1k(Russakovsky et al., 2015) , self-SL (LP) | -0.21 | 0.78 | 0.13 | 0.66 | 0.40 | 0.0012 | 0.77 | 0.91 | 0.36 | 87 |
| ResNeXt50-32x4d (Xie et al., 2017) | CNN, IN1k(Russakovsky et al., 2015) , SL | -0.21 | 0.78 | 0.04 | 0.54 | 0.53 | 0.0026 | 0.76 | 0.93 | 0.22 | 25 |
| RegNet-y-32gf (Wightman et al., 2021) | CNN, IN1k(Russakovsky et al., 2015) , A1 | -0.21 | 0.82 | 0.68 | 0.10 | 0.34 | 0.0029 | 0.76 | 0.94 | 0.28 | 145 |
| ResNeXt101-32x8d (Xie et al., 2017) | CNN, IN1k(Russakovsky et al., 2015) , SL | -0.21 | 0.79 | 0.07 | 0.59 | 0.55 | 0.0029 | 0.76 | 0.94 | 0.29 | 89 |
| ResNet50d (Wightman et al., 2021) | CNN, IN1k(Russakovsky et al., 2015) , A2 | -0.20 | 0.80 | 0.46 | 0.05 | 0.26 | 0.0027 | 0.74 | 0.94 | 0.20 | 26 |
| ResNeXt50-32x4d (Wightman et al., 2021) | CNN, IN1k(Russakovsky et al., 2015) , A1 | -0.20 | 0.81 | 0.59 | 0.05 | 0.31 | 0.0034 | 0.73 | 0.92 | 0.30 | 25 |
| ResNeXt101-32x8d (Yalniz et al., 2019) | CNN, IG1B(Yalniz et al., 2019) + IN1k(Russakovsky et al., 2015) , semi-SL | -0.19 | 0.84 | 0.33 | 0.06 | 0.30 | 0.0121 | 0.81 | 0.95 | 0.42 | 89 |
| RegNet-y-8gf (Wightman et al., 2021) | CNN, IN1k(Russakovsky et al., 2015) , A3 | -0.18 | 0.81 | 0.46 | 0.04 | 0.27 | 0.0030 | 0.75 | 0.94 | 0.19 | 39 |
| EfficientNet-v2-M (Wightman et al., 2021) | CNN, IN1k(Russakovsky et al., 2015) , A1 | -0.17 | 0.80 | 0.46 | 0.15 | 0.36 | 0.0036 | 0.76 | 0.92 | 0.29 | 53 |
| ResNet101 (Wightman et al., 2021) | CNN, IN1k(Russakovsky et al., 2015) , A1 | -0.17 | 0.81 | 0.61 | 0.07 | 0.35 | 0.0028 | 0.74 | 0.92 | 0.27 | 44 |
| ResNet152 (Wightman et al., 2021) | CNN, IN1k(Russakovsky et al., 2015) , A2 | -0.17 | 0.82 | 0.53 | 0.08 | 0.33 | 0.0029 | 0.75 | 0.93 | 0.23 | 60 |
| VIT-b/32 (Dosovitskiy et al., 2021) | Transformer, IN1k(Russakovsky et al., 2015) , SL | -0.16 | 0.76 | 0.16 | 0.58 | 0.58 | 0.0038 | 0.77 | 0.91 | 0.62 | 88 |
| EfficientNet-B2 (Wightman et al., 2021) | CNN, IN1k(Russakovsky et al., 2015) , A1 | -0.16 | 0.80 | 0.29 | 0.04 | 0.28 | 0.0021 | 0.74 | 0.93 | 0.26 | 9 |
| ResNet101 (Wightman et al., 2021) | CNN, IN1k(Russakovsky et al., 2015) , A2 | -0.15 | 0.81 | 0.49 | 0.07 | 0.32 | 0.0026 | 0.75 | 0.93 | 0.25 | 44 |
| ViT-s/16-DINO (Caron et al., 2021) | Transformer, IN1k(Russakovsky et al., 2015) , self-SL (LP) | -0.15 | 0.77 | 0.09 | 0.60 | 0.40 | 0.0012 | 0.76 | 0.91 | 0.29 | 23 |
| EfficientNet-B0 (Tan & Le, 2019) | CNN, IN1k(Russakovsky et al., 2015) , SL | -0.14 | 0.78 | 0.07 | 0.77 | 0.60 | 0.0034 | 0.77 | 0.89 | 0.25 | 5 |
| BiTM-ResNet152x2 (Kolesnikov et al., 2020) | CNN, IN21k(Deng et al., 2009), SL | -0.14 | 0.85 | 0.04 | 0.67 | 0.72 | 0.0016 | 0.80 | 0.95 | 0.28 | 236 |
| ResNeXt50-32x4d (Wightman et al., 2021) | CNN, IN1k(Russakovsky et al., 2015) , A2 | -0.13 | 0.80 | 0.56 | 0.05 | 0.27 | 0.0026 | 0.74 | 0.94 | 0.20 | 25 |
| RegNet-y-1-6gf (Radosavovic et al., 2020) | CNN, IN1k(Russakovsky et al., 2015) , SL | -0.12 | 0.78 | 0.03 | 0.54 | 0.54 | 0.0018 | 0.76 | 0.91 | 0.27 | 11 |

| Model | Configuration | QUBA Score ↑ | Acc. ↑ | Adv. Rob. ↑ | C-Rob. ↑ | OOD Rob. ↑ | Cal. Error ↓ | Class Balance ↑ | Obj. Focus ↑ | Shape Bias ↑ | Params. in Mil. ↓ |
|---|---|---|---|---|---|---|---|---|---|---|---|
| RegNet-y-8gf (Radosavovic et al., 2020) | CNN, IN1k(Russakovsky et al., 2015) , SL | -0.12 | 0.80 | 0.03 | 0.57 | 0.53 | 0.0023 | 0.77 | 0.91 | 0.24 | 39 |
| ResNet152 (He et al., 2016) | CNN, IN1k(Russakovsky et al., 2015) , SL | -0.11 | 0.78 | 0.06 | 0.57 | 0.55 | 0.0023 | 0.76 | 0.94 | 0.29 | 60 |
| EfficientNet-B4 (Wightman et al., 2021) | CNN, IN1k(Russakovsky et al., 2015) , A3 | -0.11 | 0.81 | 0.30 | 0.05 | 0.27 | 0.0024 | 0.76 | 0.93 | 0.22 | 19 |
| ConvNeXt-L (Liu et al., 2022b) | CNN, IN1k(Russakovsky et al., 2015) , SL | -0.10 | 0.84 | 0.12 | 0.72 | 0.67 | 0.0064 | 0.81 | 0.95 | 0.31 | 198 |
| DenseNet161 (Huang et al., 2017) | CNN, IN1k(Russakovsky et al., 2015) , SL | -0.09 | 0.77 | 0.08 | 0.59 | 0.57 | 0.0025 | 0.76 | 0.94 | 0.28 | 29 |
| EfficientNet-B1 (Tan & Le, 2019) | CNN, IN1k(Russakovsky et al., 2015) , SL | -0.09 | 0.79 | 0.14 | 0.63 | 0.58 | 0.0049 | 0.78 | 0.90 | 0.24 | 8 |
| EfficientNet-B0 (Xie et al., 2020) | CNN, JFT-300M(Hinton et al., 2015; Sun et al., 2017) + IN1k(Russakovsky et al., 2015) , semi-SL | -0.08 | 0.79 | 0.11 | 0.57 | 0.66 | 0.0037 | 0.78 | 0.89 | 0.24 | 5 |
| Xception (Chollet, 2017) | CNN, IN1k(Russakovsky et al., 2015) , SL | -0.08 | 0.79 | 0.05 | 0.57 | 0.52 | 0.0031 | 0.77 | 0.93 | 0.22 | 23 |
| EfficientNet-v2-S (Wightman et al., 2021) | CNN, IN1k(Russakovsky et al., 2015) , A1 | -0.08 | 0.81 | 0.38 | 0.07 | 0.33 | 0.0029 | 0.75 | 0.93 | 0.27 | 24 |
| RegNet-y-3-2gf (Radosavovic et al., 2020) | CNN, IN1k(Russakovsky et al., 2015) , SL | -0.08 | 0.79 | 0.03 | 0.55 | 0.52 | 0.0019 | 0.77 | 0.91 | 0.27 | 19 |
| RegNet-y-16gf (Wightman et al., 2021) | CNN, IN1k(Russakovsky et al., 2015) , A1 | -0.06 | 0.82 | 0.64 | 0.06 | 0.28 | 0.0028 | 0.75 | 0.95 | 0.28 | 84 |
| ResNet152 (Wightman et al., 2021) | CNN, IN1k(Russakovsky et al., 2015) , A1 | -0.06 | 0.82 | 0.63 | 0.07 | 0.34 | 0.0028 | 0.74 | 0.94 | 0.31 | 60 |
| EfficientNet-v2-M (Wightman et al., 2021) | CNN, IN1k(Russakovsky et al., 2015) , A3 | -0.06 | 0.82 | 0.26 | 0.08 | 0.36 | 0.0022 | 0.76 | 0.94 | 0.23 | 53 |
| ConvNeXt-L (Singh et al., 2023) | CNN, IN1k(Russakovsky et al., 2015) , AT | -0.05 | 0.77 | 0.59 | 0.66 | 0.49 | 0.0069 | 0.79 | 0.95 | 0.74 | 198 |
| EfficientNet-B3 (Wightman et al., 2021) | CNN, IN1k(Russakovsky et al., 2015) , A2 | -0.05 | 0.81 | 0.27 | 0.04 | 0.28 | 0.0022 | 0.75 | 0.95 | 0.24 | 12 |
| EfficientNet-v2-M (Wightman et al., 2021) | CNN, IN1k(Russakovsky et al., 2015) , A2 | -0.04 | 0.81 | 0.39 | 0.10 | 0.36 | 0.0032 | 0.76 | 0.93 | 0.32 | 53 |
| CoaT-t-lite (Xu et al., 2021b) | Transformer, IN1k(Russakovsky et al., 2015) , SL | -0.04 | 0.78 | 0.12 | 0.60 | 0.56 | 0.0053 | 0.79 | 0.93 | 0.25 | 6 |
| XCiT-l24-16 (Ali et al., 2021) | Transformer, IN1k(Russakovsky et al., 2015) , SL | -0.04 | 0.83 | 0.22 | 0.74 | 0.72 | 0.0037 | 0.80 | 0.93 | 0.34 | 189 |
| CrossViT-9†(Chen et al., 2021a) | Transformer, IN1k(Russakovsky et al., 2015) , SL | -0.01 | 0.77 | 0.15 | 0.64 | 0.56 | 0.0041 | 0.78 | 0.93 | 0.26 | 9 |
| DenseNet201 (Huang et al., 2017) | CNN, IN1k(Russakovsky et al., 2015) , SL | -0.00 | 0.77 | 0.08 | 0.57 | 0.59 | 0.0019 | 0.76 | 0.95 | 0.27 | 20 |
| ViT-s/16 (Steiner et al., 2022) | Transformer, IN1k(Russakovsky et al., 2015) , SL | 0.01 | 0.79 | 0.07 | 0.67 | 0.56 | 0.0018 | 0.76 | 0.91 | 0.30 | 22 |
| RegNet-y-16gf (Wightman et al., 2021) | CNN, IN1k(Russakovsky et al., 2015) , A2 | 0.01 | 0.82 | 0.58 | 0.06 | 0.30 | 0.0025 | 0.76 | 0.95 | 0.30 | 84 |
| InceptionV3 (Szegedy et al., 2016) | CNN, IN1k(Russakovsky et al., 2015) , SL | 0.02 | 0.77 | 0.16 | 0.56 | 0.55 | 0.0017 | 0.76 | 0.94 | 0.30 | 27 |
| RegNet-y-4gf (Wightman et al., 2021) | CNN, IN1k(Russakovsky et al., 2015) , A2 | 0.02 | 0.81 | 0.55 | 0.04 | 0.28 | 0.0021 | 0.75 | 0.94 | 0.24 | 21 |
| RegNet-y-8gf (Wightman et al., 2021) | CNN, IN1k(Russakovsky et al., 2015) , A2 | 0.04 | 0.82 | 0.58 | 0.05 | 0.29 | 0.0024 | 0.75 | 0.94 | 0.26 | 39 |
| EfficientNet-B3 (Wightman et al., 2021) | CNN, IN1k(Russakovsky et al., 2015) , A1 | 0.04 | 0.81 | 0.33 | 0.05 | 0.33 | 0.0023 | 0.75 | 0.94 | 0.28 | 12 |

| Model | Configuration | QUBA Score↑ | Acc.↑ | Adv. Rob.↑ | C-Rob.↑ | OOD Rob.↑ | Cal. Error↓ | Class Balance↑ | Obj. Focus↑ | Shape Bias↑ | Params. in Mil.↓ |
|---|---|---|---|---|---|---|---|---|---|---|---|
| EfficientNet-B2 (Tan & Le, 2019) | CNN, IN1k(Russakovsky et al., 2015), SL | 0.04 | 0.81 | 0.14 | 0.60 | 0.63 | 0.0041 | 0.79 | 0.90 | 0.21 | 9 |
| DeiT3-l (Touvron et al., 2022) | Transformer, IN1k(Russakovsky et al., 2015), SL | 0.05 | 0.85 | 0.32 | 0.82 | 0.76 | 0.0031 | 0.81 | 0.96 | 0.50 | 304 |
| EfficientNet-v2-S (Wightman et al., 2021) | CNN, IN1k(Russakovsky et al., 2015), A2 | 0.05 | 0.82 | 0.32 | 0.06 | 0.34 | 0.0026 | 0.76 | 0.94 | 0.28 | 24 |
| BiTM-ResNet50x1 (Kolesnikov et al., 2020) | CNN, IN21k(Deng et al., 2009), SL | 0.06 | 0.80 | 0.01 | 0.55 | 0.65 | 0.0012 | 0.78 | 0.91 | 0.19 | 26 |
| ResNet50 (Yalniz et al., 2019) | CNN, IG1B(Yalniz et al., 2019) + IN1k(Russakovsky et al., 2015), semi-SL | 0.07 | 0.81 | 0.20 | 0.08 | 0.44 | 0.0022 | 0.77 | 0.93 | 0.29 | 26 |
| EfficientNet-B4 (Tan & Le, 2019) | CNN, IN1k(Russakovsky et al., 2015), SL | 0.07 | 0.83 | 0.24 | 0.56 | 0.76 | 0.0132 | 0.83 | 0.93 | 0.21 | 19 |
| EfficientNet-B4 (Wightman et al., 2021) | CNN, IN1k(Russakovsky et al., 2015), A2 | 0.08 | 0.82 | 0.34 | 0.06 | 0.30 | 0.0023 | 0.76 | 0.93 | 0.28 | 19 |
| Swin-L (Liu et al., 2025) | Transformer, IN1k(Russakovsky et al., 2015), AT | 0.08 | 0.79 | 0.60 | 0.67 | 0.70 | 0.0074 | 0.79 | 0.94 | 0.73 | 196 |
| RegNet-y-4gf (Wightman et al., 2021) | CNN, IN1k(Russakovsky et al., 2015), A1 | 0.08 | 0.81 | 0.56 | 0.04 | 0.28 | 0.0022 | 0.75 | 0.94 | 0.28 | 21 |
| RegNet-y-8gf (Wightman et al., 2021) | CNN, IN1k(Russakovsky et al., 2015), A1 | 0.08 | 0.82 | 0.62 | 0.06 | 0.30 | 0.0023 | 0.75 | 0.94 | 0.25 | 39 |
| LeViT-128 (Graham et al., 2021) | Transformer, IN1k(Russakovsky et al., 2015), SL | 0.09 | 0.78 | 0.17 | 0.64 | 0.59 | 0.0017 | 0.76 | 0.92 | 0.26 | 9 |
| EfficientNet-B4 (Wightman et al., 2021) | CNN, IN1k(Russakovsky et al., 2015), A1 | 0.10 | 0.82 | 0.38 | 0.09 | 0.34 | 0.0028 | 0.76 | 0.95 | 0.24 | 19 |
| ConvNeXt-T (Singh et al., 2023) | CNN, IN1k(Russakovsky et al., 2015), AT | 0.10 | 0.73 | 0.51 | 0.60 | 0.61 | 0.0078 | 0.78 | 0.92 | 0.72 | 29 |
| ConvNeXt-T (Liu et al., 2022b) | CNN, IN1k(Russakovsky et al., 2015), SL | 0.10 | 0.83 | 0.09 | 0.65 | 0.60 | 0.0077 | 0.81 | 0.92 | 0.25 | 29 |
| SwinV2-t/8 (Liu et al., 2022a) | Transformer, IN1k(Russakovsky et al., 2015), SL | 0.11 | 0.82 | 0.04 | 0.66 | 0.53 | 0.0044 | 0.80 | 0.92 | 0.18 | 28 |
| PiT-xs (Heo et al., 2021) | Transformer, IN1k(Russakovsky et al., 2015), SL | 0.11 | 0.78 | 0.18 | 0.66 | 0.58 | 0.0046 | 0.78 | 0.92 | 0.31 | 11 |
| ConvNeXt-L (Liu et al., 2025) | CNN, IN1k(Russakovsky et al., 2015), AT | 0.11 | 0.78 | 0.59 | 0.65 | 0.86 | 0.0073 | 0.79 | 0.95 | 0.73 | 198 |
| InceptionV4 (Szegedy et al., 2017) | CNN, IN1k(Russakovsky et al., 2015), SL | 0.13 | 0.80 | 0.08 | 0.61 | 0.54 | 0.0014 | 0.77 | 0.94 | 0.24 | 43 |
| CoaT-mi-lite (Xu et al., 2021b) | Transformer, IN1k(Russakovsky et al., 2015), SL | 0.14 | 0.79 | 0.15 | 0.63 | 0.57 | 0.0051 | 0.79 | 0.93 | 0.27 | 11 |
| Swin-t (Liu et al., 2021) | Transformer, IN1k(Russakovsky et al., 2015), SL | 0.14 | 0.81 | 0.05 | 0.66 | 0.59 | 0.0037 | 0.79 | 0.92 | 0.22 | 28 |
| FastViT-sa12 (Vasu et al., 2023) | Transformer, IN1k(Russakovsky et al., 2015), SL | 0.15 | 0.81 | 0.14 | 0.63 | 0.57 | 0.0065 | 0.80 | 0.94 | 0.21 | 12 |
| TinyViT-11M/16 (Wu et al., 2022) | Transformer, IN1k(Russakovsky et al., 2015), SL | 0.17 | 0.82 | 0.21 | 0.69 | 0.63 | 0.0096 | 0.81 | 0.94 | 0.26 | 11 |
| MaxViT-l (Tu et al., 2022) | Transformer, IN1k(Russakovsky et al., 2015), SL | 0.17 | 0.85 | 0.41 | 0.77 | 0.71 | 0.0053 | 0.81 | 0.95 | 0.35 | 212 |
| ConvNeXt-B (Liu et al., 2022b) | CNN, IN1k(Russakovsky et al., 2015), SL | 0.18 | 0.84 | 0.12 | 0.70 | 0.66 | 0.0089 | 0.82 | 0.95 | 0.30 | 89 |
| NasNet-l (Zoph et al., 2018) | CNN, IN1k(Russakovsky et al., 2015), SL | 0.19 | 0.83 | 0.19 | 0.68 | 0.55 | 0.0029 | 0.79 | 0.94 | 0.24 | 89 |
| SwinV2-b/8 (Liu et al., 2022a) | Transformer, IN1k(Russakovsky et al., 2015), SL | 0.19 | 0.84 | 0.04 | 0.70 | 0.60 | 0.0033 | 0.81 | 0.93 | 0.20 | 88 |

| Model | Configuration | QUBA Score↑ | Acc.↑ | Adv. Rob.↑ | C-Rob.↑ | OOD Rob.↑ | Cal. Error↓ | Class Balance↑ | Obj. Focus↑ | Shape Bias↑ | Params. in Mil.↓ |
|---|---|---|---|---|---|---|---|---|---|---|---|
| VIT-b/16 (Doso-vitskiy et al., 2021) | Transformer, IN1k(Russakovsky et al., 2015) , SL | 0.20 | 0.81 | 0.18 | 0.66 | 0.56 | 0.0034 | 0.79 | 0.93 | 0.40 | 87 |
| Inception-ResNetv2 (Szegedy et al., 2017) | CNN, IN1k(Russakovsky et al., 2015) , SL | 0.20 | 0.80 | 0.15 | 0.63 | 0.56 | 0.0022 | 0.78 | 0.95 | 0.26 | 56 |
| ConvNeXt-S (Liu et al., 2022b) | CNN, IN1k(Russakovsky et al., 2015) , SL | 0.20 | 0.84 | 0.11 | 0.68 | 0.65 | 0.0083 | 0.81 | 0.94 | 0.25 | 50 |
| Swin-b (Liu et al., 2021) | Transformer, IN1k(Russakovsky et al., 2015) , SL | 0.23 | 0.84 | 0.15 | 0.71 | 0.60 | 0.0030 | 0.80 | 0.93 | 0.23 | 88 |
| EVA02-t (Fang et al., 2024b) | Transformer, IN21k(Deng et al., 2009), self-SL (E2E) | 0.23 | 0.81 | 0.10 | 0.65 | 0.58 | 0.0031 | 0.79 | 0.93 | 0.21 | 6 |
| ConvNeXt-S (Singh et al., 2023) | CNN, IN1k(Russakovsky et al., 2015) , AT | 0.23 | 0.74 | 0.52 | 0.61 | 0.76 | 0.0081 | 0.78 | 0.94 | 0.72 | 50 |
| EfficientNet-B7 (Tan & Le, 2019) | CNN, IN1k(Russakovsky et al., 2015) , SL | 0.23 | 0.84 | 0.22 | 0.67 | 0.57 | 0.0061 | 0.81 | 0.92 | 0.26 | 66 |
| ViT-L/14-DINOv2-FT (Oquab et al., 2024) | Transformer, LVD142m(Oquab et al., 2024), self-SL (E2E) | 0.24 | 0.87 | 0.32 | 0.84 | 0.86 | 0.0050 | 0.82 | 0.97 | 0.51 | 310 |
| ViT-s/16 (Singh et al., 2023) | Transformer, IN1k(Russakovsky et al., 2015) , AT | 0.26 | 0.73 | 0.49 | 0.63 | 0.72 | 0.0081 | 0.78 | 0.94 | 0.72 | 23 |
| SwinV2-s/8 (Liu et al., 2022a) | Transformer, IN1k(Russakovsky et al., 2015) , SL | 0.26 | 0.84 | 0.04 | 0.71 | 0.59 | 0.0036 | 0.80 | 0.93 | 0.20 | 50 |
| EfficientNet-v2-L (Tan & Le, 2021) | CNN, IN1k(Russakovsky et al., 2015) , SL | 0.26 | 0.85 | 0.17 | 0.74 | 0.64 | 0.0052 | 0.82 | 0.94 | 0.27 | 118 |
| ViT-l/16 (Steiner et al., 2022) | Transformer, IN21k(Deng et al., 2009), SL | 0.26 | 0.86 | 0.24 | 0.84 | 0.77 | 0.0011 | 0.81 | 0.96 | 0.54 | 304 |
| LeViT-256 (Graham et al., 2021) | Transformer, IN1k(Russakovsky et al., 2015) , SL | 0.26 | 0.82 | 0.18 | 0.66 | 0.58 | 0.0022 | 0.77 | 0.93 | 0.23 | 19 |
| EfficientFormer-l1 (Li et al., 2022b) | Transformer, IN1k(Russakovsky et al., 2015) , SL | 0.26 | 0.81 | 0.14 | 0.63 | 0.57 | 0.0014 | 0.77 | 0.94 | 0.21 | 12 |
| EfficientNet-B6 (Tan & Le, 2019) | CNN, IN1k(Russakovsky et al., 2015) , SL | 0.27 | 0.84 | 0.21 | 0.59 | 0.55 | 0.0065 | 0.82 | 0.92 | 0.23 | 43 |
| ViT-b/32 (Steiner et al., 2022) | Transformer, IN21k(Deng et al., 2009), SL | 0.27 | 0.81 | 0.09 | 0.75 | 0.47 | 0.0011 | 0.78 | 0.93 | 0.53 | 88 |
| EfficientNet-v2-S (Tan & Le, 2021) | CNN, IN1k(Russakovsky et al., 2015) , SL | 0.27 | 0.83 | 0.09 | 0.66 | 0.32 | 0.0027 | 0.80 | 0.92 | 0.24 | 22 |
| ViT-l-14-DINOv2 (Oquab et al., 2024) | Transformer, LVD142m(Oquab et al., 2024), self-SL (LP) | 0.27 | 0.86 | 0.19 | 0.86 | 0.87 | 0.0010 | 0.81 | 0.94 | 0.60 | 310 |
| Swin-s (Liu et al., 2021) | Transformer, IN1k(Russakovsky et al., 2015) , SL | 0.27 | 0.83 | 0.08 | 0.70 | 0.57 | 0.0025 | 0.80 | 0.93 | 0.20 | 50 |
| TinyViT-5M/16 (Wu et al., 2022) | Transformer, IN21k(Deng et al., 2009), SL | 0.28 | 0.81 | 0.09 | 0.67 | 0.66 | 0.0041 | 0.79 | 0.92 | 0.33 | 5 |
| Swin-B (Liu et al., 2025) | Transformer, IN1k(Russakovsky et al., 2015) , AT | 0.28 | 0.77 | 0.59 | 0.65 | 0.74 | 0.0083 | 0.79 | 0.94 | 0.73 | 88 |
| FastViT-sa24 (Vasu et al., 2023) | Transformer, IN1k(Russakovsky et al., 2015) , SL | 0.30 | 0.83 | 0.24 | 0.69 | 0.62 | 0.0089 | 0.82 | 0.94 | 0.26 | 22 |
| XCiT-m24-16 (Ali et al., 2021) | Transformer, IN1k(Russakovsky et al., 2015) , SL | 0.30 | 0.83 | 0.21 | 0.74 | 0.68 | 0.0038 | 0.80 | 0.93 | 0.33 | 84 |
| EfficientNet-v2-S (Tan & Le, 2021) | CNN, IN21k(Deng et al., 2009), SL | 0.31 | 0.83 | 0.06 | 0.68 | 0.13 | 0.0040 | 0.80 | 0.93 | 0.34 | 22 |
| EfficientFormer-l7 (Li et al., 2022b) | Transformer, IN1k(Russakovsky et al., 2015) , SL | 0.31 | 0.83 | 0.22 | 0.70 | 0.68 | 0.0023 | 0.78 | 0.94 | 0.23 | 82 |
| EfficientNet-B3 (Tan & Le, 2019) | CNN, IN1k(Russakovsky et al., 2015) , SL | 0.31 | 0.82 | 0.22 | 0.53 | 0.70 | 0.0065 | 0.80 | 0.93 | 0.27 | 12 |

| Model | Configuration | QUBA Score ↑ | Acc.↑ | Adv. Rob.↑ | C-Rob.↑ | OOD Rob.↑ | Cal. Error↓ | Class Balance↑ | Obj. Focus↑ | Shape Bias↑ | Params. in Mil.↓ |
|---|---|---|---|---|---|---|---|---|---|---|---|
| EfficientNet-B1 (Xie et al., 2020) | CNN, JFT-300M(Hinton et al., 2015; Sun et al., 2017) + IN1k(Russakovsky et al., 2015) , semi-SL | 0.31 | 0.81 | 0.15 | 0.62 | 0.72 | 0.0043 | 0.79 | 0.92 | 0.25 | 8 |
| EfficientNet-B5 (Tan & Le, 2019) | CNN, IN1k(Russakovsky et al., 2015) , SL | 0.31 | 0.83 | 0.19 | 0.55 | 0.59 | 0.0054 | 0.81 | 0.92 | 0.25 | 30 |
| EfficientNet-v2-L (Tan & Le, 2021) | CNN, IN21k(Deng et al., 2009), SL | 0.33 | 0.86 | 0.17 | 0.73 | 0.13 | 0.0046 | 0.81 | 0.96 | 0.38 | 118 |
| BiTM-ResNet101x1 (Kolesnikov et al., 2020) | CNN, IN21k(Deng et al., 2009), SL | 0.33 | 0.82 | 0.02 | 0.61 | 0.71 | 0.0011 | 0.79 | 0.94 | 0.23 | 44 |
| DeiT-s (Touvron et al., 2021a) | Transformer, IN1k(Russakovsky et al., 2015) , SL | 0.34 | 0.80 | 0.22 | 0.71 | 0.59 | 0.0043 | 0.79 | 0.94 | 0.34 | 22 |
| ViT-L/14-DINOv2-reg-LP (Darcet et al., 2024) | Transformer, LVD142m(Oquab et al., 2024), self-SL (LP) | 0.34 | 0.87 | 0.21 | 0.88 | 0.88 | 0.0010 | 0.81 | 0.94 | 0.62 | 310 |
| DeiT3-l (Touvron et al., 2022) | Transformer, IN21k(Deng et al., 2009), SL | 0.34 | 0.87 | 0.35 | 0.80 | 0.87 | 0.0024 | 0.82 | 0.96 | 0.51 | 304 |
| LeViT-384 (Graham et al., 2021) | Transformer, IN1k(Russakovsky et al., 2015) , SL | 0.34 | 0.83 | 0.19 | 0.67 | 0.63 | 0.0023 | 0.78 | 0.94 | 0.24 | 39 |
| ConvNeXt-B (Liu et al., 2025) | CNN, IN1k(Russakovsky et al., 2015) , AT | 0.34 | 0.77 | 0.57 | 0.64 | 0.62 | 0.0075 | 0.79 | 0.95 | 0.73 | 89 |
| InceptionNext-t (Yu et al., 2024) | Transformer, IN1k(Russakovsky et al., 2015) , SL | 0.35 | 0.82 | 0.21 | 0.68 | 0.64 | 0.0082 | 0.81 | 0.94 | 0.31 | 28 |
| ConvNeXtV2-N (Woo et al., 2023) | CNN, IN1k(Russakovsky et al., 2015) , self-SL (E2E) | 0.35 | 0.82 | 0.18 | 0.65 | 0.65 | 0.0069 | 0.80 | 0.94 | 0.29 | 16 |
| ConNeXt-B (Singh et al., 2023) | CNN, IN1k(Russakovsky et al., 2015) , AT | 0.36 | 0.76 | 0.57 | 0.64 | 0.80 | 0.0069 | 0.78 | 0.95 | 0.74 | 89 |
| MViTv2-l (Li et al., 2022a) | Transformer, IN1k(Russakovsky et al., 2015) , SL | 0.36 | 0.85 | 0.44 | 0.80 | 0.82 | 0.0035 | 0.81 | 0.96 | 0.34 | 218 |
| CLIP-B32-OpenAI-FT-Vision-Encoder (Cherti et al., 2023) | ViL, WIT400M(Radford et al., 2021), self-SL (E2E) | 0.36 | 0.82 | 0.17 | 0.70 | 0.80 | 0.0020 | 0.79 | 0.92 | 0.40 | 88 |
| ViT-b/16-MAE (He et al., 2022) | Transformer, IN1k(Russakovsky et al., 2015) , self-SL (E2E) | 0.36 | 0.84 | 0.25 | 0.71 | 0.58 | 0.0049 | 0.80 | 0.95 | 0.36 | 87 |
| ViT-b/16 (Singh et al., 2023) | Transformer, IN1k(Russakovsky et al., 2015) , AT | 0.36 | 0.77 | 0.55 | 0.69 | 0.50 | 0.0072 | 0.79 | 0.95 | 0.77 | 87 |
| CaiT-xxs24 (Touvron et al., 2021b) | Transformer, IN1k(Russakovsky et al., 2015) , SL | 0.36 | 0.81 | 0.16 | 0.70 | 0.57 | 0.0022 | 0.78 | 0.94 | 0.23 | 12 |
| DeiT-b (Touvron et al., 2021a) | Transformer, IN1k(Russakovsky et al., 2015) , SL | 0.37 | 0.82 | 0.23 | 0.75 | 0.62 | 0.0038 | 0.80 | 0.95 | 0.39 | 87 |
| PiT-s (Heo et al., 2021) | Transformer, IN1k(Russakovsky et al., 2015) , SL | 0.39 | 0.81 | 0.23 | 0.70 | 0.61 | 0.0042 | 0.79 | 0.93 | 0.34 | 24 |
| InceptionNext-b (Yu et al., 2024) | Transformer, IN1k(Russakovsky et al., 2015) , SL | 0.40 | 0.84 | 0.28 | 0.74 | 0.72 | 0.0081 | 0.82 | 0.94 | 0.35 | 87 |
| PiT-b (Heo et al., 2021) | Transformer, IN1k(Russakovsky et al., 2015) , SL | 0.40 | 0.82 | 0.28 | 0.74 | 0.62 | 0.0037 | 0.80 | 0.94 | 0.33 | 74 |
| EfficientNet-B2 (Xie et al., 2020) | CNN, JFT-300M(Hinton et al., 2015; Sun et al., 2017) + IN1k(Russakovsky et al., 2015) , semi-SL | 0.40 | 0.82 | 0.17 | 0.64 | 0.72 | 0.0044 | 0.80 | 0.93 | 0.24 | 9 |
| SwinV2-t/16 (Liu et al., 2022a) | Transformer, IN1k(Russakovsky et al., 2015) , SL | 0.40 | 0.83 | 0.18 | 0.70 | 0.62 | 0.0043 | 0.80 | 0.94 | 0.25 | 28 |
| EfficientFormer-l3 (Li et al., 2022b) | Transformer, IN1k(Russakovsky et al., 2015) , SL | 0.41 | 0.83 | 0.20 | 0.69 | 0.65 | 0.0020 | 0.78 | 0.94 | 0.24 | 31 |

| Model | Configuration | QUBA Score↑ | Acc.↑ | Adv. Rob.↑ | C-Rob.↑ | OOD Rob.↑ | Cal. Error↓ | Class Balance↑ | Obj. Focus↑ | Shape Bias↑ | Params. in Mil.↓ |
|---|---|---|---|---|---|---|---|---|---|---|---|
| XCiT-s24-16 (Ali et al., 2021) | Transformer, IN1k(Russakovsky et al., 2015) , SL | 0.42 | 0.83 | 0.20 | 0.74 | 0.67 | 0.0040 | 0.80 | 0.93 | 0.32 | 48 |
| ViT-S/14-DINOv2-reg-LP (Darcet et al., 2024) | Transformer, LVD142m(Oquab et al., 2024), self-SL (LP) | 0.42 | 0.81 | 0.06 | 0.68 | 0.71 | 0.0011 | 0.78 | 0.93 | 0.35 | 24 |
| FastViT-sa36 (Vasu et al., 2023) | Transformer, IN1k(Russakovsky et al., 2015) , SL | 0.43 | 0.83 | 0.27 | 0.71 | 0.63 | 0.0083 | 0.82 | 0.94 | 0.26 | 32 |
| DeiT3-s (Touvron et al., 2022) | Transformer, IN1k(Russakovsky et al., 2015) , SL | 0.43 | 0.81 | 0.15 | 0.74 | 0.59 | 0.0023 | 0.77 | 0.95 | 0.34 | 22 |
| EfficientNet-v2-M (Tan & Le, 2021) | CNN, IN1k(Russakovsky et al., 2015) , SL | 0.43 | 0.85 | 0.15 | 0.71 | 0.53 | 0.0056 | 0.82 | 0.95 | 0.25 | 54 |
| EfficientNet-v2-M (Tan & Le, 2021) | CNN, IN21k(Deng et al., 2009), SL | 0.43 | 0.85 | 0.15 | 0.71 | 0.16 | 0.0043 | 0.81 | 0.95 | 0.36 | 54 |
| ViT-s/16 (Steiner et al., 2022) | Transformer, IN21k(Deng et al., 2009), SL | 0.45 | 0.81 | 0.03 | 0.71 | 0.45 | 0.0010 | 0.78 | 0.94 | 0.38 | 22 |
| ConViT-b (d'Ascoli et al., 2021) | Transformer, IN1k(Russakovsky et al., 2015) , SL | 0.45 | 0.82 | 0.27 | 0.75 | 0.71 | 0.0043 | 0.80 | 0.95 | 0.39 | 86 |
| InceptionNext-s (Yu et al., 2024) | Transformer, IN1k(Russakovsky et al., 2015) , SL | 0.46 | 0.84 | 0.28 | 0.71 | 0.67 | 0.0072 | 0.82 | 0.95 | 0.29 | 49 |
| ViT-S/14-DINOv2-reg-LP (Darcet et al., 2024) | Transformer, LVD142m(Oquab et al., 2024), self-SL (E2E) | 0.46 | 0.81 | 0.15 | 0.72 | 0.70 | 0.0052 | 0.79 | 0.96 | 0.36 | 24 |
| ViT-s-14-DINOv2 (Oquab et al., 2024) | Transformer, LVD142m(Oquab et al., 2024), self-SL (LP) | 0.48 | 0.81 | 0.07 | 0.68 | 0.76 | 0.0010 | 0.78 | 0.92 | 0.36 | 24 |
| ConViT-s (d'Ascoli et al., 2021) | Transformer, IN1k(Russakovsky et al., 2015) , SL | 0.48 | 0.81 | 0.25 | 0.73 | 0.61 | 0.0046 | 0.80 | 0.94 | 0.34 | 28 |
| ViT-b/16-DINO (Caron et al., 2021) | Transformer, IN1k(Russakovsky et al., 2015) , self-SL (E2E) | 0.49 | 0.83 | 0.29 | 0.76 | 0.62 | 0.0039 | 0.80 | 0.94 | 0.42 | 87 |
| ViT-S/14-DINOv2-FT (Oquab et al., 2024) | Transformer, LVD142m(Oquab et al., 2024), self-SL (E2E) | 0.49 | 0.82 | 0.18 | 0.72 | 0.70 | 0.0054 | 0.80 | 0.96 | 0.33 | 24 |
| ConvNeXtV2-N (Woo et al., 2023) | CNN, IN21k(Deng et al., 2009), self-SL (E2E) | 0.50 | 0.82 | 0.12 | 0.68 | 0.63 | 0.0027 | 0.79 | 0.94 | 0.32 | 16 |
| TinyViT-21M/16 (Wu et al., 2022) | Transformer, IN1k(Russakovsky et al., 2015) , SL | 0.50 | 0.83 | 0.24 | 0.73 | 0.66 | 0.0077 | 0.81 | 0.95 | 0.32 | 21 |
| CLIP-B32-Laion2B-FT-Vision-Encoder (Cherti et al., 2023) | ViL, WIT400M(Radford et al., 2021), self-SL (E2E) | 0.50 | 0.83 | 0.17 | 0.70 | 0.82 | 0.0020 | 0.79 | 0.93 | 0.45 | 88 |
| CoaT-s-lite (Xu et al., 2021b) | Transformer, IN1k(Russakovsky et al., 2015) , SL | 0.52 | 0.82 | 0.30 | 0.70 | 0.59 | 0.0038 | 0.80 | 0.93 | 0.30 | 20 |
| MViTv2-t (Li et al., 2022a) | Transformer, IN1k(Russakovsky et al., 2015) , SL | 0.52 | 0.82 | 0.28 | 0.72 | 0.72 | 0.0054 | 0.80 | 0.95 | 0.29 | 24 |
| ViT-L/14-DINOv2-reg-LP (Darcet et al., 2024) | Transformer, LVD142m(Oquab et al., 2024), self-SL (E2E) | 0.52 | 0.88 | 0.43 | 0.87 | 0.90 | 0.0044 | 0.83 | 0.96 | 0.63 | 310 |
| DaViT-t (Ding et al., 2022) | Transformer, IN1k(Russakovsky et al., 2015) , SL | 0.54 | 0.83 | 0.19 | 0.72 | 0.62 | 0.0038 | 0.80 | 0.94 | 0.34 | 28 |
| MaxViT-b (Tu et al., 2022) | Transformer, IN1k(Russakovsky et al., 2015) , SL | 0.54 | 0.85 | 0.42 | 0.77 | 0.72 | 0.0045 | 0.81 | 0.95 | 0.35 | 120 |
| CrossViT-18†(Chen et al., 2021a) | Transformer, IN1k(Russakovsky et al., 2015) , SL | 0.54 | 0.83 | 0.26 | 0.74 | 0.62 | 0.0036 | 0.80 | 0.94 | 0.36 | 44 |
| SwinV2-b/16 (Liu et al., 2022a) | Transformer, IN1k(Russakovsky et al., 2015) , SL | 0.55 | 0.85 | 0.29 | 0.76 | 0.66 | 0.0032 | 0.81 | 0.95 | 0.30 | 88 |
| ConvNeXt-T (Liu et al., 2022b) | CNN, IN21k(Deng et al., 2009), SL | 0.56 | 0.83 | 0.16 | 0.71 | 0.69 | 0.0037 | 0.80 | 0.95 | 0.29 | 29 |

| Model | Configuration | QUBA Score↑ | Acc.↑ | Adv. Rob.↑ | C-Rob.↑ | OOD Rob.↑ | Cal. Error↓ | Class Balance↑ | Obj. Focus↑ | Shape Bias↑ | Params. in Mil.↓ |
|---|---|---|---|---|---|---|---|---|---|---|---|
| ConvNeXtV2-L (Woo et al., 2023) | CNN, IN1k(Russakovsky et al., 2015) , self-SL (E2E) | 0.57 | 0.86 | 0.35 | 0.79 | 0.78 | 0.0028 | 0.81 | 0.97 | 0.42 | 198 |
| MaxViT-t (Tu et al., 2022) | Transformer, IN1k(Russakovsky et al., 2015) , SL | 0.59 | 0.84 | 0.26 | 0.74 | 0.66 | 0.0037 | 0.81 | 0.93 | 0.28 | 31 |
| CrossViT-15†(Chen et al., 2021a) | Transformer, IN1k(Russakovsky et al., 2015) , SL | 0.59 | 0.82 | 0.25 | 0.73 | 0.62 | 0.0037 | 0.80 | 0.94 | 0.36 | 28 |
| ViT-b-14-DINOv2 (Oquab et al., 2024) | Transformer, LVD142m(Oquab et al., 2024), self-SL (LP) | 0.59 | 0.85 | 0.10 | 0.77 | 0.56 | 0.0011 | 0.80 | 0.94 | 0.45 | 90 |
| SwinV2-l/12to16 (Liu et al., 2022a) | Transformer, IN21k(Deng et al., 2009), SL | 0.59 | 0.87 | 0.29 | 0.82 | 0.80 | 0.0038 | 0.82 | 0.97 | 0.42 | 197 |
| ConvNeXtV2-T (Woo et al., 2023) | CNN, IN1k(Russakovsky et al., 2015) , self-SL (E2E) | 0.60 | 0.83 | 0.20 | 0.69 | 0.65 | 0.0026 | 0.80 | 0.94 | 0.32 | 29 |
| EfficientNet-B3 (Xie et al., 2020) | CNN, JFT-300M(Hinton et al., 2015; Sun et al., 2017) + IN1k(Russakovsky et al., 2015) , semi-SL | 0.61 | 0.84 | 0.18 | 0.67 | 0.77 | 0.0043 | 0.81 | 0.93 | 0.27 | 12 |
| SwinV2-s/16 (Liu et al., 2022a) | Transformer, IN1k(Russakovsky et al., 2015) , SL | 0.61 | 0.84 | 0.26 | 0.75 | 0.62 | 0.0033 | 0.80 | 0.94 | 0.30 | 50 |
| ConvNeXt-L (Liu et al., 2022b) | CNN, IN21k(Deng et al., 2009), SL | 0.62 | 0.87 | 0.22 | 0.81 | 0.83 | 0.0016 | 0.81 | 0.97 | 0.40 | 198 |
| DeiT3-s (Touvron et al., 2022) | Transformer, IN21k(Deng et al., 2009), SL | 0.63 | 0.83 | 0.28 | 0.67 | 0.70 | 0.0033 | 0.80 | 0.92 | 0.35 | 22 |
| DaViT-b (Ding et al., 2022) | Transformer, IN1k(Russakovsky et al., 2015) , SL | 0.64 | 0.85 | 0.30 | 0.75 | 0.68 | 0.0020 | 0.80 | 0.94 | 0.35 | 88 |
| DeiT3-b (Touvron et al., 2022) | Transformer, IN1k(Russakovsky et al., 2015) , SL | 0.64 | 0.84 | 0.27 | 0.79 | 0.70 | 0.0036 | 0.80 | 0.95 | 0.43 | 87 |
| Hiera-L (Ryali et al., 2023) | Transformer, IN1k(Russakovsky et al., 2015) , self-SL (E2E) | 0.64 | 0.86 | 0.32 | 0.81 | 0.78 | 0.0127 | 0.93 | 0.95 | 0.42 | 214 |
| CoaT-me-lite (Xu et al., 2021b) | Transformer, IN1k(Russakovsky et al., 2015) , SL | 0.64 | 0.84 | 0.22 | 0.75 | 0.66 | 0.0037 | 0.81 | 0.95 | 0.31 | 45 |
| EfficientNet-B4 (Xie et al., 2020) | CNN, JFT-300M(Hinton et al., 2015; Sun et al., 2017) + IN1k(Russakovsky et al., 2015) , semi-SL | 0.65 | 0.85 | 0.20 | 0.72 | 0.77 | 0.0050 | 0.81 | 0.92 | 0.28 | 19 |
| TinyViT-11M/16 (Wu et al., 2022) | Transformer, IN21k(Deng et al., 2009), SL | 0.65 | 0.83 | 0.11 | 0.70 | 0.73 | 0.0039 | 0.80 | 0.95 | 0.33 | 11 |
| CLIP-B16-OpenAI-FT-Vision-Encoder (Cherti et al., 2023) | ViL, WIT400M(Radford et al., 2021), self-SL (E2E) | 0.66 | 0.85 | 0.17 | 0.72 | 0.82 | 0.0024 | 0.81 | 0.94 | 0.34 | 87 |
| ViT-b/16 (Steiner et al., 2022) | Transformer, IN21k(Deng et al., 2009), SL | 0.67 | 0.85 | 0.10 | 0.79 | 0.39 | 0.0009 | 0.80 | 0.96 | 0.47 | 87 |
| ConvNeXtV2-L (Woo et al., 2023) | CNN, IN21k(Deng et al., 2009), self-SL (E2E) | 0.68 | 0.87 | 0.32 | 0.80 | 0.81 | 0.0023 | 0.81 | 0.97 | 0.45 | 198 |
| CaiT-xs24 (Touvron et al., 2021b) | Transformer, IN1k(Russakovsky et al., 2015) , SL | 0.68 | 0.84 | 0.26 | 0.75 | 0.65 | 0.0022 | 0.80 | 0.95 | 0.22 | 27 |
| DeiT3-m (Touvron et al., 2022) | Transformer, IN21k(Deng et al., 2009), SL | 0.69 | 0.85 | 0.24 | 0.69 | 0.71 | 0.0032 | 0.81 | 0.93 | 0.36 | 39 |
| MViTv2-s (Li et al., 2022a) | Transformer, IN1k(Russakovsky et al., 2015) , SL | 0.70 | 0.84 | 0.34 | 0.75 | 0.75 | 0.0048 | 0.80 | 0.96 | 0.29 | 35 |
| DeiT3-m (Touvron et al., 2022) | Transformer, IN1k(Russakovsky et al., 2015) , SL | 0.71 | 0.83 | 0.27 | 0.76 | 0.66 | 0.0040 | 0.80 | 0.95 | 0.42 | 39 |

| Model | Configuration | QUBA Score↑ | Acc.↑ | Adv. Rob.↑ | C-Rob.↑ | OOD Rob.↑ | Cal. Error↓ | Class Balance↑ | Obj. Focus↑ | Shape Bias↑ | Params. in Mil.↓ |
|---|---|---|---|---|---|---|---|---|---|---|---|
| CLIP-B16-Laion2B-FT-Vision-Encoder (Cherti et al., 2023) | ViL, WIT400M(Radford et al., 2021), self-SL (E2E) | 0.73 | 0.85 | 0.14 | 0.74 | 0.87 | 0.0027 | 0.81 | 0.95 | 0.38 | 87 |
| ViT-B/14-DINOv2-reg-LP (Darcet et al., 2024) | Transformer, LVD142m(Oquab et al., 2024), self-SL (LP) | 0.74 | 0.85 | 0.12 | 0.79 | 0.79 | 0.0011 | 0.80 | 0.94 | 0.49 | 90 |
| ConvNeXtV2-T (Woo et al., 2023) | CNN, IN21k(Deng et al., 2009), self-SL (E2E) | 0.75 | 0.84 | 0.16 | 0.72 | 0.73 | 0.0026 | 0.80 | 0.95 | 0.35 | 29 |
| DaViT-s (Ding et al., 2022) | Transformer, IN1k(Russakovsky et al., 2015) , SL | 0.76 | 0.84 | 0.24 | 0.75 | 0.70 | 0.0022 | 0.80 | 0.95 | 0.35 | 50 |
| MViTv2-b (Li et al., 2022a) | Transformer, IN1k(Russakovsky et al., 2015) , SL | 0.76 | 0.84 | 0.34 | 0.76 | 0.73 | 0.0035 | 0.81 | 0.95 | 0.33 | 52 |
| ConvNeXtV2-B (Woo et al., 2023) | CNN, IN1k(Russakovsky et al., 2015) , self-SL (E2E) | 0.76 | 0.85 | 0.28 | 0.77 | 0.70 | 0.0033 | 0.81 | 0.96 | 0.42 | 89 |
| EfficientNet-B5 (Xie et al., 2020) | CNN, JFT-300M(Hinton et al., 2015; Sun et al., 2017) + IN1k(Russakovsky et al., 2015) , semi-SL | 0.82 | 0.86 | 0.20 | 0.76 | 0.80 | 0.0049 | 0.82 | 0.94 | 0.31 | 30 |
| CaiT-s24 (Touvron et al., 2021b) | Transformer, IN1k(Russakovsky et al., 2015) , SL | 0.82 | 0.85 | 0.30 | 0.77 | 0.64 | 0.0022 | 0.80 | 0.95 | 0.29 | 47 |
| BeiTV2-b (Peng et al., 2022) | Transformer, IN1k(Russakovsky et al., 2015) , self-SL (E2E) | 0.82 | 0.86 | 0.33 | 0.79 | 0.55 | 0.0035 | 0.81 | 0.96 | 0.45 | 86 |
| Hiera-T (Ryali et al., 2023) | Transformer, IN1k(Russakovsky et al., 2015) , self-SL (E2E) | 0.82 | 0.83 | 0.19 | 0.70 | 0.66 | 0.0114 | 0.92 | 0.93 | 0.27 | 28 |
| DeiT3-b (Touvron et al., 2022) | Transformer, IN21k(Deng et al., 2009), SL | 0.83 | 0.86 | 0.26 | 0.73 | 0.76 | 0.0027 | 0.81 | 0.95 | 0.40 | 87 |
| ConvNeXt-S (Liu et al., 2022b) | CNN, IN21k(Deng et al., 2009), SL | 0.83 | 0.85 | 0.20 | 0.76 | 0.76 | 0.0022 | 0.80 | 0.96 | 0.31 | 50 |
| BeiT-b (Bao et al., 2022) | Transformer, IN21k(Deng et al., 2009), self-SL (E2E) | 0.84 | 0.85 | 0.07 | 0.79 | 0.80 | 0.0036 | 0.81 | 0.96 | 0.53 | 86 |
| ViT-B/14-DINOv2-reg-LP (Darcet et al., 2024) | Transformer, LVD142m(Oquab et al., 2024), self-SL (E2E) | 0.84 | 0.86 | 0.26 | 0.81 | 0.82 | 0.0048 | 0.81 | 0.97 | 0.41 | 90 |
| ViT-B/14-DINOv2-FT (Oquab et al., 2024) | Transformer, LVD142m(Oquab et al., 2024), self-SL (E2E) | 0.85 | 0.85 | 0.22 | 0.79 | 0.79 | 0.0050 | 0.81 | 0.97 | 0.49 | 90 |
| ConvNeXt-B (Liu et al., 2022b) | CNN, IN21k(Deng et al., 2009), SL | 0.85 | 0.86 | 0.20 | 0.78 | 0.82 | 0.0019 | 0.81 | 0.96 | 0.33 | 89 |
| EVA02-s (Fang et al., 2024b) | Transformer, IN21k(Deng et al., 2009), self-SL (E2E) | 0.88 | 0.86 | 0.17 | 0.74 | 0.77 | 0.0030 | 0.81 | 0.95 | 0.28 | 22 |
| TinyViT-21M/16 (Wu et al., 2022) | Transformer, IN21k(Deng et al., 2009), SL | 0.90 | 0.85 | 0.13 | 0.75 | 0.80 | 0.0034 | 0.81 | 0.95 | 0.37 | 21 |
| SwinV2-b/12to16 (Liu et al., 2022a) | Transformer, IN21k(Deng et al., 2009), SL | 0.90 | 0.86 | 0.26 | 0.81 | 0.81 | 0.0040 | 0.82 | 0.96 | 0.41 | 88 |
| Hiera-S (Ryali et al., 2023) | Transformer, IN1k(Russakovsky et al., 2015) , self-SL (E2E) | 0.93 | 0.84 | 0.19 | 0.74 | 0.70 | 0.0118 | 0.93 | 0.93 | 0.32 | 35 |
| EfficientNet-B7 (Xie et al., 2020) | CNN, JFT-300M(Hinton et al., 2015; Sun et al., 2017) + IN1k(Russakovsky et al., 2015) , semi-SL | 0.93 | 0.87 | 0.29 | 0.77 | 0.86 | 0.0064 | 0.83 | 0.95 | 0.44 | 66 |
| EfficientNet-B6 (Xie et al., 2020) | CNN, JFT-300M(Hinton et al., 2015; Sun et al., 2017) + IN1k(Russakovsky et al., 2015) , semi-SL | 0.94 | 0.86 | 0.25 | 0.77 | 0.83 | 0.0048 | 0.82 | 0.95 | 0.35 | 43 |

| Model | Configuration | QUBA Score ↑ | Acc.↑ | Adv. Rob.↑ | C-Rob.↑ | OOD Rob.↑ | Cal. Error↓ | Class Balance↑ | Obj. Focus↑ | Shape Bias↑ | Params. in Mil.↓ |
|---|---|---|---|---|---|---|---|---|---|---|---|
| Hiera-B (Ryali et al., 2023) | Transformer, IN1k(Russakovsky et al., 2015), self-SL (E2E) | 0.95 | 0.85 | 0.23 | 0.76 | 0.76 | 0.0130 | 0.93 | 0.94 | 0.34 | 52 |
| ConvNeXtV2-B (Woo et al., 2023) | CNN, IN21k(Deng et al., 2009), self-SL (E2E) | 0.96 | 0.87 | 0.28 | 0.79 | 0.82 | 0.0023 | 0.81 | 0.96 | 0.40 | 89 |
| Hiera-B-Plus (Ryali et al., 2023) | Transformer, IN1k(Russakovsky et al., 2015), self-SL (E2E) | 1.03 | 0.85 | 0.24 | 0.78 | 0.74 | 0.0130 | 0.93 | 0.95 | 0.43 | 70 |
| EVA02-b (Fang et al., 2024b) | Transformer, IN21k(Deng et al., 2009), self-SL (E2E) | 1.08 | 0.88 | 0.21 | 0.81 | 0.86 | 0.0039 | 0.83 | 0.97 | 0.34 | 87 |

