# OpenReview forum: "Beyond Accuracy: What Matters in Designing Well-Behaved Image Classification Models?"
_TMLR — Accepted by TMLR_

### Review · Reviewer_UDNH · 2025-08-29

**Summary Of Contributions:**

This paper studies image classification models beyond the traditional focus on accuracy, introducing a broad evaluation of nine quality dimensions including robustness, calibration, fairness, object focus, shape bias, etc. The authors analyze many backbone models under different training paradigms and architectures, revealing interesting trade-offs and synergies. They find that self-supervised and semi-supervised training generally improve multiple quality dimensions, while adversarial training strengthens robustness but harms accuracy and fairness.

**Additional Comments:**

Overall, i think it's a nice reading, but some claims and writing needs to get further tighten.

**Audience:**

Yes

**Audience Explanation:**

Audience working in deep learning computer vision will be interested.

**Broader Impact Concerns:**

None.

**Claims And Evidence:**

Yes

**Claims Explanation:**

The claims are fair within the scope of the paper, the paper comes with a clear self-noted limitation writing mentioning that these conclusions might not generalize.

**Requested Changes:**

1. The paper has offered some extensive scope and some interesting conclusions. Unfortunatley, some of the conclusions might not be objective enough, for example, when the authors mention the advantages of self-supervise training, what are exactly the setting of the self-supervised training? The authors offered some details in LP and E2E, but these read like the models are still fine-tuned in ImageNet, in this case, is it really the advantages of SST that leads to the performances, or is it more like a clear initalization of the parameters (obtained through SST) that lead to the performances. How much can we credit to SST, instead of fine-tuning with a clever set of initializations.
   - to answer this, the authors might need to offer more detailed discussions in the configurations of fine-tuning after SST, and probably need to demonstrate the performance differences along the convergence of fine-tuning after SST.

2. training on large dataset can help seems to be a very under-whelming conclusion. Similiarly, the transformer and cnn argument.
   - In addition, it has been a long discussing/debating topic of how to measure the performances w. this kind of questions. When the authors mention that training on large dataset can help, they are measuring pure performances. On the other hand, people might get different opinions if they measure the gap between other performances vs. i.i.d test accuracy.

3. Following the above comment, a more clear discussion of the measurement and a more clear note that the conclusion will depend on the measurement might be necessary.

4. The authors might have to use autoAttack for adversarial attack, which is a standard of the community now.

5. Given the scope of this paper, it's well understood that the authors will not be able to cite every paper, here are some suggestions:
     - in terms of shape bias and mitigation [1, 2, 3]
     - in terms of AT improves shape bias [3, 4]
     - a relevant work trying to offer a full package view of many of these factors of ML and computer vision, but from a more methodology view [5]


[1] The origins and prevalence of texture bias in convolutional neural networks

[2] Measuring the tendency of cnns to learn surface statistical regularities

[3] High-frequency component helps explain the generalization of convolutional neural networks

[4] Learning perceptually-aligned representations via adversarial robustness

[5] Towards Trustworthy and Aligned Machine Learning:A Data-centric Survey with Causality Perspectives

---

> ### Author Response · Authors · 2025-10-28
>
> We thank the reviewer for their insightful comments. We address all points below and have highlighted the corresponding changes in color in the revised manuscript.
>
> > Point 1: Clarifications regarding the self-supervised training setup.
>
> We thank the reviewer for catching this nuance. As described in our section on self-supervised learning, we consider two standard **transfer** settings: *(i)* linear probing (LP), where only the final classification head is trained, and *(ii)* end-to-end fine-tuning (E2E), where all model parameters are updated on ImageNet-1k. The clear gap between LP and E2E shows that self-supervised pretraining alone does not yield well-behavedness; rather, it provides a strong initialization that, when followed by full model fine-tuning, consistently improves well-behavedness. We made our manuscript more precise by clarifying that self-supervised learning provides a strong initialization, which, in combination with end-to-end fine-tuning, leads to improvements (see Abstract, Sec. 1, Fig. 1, Sec. 3.1 — paragraph on self-supervised learning, and Sec. 6).
>
> > Point 2 (first part): Unsurprising results.
>
> While the results regarding dataset size and the comparison between Transformers and CNNs might at first seem unsurprising, we are not aware of any prior work that has quantified these findings with the same scope and level of granularity as our study. In particular, the breadth of our model pool enables a very fair comparison between Transformers and CNNs, as we explicitly control for models having the same order of magnitude in parameter count and being released in similar time periods. Within this controlled setting, we found it actually quite surprising that CNNs and Transformers reach nearly identical accuracy, yet Transformers consistently dominate across all other measured metrics (see Tab. 2 (h)).
>
> > Point 2 (second part) & point 3: Relationships between the chosen metrics and the conclusions.
>
> Regarding point 3 and the related subpoint of point 2, we fully agree with the reviewer that the interpretation of results depends on the chosen evaluation metric, and that different metrics can emphasize different aspects of performance. We appreciate this valuable reminder, as it highlights a general challenge shared by most evaluation protocols and papers in the field. While this limitation is difficult to eliminate entirely, we have revised the manuscript to make this dependence more transparent. In particular, we have refined the opening paragraph of Section 2 to clarify how our metrics should be interpreted, and we have added a note in the Limitations section emphasizing that our conclusions are grounded in the reported metrics and could vary under alternative protocols.
>
> Please note that the robustness metrics in our work are computed *relative* to the i.i.d. test accuracy.
>
> > Point 4: AutoAttack
>
> We thank the reviewer for this valuable feedback on our adversarial evaluation protocol. We agree that *AutoAttack* has become a standard for measuring adversarial robustness.
> However, we opted for a more straightforward setup, as we found the success rate of *AutoAttack* to be excessively high for our purposes — nearly all non-adversarially robust models are reduced below 0.1 accuracy, effectively collapsing. Moreover, prior work has criticized *AutoAttack* for producing perturbations that significantly alter the input images, making adversarial examples easily detectable, and for exhibiting sensitivity to image resolution (Lorenz et al., "Is RobustBench/AutoAttack a suitable Benchmark for Adversarial Robustness?", 2021).
>
> To address the requested change, we have included a detailed analysis in Appendix C.4, comparing results obtained using our adversarial robustness measure and *AutoAttack*. Regarding the analysis of training strategies and architectures, nearly all conclusions remain consistent. Similarly, all statistically significant correlations remain largely unchanged. The set of top-performing models differs, as adversarial robustness measured through *AutoAttack* tends to overshadow other dimensions. This occurs because most models collapse, leading to a very small mean and standard deviation in adversarial robustness. As a result, adversarially robust models now lie several standard deviations farther from the mean than before. However, this effect can be easily mitigated by adjusting the weights of our proposed QUBA score to specific user needs, as discussed in Sec. 5. Please refer to Appendix C.4 for further details. If the reviewer considers it critical to include *AutoAttack* in the evaluation of the main text as well, we would be happy to do so upon request.

---

> > ### Author Response · Authors · 2025-10-28
> >
> > > Point 5: Additional references.
> >
> > We thank the reviewer for providing these helpful suggestions. We appreciate the pointers to related work and have incorporated the suggested references into the revised manuscript as follows:
> > - Hermann, 2019, The Origins and Prevalence of Texture Bias in Convolutional Neural Networks
> >   - Sec. 3: A[1,2,3] training.
> >   - Sec. 4: Comparison to related work.
> > - Jo, 2017, Measuring the Tendency of CNNs to Learn Surface Statistical Regularities
> >   - Sec. 4: Comparison to related work (two times)
> >   - Sec. 4: Discovering new relationships
> > - Wang, 2019, High Frequency Component Helps Explain the Generalization of Convolutional Neural Networks
> >   - Sec. 2: Shape bias
> >   - Sec. 3: Adversarial training
> > - Engstrom, 2019, Adversarial Robustness as a Prior for Learned Representations
> >   - Sec. 3: Adversarial training (two times)
> > - Liu, 2023, Towards Trustworthy and Aligned Machine Learning: A Data-centric Survey with Causality Perspectives
> >   - Sec. 3: Vision-language (ViL) models
> >   - Sec. 4: Discovering new relationships

---

### Review · Reviewer_wn2T · 2025-09-16

**Summary Of Contributions:**

This paper addresses the limitations of evaluating deep neural networks (DNNs) for image classification solely based on accuracy. The authors systematically study nine different quality dimensions—accuracy, adversarial robustness, corruption robustness, out-of-domain robustness, calibration error, class balance, object focus, shape bias, and parameter efficiency—on a large zoo of 326 backbone models. They analyze the effect of different training paradigms (supervised, adversarial, self-/semi-supervised, vision-language models) and architectures (CNNs, Transformers, B-cos, etc.) on these dimensions.

**Strengths:**

Very broad and systematic evaluation across multiple dimensions.

Comprehensive experimental study with 326 models, offering a “bird’s-eye view” that is rare in the field.

The QUBA score provides a practical tool for the community.

Well-written, clear figures and tables to support conclusions.



**Weaknesses:**

All evaluations are limited to ImageNet-1k and its variants; it is unclear if the findings generalize to other domains or tasks.

Some dimensions (e.g., interpretability, OOD detection beyond classification) are excluded.

While the breadth is excellent, depth of analysis on individual surprising findings is limited.

**Audience:**

Yes

**Audience Explanation:**

The findings are highly relevant to both academic researchers and practitioners in machine learning and computer vision. The paper addresses the community’s over-reliance on accuracy as a single metric and provides actionable insights into robustness, fairness, and calibration. The proposed benchmark and QUBA score can serve as useful tools for model selection and design, making the paper broadly interesting to TMLR’s readership.

**Broader Impact Concerns:**

The paper mainly benchmarks existing models and does not propose new architectures or training pipelines. Thus, the ethical risks are limited. However, as the study evaluates fairness only through class balance on ImageNet-1k, a dataset with its own biases, the conclusions about fairness may not fully capture real-world concerns. The authors should briefly acknowledge dataset bias as a limitation.

**Claims And Evidence:**

Yes

**Claims Explanation:**

The claims are consistently backed by large-scale empirical evidence. The authors use a diverse set of models and standardized evaluation protocols, minimizing cherry-picking. Statistical significance testing is reported in the comparisons, and the proposed QUBA score is validated for stability. While the scope is limited to ImageNet-1k, within that scope the evidence is convincing and the methodology is rigorous.

**Requested Changes:**

Critical: Clarify the limitations regarding generalisation beyond ImageNet-1k.

Strengthening (not critical): Provide deeper discussion of some surprising results (e.g., why self-supervised linear probing reduces object focus despite attention maps suggesting otherwise).

Strengthening (not critical): Discuss whether QUBA score weightings could be made adaptive or user-configurable in practice, possibly with a released toolkit.

---

> ### Author Response · Authors · 2025-10-28
>
> We thank the reviewer for their insightful comments. We address all points below and have highlighted the
> corresponding changes in color in the revised manuscript.
>
> > Point 1: Clarify the limitations regarding generalization beyond ImageNet-1k.
>
> We thank the reviewer for highlighting this important limitation. We have included it in the Limitations section (second point). To further emphasize this point, we now explicitly state in both the Introduction and Conclusion sections that our analysis focuses on ImageNet-1k classification. While we agree that this is an important consideration to discuss, we note that extending the scope beyond ImageNet-1k would go beyond the aims of the current study, an assessment that aligns with the view expressed by Reviewer kA2D.
>
> > Point 2: Deeper analysis.
>
> While we provide hypotheses throughout the paper to explain certain findings (e.g., why self-supervised pre-training improves fairness), we agree with the reviewer that our work prioritizes breadth over depth. As discussed in the Limitations section, this approach prevents us from conducting detailed analyses of specific observations, which would require dedicated empirical studies beyond the scope of this paper. Indeed, investigating theoretical connections between even two quality dimensions is often the focus of an entire publication (e.g., Minderer et al., 2021; Tsipras et al., 2019; Xiao et al., 2021). For this reason, we acknowledge and discuss this point in the paper (e.g., Limitations section), while leaving the non-critical request for a more in-depth analysis to future work.
>
> > Point 3: User-configurable QUBA score weightings.
>
> We thank the reviewer for this valuable recommendation. As discussed in Section 5, in the paragraph on different weightings, it is indeed possible — and potentially important — to adjust the weights of the QUBA score. Following the reviewer’s suggestion, we have now incorporated this adaptive weighting option into our interactive plot in the supplemental material, making this feature more accessible to a broad audience.
>
> > Point 4: ImageNet biases and fairness.
>
> We thank the reviewer for pointing out this important limitation, which we have now acknowledged in Section 2, in the paragraph on class balance.

---

### Review · Reviewer_kA2D · 2025-10-17

**Summary Of Contributions:**

The paper investigates the performance quality of various models on the ImageNet-1k dataset. It employs multiple evaluation metrics and directs its analysis toward several aspects: model behavior when the amount of training data is varied, behavior under self-supervision, and behavior under domain shifts.
A major goal of the paper is to evaluate performance beyond accuracy. The paper presents extensive and comprehensive results (and I would like to strongly emphasize this point).

**Additional Comments:**

While I have highlighted the strengths in the previous section, the most notable weakness is the limitation of the evaluation to ImageNet and image classification only. Nevertheless, the amount of experimentation conducted to support the conclusions is impressive, and asking the authors to broaden the scope would be unreasonable and would likely dilute the paper’s message. In my view, extending the scope should be the focus of future work. Therefore, I believe the paper is well-balanced and appropriate in its current form.

**Audience:**

Yes

**Audience Explanation:**

The paper delves deeply in explaining robustness with respect to classification. While the problem of image classification is of secondary importance today, yet many model trained on ImageNet are still used as starting point in further training. This paper helps to understand the consequences of pra-training

**Broader Impact Concerns:**

It is not the case

**Claims And Evidence:**

Yes

**Claims Explanation:**

The paper is highly practical in nature. The material is divided into two parts: detailed results are presented in the appendix, while the main paper focuses on describing the experiments, formulating hypotheses, summarizing results, and presenting conclusions.
In the abstract, the paper makes three claims, all of which are supported by the results. The extensive findings presented in the appendix reinforce the conclusions regarding all three aspects: domain shifts, self-supervised learning, and variations in the amount of training data.
I find the proposed metric, QUBA, to be relevant and the built conclusion also, interesting.

**Requested Changes:**

None. While there are some phrases which made be more clear, he overall text is understandable and it represent the authors. In my view, the paper is fine

---

> ### Author Response · Authors · 2025-10-28
>
> We thank the reviewer for their insightful comments and the very positive assessment of our work.
> We fully agree with the reviewer’s assessment regarding the limitation of focusing solely on ImageNet-1k classification and appreciate the recognition that the current scope remains significant, with this limitation appropriately left for future work. To make this point more transparent, and in line with Reviewer wn2T’s suggestion, we have made it more explicit in both the Introduction and Conclusion that our analysis focuses on ImageNet-1k classification.

---

### Decision · Action_Editor_KN8o · 2025-12-02

**Recommendation:** Accept as is

**Additional Comments:**

This paper presents an extremely comprehensive set of experiments on ImageNet classifiers: hundreds of models are trained and evaluated across nine metrics measuring different aspects of model performance. This allows the authors to obtain insights beyond what models achieve the highest accuracy, e.g. to understand how different training objectives affect robustness. Lastly, the authors propose a metric which takes into account all aspects of model performance rather than just accuracy.

The reviewers all praised how comprehensive the experiments were, and highlighted that the resulting insights are of clear interest to the community; I thus recommend acceptance.

**Audience:**

Yes

**Audience Explanation:**

Yes, reviewers unanimously agree.

**Claims And Evidence:**

Yes

**Claims Explanation:**

Yes, reviewers unanimously agree.